# Efficient Neural Causal Discovery without Acyclicity Constraints

**Phillip Lippe**
University of Amsterdam
QUVA lab
p.lippe@uva.nl

**Taco Cohen**
Qualcomm AI Research*
tacos@qti.qualcomm.com

**Efstratios Gavves**
University of Amsterdam
QUVA lab
e.gavves@uva.nl

## Abstract

Learning the structure of a causal graphical model using both observational and interventional data is a fundamental problem in many scientific fields. A promising direction is continuous optimization for score-based methods, which, however, require constrained optimization to enforce acyclicity or lack convergence guarantees. In this paper, we present ENCO, an efficient structure learning method for directed, acyclic causal graphs leveraging observational and interventional data. ENCO formulates the graph search as an optimization of independent edge likelihoods, with the edge orientation being modeled as a separate parameter. Consequently, we provide for ENCO convergence guarantees when interventions on all variables are available, without having to constrain the score function with respect to acyclicity. In experiments, we show that ENCO can efficiently recover graphs with hundreds of nodes, an order of magnitude larger than what was previously possible, while handling deterministic variables and discovering latent confounders.

## 1 Introduction

Uncovering and understanding causal mechanisms is an important problem not only in machine learning (Schölkopf et al., 2021; Pearl, 2009) but also in various scientific disciplines such as computational biology (Friedman et al., 2000; Sachs et al., 2005), epidemiology (Robins et al., 2000; Vandenbroucke et al., 2016), and economics (Pearl, 2009; Hicks et al., 1980). A common task of interest is *causal structure learning* (Pearl, 2009; Peters et al., 2017), which aims at learning a directed acyclic graph (DAG) in which edges represent causal relations between variables. While observational data alone is in general not sufficient to identify the DAG (Yang et al., 2018; Hauser & Bühlmann, 2012), interventional data can improve identifiability up to finding the exact graph (Eberhardt et al., 2005; Eberhardt, 2008). Unfortunately, the solution space of DAGs grows super-exponentially with the variable count, requiring efficient methods for large graphs. Current methods are typically applied to a few dozens of variables and cannot scale so well, which is imperative for modern applications like learning causal relations with gene editing interventions (Dixit et al., 2016; Macosko et al., 2015).

A promising new direction for scaling up DAG discovery methods are continuous-optimization methods (Zheng et al., 2018; 2020; Zhu et al., 2020; Ke et al., 2019; Brouillard et al., 2020; Yu et al., 2019). In contrast to score-based and constrained-based (Peters et al., 2017; Guo et al., 2020) methods, continuous-optimization methods reinterpret the search over discrete graph topologies as a continuous problem with neural networks as function approximators, for which efficient solvers are amenable. To restrict the search space to acyclic graphs, Zheng et al. (2018) first proposed to view the search as a constrained optimization problem using an augmented Lagrangian procedure to solve it. While several improvements have been explored (Zheng et al., 2020; Brouillard et al., 2020; Yu et al., 2019; Lachapelle et al., 2020), constrained optimization methods remain slow and hard to train. Alternatively, it is possible to apply a regularizer (Zhu et al., 2020; Ke et al., 2019) to penalize cyclic graphs. While simpler to optimize, methods relying on acyclicity regularizers commonly lack guarantees for finding the correct causal graph, often converging to suboptimal solutions. Despite the advances, beyond linear, continuous settings (Ng et al., 2020; Varando, 2020) continuous optimization methods still cannot scale to more than 100 variables, often due to difficulties in enforcing acyclicity.

---

*Qualcomm AI Research is an initiative of Qualcomm Technologies, Inc.

In this work, we address both problems. By modeling the orientation of an edge as a separate parameter, we can define the score function without any acyclicity constraints or regularizers. This allows for unbiased low-variance gradient estimators that scale learning to much larger graphs. Yet, if we are able to intervene on all variables, the proposed optimization is guaranteed to converge to the correct, acyclic graph. Importantly, since such interventions might not always be available, we show that our algorithm performs robustly even when intervening on fewer variables and having small sample sizes. We call our method ENCO for *Efficient Neural Causal Discovery*.

We make the following four contributions. Firstly, we propose ENCO, a causal structure learning method for observational and interventional data using continuous optimization. Different from recent methods, ENCO models the edge orientation as a separate parameter. Secondly, we derive unbiased, low-variance gradient estimators, which is crucial for scaling up the model to large numbers of variables. Thirdly, we show that ENCO is guaranteed to converge to the correct causal graph if interventions on all variables are available, despite not having any acyclicity constraints. Yet, we show in practice that the algorithm works on partial intervention sets as well. Fourthly, we extend ENCO to detecting latent confounders. In various experimental settings, ENCO recovers graphs accurately, making less than one error on graphs with 1,000 variables in less than nine hours of computation.

## 2    BACKGROUND AND RELATED WORK

### 2.1    CAUSAL GRAPHICAL MODELS

A causal graphical model (CGM) is defined by a distribution $\mathbb{P}$ over a set of random variables $\boldsymbol{X} = \{X_1, ..., X_N\}$ and a directed acyclic graph (DAG) $G = (V, E)$. Each node $i \in V$ corresponds to the random variable $X_i$ and an edge $(i, j) \in E$ represents a direct causal relation from variable $X_i$ to $X_j$: $X_i \rightarrow X_j$. A joint distribution $\mathbb{P}$ is faithful to a graph $G$ if all and only the conditional independencies present in $\mathbb{P}$ are entailed by $G$ (Pearl, 1988). Vice versa, a distribution $\mathbb{P}$ is Markov to a graph $G$ if the joint distribution can be factorized as $p(\boldsymbol{X}) = \prod_{i=1}^{N} p_i(X_i | \text{pa}(X_i))$ where $\text{pa}(X_i)$ is the set of parents of the node $i$ in $G$. An important concept which distinguishes CGMs from standard Bayesian Networks are interventions. An intervention on a variable $X_i$ describes the local replacement of its conditional distribution $p_i(X_i | \text{pa}(X_i))$ by a new distribution $\tilde{p}(X_i | \text{pa}(X_i))$. $X_i$ is thereby referred to as the intervention target. An intervention is "perfect" when the new distribution is independent of the original parents, *i.e.* $\tilde{p}(X_i | \text{pa}(X_i)) = \tilde{p}(X_i)$.

### 2.2    CAUSAL STRUCTURE LEARNING

Discovering the graph $G$ from samples of a joint distribution $\mathbb{P}$ is called *causal structure learning* or *causal discovery*, a fundamental problem in causality (Pearl, 2009; Peters et al., 2017). While often performed from observational data, i.e. samples from $\mathbb{P}$ (see Glymour et al. (2019) for an overview), we focus in this paper on algorithms that recover graphs from joint observational and interventional data. Commonly, such methods are grouped into constraint-based and score-based approaches.

**Constraint-based methods** use conditional independence tests to identify causal relations (Monti et al., 2019; Spirtes et al., 2000; Kocaoglu et al., 2019; Jaber et al., 2020; Sun et al., 2007; Hyttinen et al., 2014). For instance, the Invariant Causal Prediction (ICP) algorithm (Peters et al., 2016; Christina et al., 2018) exploits that causal mechanisms remain unchanged under an intervention except the one intervened on (Pearl, 2009; Schölkopf et al., 2012). We rely on a similar idea by testing for mechanisms that generalize from the observational to the interventional setting. Another line of work is to extend methods working on observations only to interventions by incorporating those as additional constraints in the structure learning process (Mooij et al., 2020; Jaber et al., 2020).

**Score-based methods**, on the other hand, search through the space of all possible causal structures with the goal of optimizing a specified metric (Tsamardinos et al., 2006; Ke et al., 2019; Goudet et al., 2017; Zhu et al., 2020). This metric, also referred to as score function, is usually a combination of how well the structure fits the data, for instance in terms of log-likelihood, as well as regularizers for encouraging sparsity. Since the search space of DAGs is super-exponential in the number of nodes, many methods rely on a greedy search, yet returning graphs in the true equivalence class (Meek, 1997; Hauser & Bühlmann, 2012; Wang et al., 2017; Yang et al., 2018). For instance, GIES (Hauser & Bühlmann, 2012) repeatedly adds, removes, and flips the directions of edges in a proposal graph

Figure 1: Visualization of the two training stages of ENCO, distribution fitting and graph fitting, on an example graph with 3 variables ($X_1, X_2, X_3$). The graph on the right further shows how the parameters $\gamma$ and $\theta$ correspond to edge probabilities. We learn those parameters by comparing multiple graph samples on how well they generalize from observational to interventional data.

until no higher-scoring graph can be found. The Interventional Greedy SP (IGSP) algorithm (Wang et al., 2017) is a hybrid method using conditional independence tests in its score function.

**Continuous-optimization methods** are score-based methods that avoid the combinatorial greedy search over DAGs by using gradient-based methods (Zheng et al., 2018; Ke et al., 2019; Lachapelle et al., 2020; Yu et al., 2019; Zheng et al., 2020; Zhu et al., 2020; Brouillard et al., 2020). Thereby, the adjacency matrix is parameterized by weights that represent linear factors or probabilities of having an edge between a pair of nodes. The main challenge of such methods is how to limit the search space to acyclic graphs. One common approach is to view the search as a constrained optimization problem and deploy an augmented Lagrangian procedure to solve it (Zheng et al., 2018; 2020; Yu et al., 2019; Brouillard et al., 2020), including NOTEARS (Zheng et al., 2018) and DCDI (Brouillard et al., 2020). Alternatively, Ke et al. (2019); Ng et al. (2020) propose to use a regularization term penalizing cyclic graphs while allowing unconstrained optimization. However, the regularizer must be designed and weighted such that the correct, acyclic causal graph is the global optimum of the score function.

## 3 EFFICIENT NEURAL CAUSAL DISCOVERY

### 3.1 SCOPE AND ASSUMPTIONS

We consider the task of finding a directed acyclic graph $G = (V, E)$ with $N$ variables of an unknown CGM given observational and interventional samples. Firstly, we assume that: (1) The CGM is causally sufficient, *i.e.*, all common causes of variables are included and observable; (2) We have $N$ interventional datasets, each sparsely intervening on a different variable; (3) The interventions are "perfect" and "stochastic", meaning the intervention does not set the variable necessarily to a single value. Thereby, we do not strictly require faithfulness, thus also recovering some graphs violating faithfulness. We emphasize that we place no constraints on the domains of the variables (they can be discrete, continuous, or mixed) or the distributions of the interventions. We discuss later how to extend the algorithm to infer causal mechanisms in graphs with latent confounding causal variables. Further, we discuss how to extend the algorithm to support interventions to subsets of variables only.

### 3.2 OVERVIEW

ENCO learns a causal graph from observational and interventional data by modelling a probability for every possible directed edge between pairs of variables. The goal is that the probabilities corresponding to the edges of the ground truth graph converge to one, while the probabilities of all other edges converge to zero. For this to happen, we exploit the idea of independent causal mechanisms (Pearl, 2009; Peters et al., 2016), according to which the conditional distributions for all variables in the ground-truth CGM stay invariant under an intervention, except for the intervened ones. By contrast, for graphs modelling the same joint distribution but with a flipped or additional edge, this does not hold (Peters et al., 2016). In short, we search for the graph which generalizes best from observational to interventional data. To implement the optimization, we alternate between two learning stages, that is distribution fitting and graph fitting, visually summarized in Figure 1. Ke et al. (2019) proposed a similar two-phase framework, but importantly differ in the graph parameterization.

**Distribution fitting** trains a neural network $f_{\phi_i}$ per variable $X_i$ parameterized by $\phi_i$ to model its observational, conditional data distribution $p(X_i|...)$. The input to the network are all other variables,

$\boldsymbol{X}_{-i}$. For simplicity, we want this neural network to model the conditional of the variable $X_i$ with respect to any possible set of parent variables. We, therefore, apply a dropout-like scheme to the input to simulate different sets of parents, similar as (Ke et al., 2019; Ivanov et al., 2019; Li et al., 2020; Brouillard et al., 2020). In that case, during training, we randomly set an input variable $X_j$ to zero based on the probability of its corresponding edge $X_j \rightarrow X_i$, and minimize

$$\min_{\phi_i} \mathbb{E}_{\boldsymbol{X}} \mathbb{E}_{\boldsymbol{M}} \left[ -\log f_{\phi_i}(X_i; \boldsymbol{M}_{-i} \odot \boldsymbol{X}_{-i}) \right], \tag{1}$$

where $M_j \sim \text{Ber}(p(X_j \rightarrow X_i))$. For categorical random variables $X_i$, we apply a softmax output activation for $f_{\phi_i}$, and for continuous ones, we use Normalizing Flows (Rezende & Mohamed, 2015).

**Graph fitting** uses the learned networks to score and compare different graphs on interventional data. For parameterizing the edge probabilities, we use two sets of parameters: $\boldsymbol{\gamma} \in \mathbb{R}^{N \times N}$ represents the existence of edges in a graph, and $\boldsymbol{\theta} \in \mathbb{R}^{N \times N}$ the orientation of the edges. The likelihood of an edge is determined by $p(X_i \rightarrow X_j) = \sigma(\gamma_{ij}) \cdot \sigma(\theta_{ij})$, with $\sigma(...)$ being the sigmoid function and $\theta_{ij} = -\theta_{ji}$. The probability of the two orientations always sum to one. The benefit of separating the edge probabilities into two independent parameters $\boldsymbol{\gamma}$ and $\boldsymbol{\theta}$ is that it gives us more control over the gradient updates. The existence of an (undirected) edge can usually be already learned from observational or arbitrary interventional data alone, excluding deterministic variables (Pearl, 2009). In contrast, the orientation can only be reliably detected from data for which an intervention is performed on its adjacent nodes, *i.e.*, $X_i$ or $X_j$ for learning $\theta_{ij}$. While other interventions eventually provide information on the edge direction, *e.g.*, intervening on a node $X_k$ which is a child of $X_i$ and a parent of $X_j$, we do not know the relation of $X_k$ to $X_i$ and $X_j$ at this stage, as we are in the process of learning the structure. Despite having just one variable for the orientation, $\gamma_{ij}$ and $\gamma_{ji}$ are learned as two separate parameters. One reason is that on interventional data, an edge can improve the log-likelihood estimate in one direction, but not necessarily the other, leading to conflicting gradients.

We optimize the graph parameters $\boldsymbol{\gamma}$ and $\boldsymbol{\theta}$ by minimizing

$$\tilde{\mathcal{L}} = \mathbb{E}_{\hat{I} \sim p_I(I)} \mathbb{E}_{\tilde{p}_{\hat{I}}(\boldsymbol{X})} \mathbb{E}_{p_{\boldsymbol{\gamma},\boldsymbol{\theta}}(C)} \left[ \sum_{i=1}^{N} \mathcal{L}_C(X_i) \right] + \lambda_{\text{sparse}} \sum_{i=1}^{N} \sum_{j=1}^{N} \sigma(\gamma_{ij}) \cdot \sigma(\theta_{ij}) \tag{2}$$

where $p_I(I)$ is the distribution over which variable to intervene on (usually uniform), and $\tilde{p}_{\hat{I}}(\boldsymbol{X})$ the joint distribution of all variables under the intervention $\hat{I}$. In other words, these two distributions represent our interventional data distribution. With $p_{\boldsymbol{\gamma},\boldsymbol{\theta}}(C)$, we denote the distribution over adjacency matrices $C$ under $\boldsymbol{\gamma}, \boldsymbol{\theta}$, where $C_{ij} \sim \text{Ber}(\sigma(\gamma_{ij})\sigma(\theta_{ij}))$. $\mathcal{L}_C(X_i)$ is the negative log-likelihood estimate of variable $X_i$ conditioned on the parents according to $C$: $\mathcal{L}_C(X_i) = -\log f_{\phi_i}(X_i; C_{\cdot,i} \odot \boldsymbol{X}_{-i})$. The second term of Equation 2 is an $\ell_1$-regularizer on the edge probabilities. It acts as a prior, selecting the sparsest graph of those with similar likelihood estimates by removing redundant edges.

**Prediction.** Alternating between the distribution and graph fitting stages allows us to fine-tune the neural networks to the most probable parent sets along the training. After training, we obtain a graph prediction by selecting the edges for which $\sigma(\gamma_{ij})$ and $\sigma(\theta_{ij})$ are greater than 0.5. The orientation parameters prevent loops between any two variables, since $\sigma(\theta_{ij})$ can only be greater than 0.5 in one direction. Although the orientation parameters do not guarantee the absence of loops with more variable, we show that under certain conditions ENCO yet converges to the correct, acyclic graph.

### 3.3 Low-variance gradient estimators for edge parameters

To update $\boldsymbol{\gamma}$ and $\boldsymbol{\theta}$ based on Equation 2, we need to determine their gradients through the expectation $\mathbb{E}_{p_{\boldsymbol{\gamma},\boldsymbol{\theta}}(C)}$, where $C$ is a discrete variable. For this, we apply REINFORCE (Williams, 1992). For clarity of exposition, we limit the discussion here to the final results and provide the detailed derivations in Appendix A. For parameter $\gamma_{ij}$, we obtain the following gradient:

$$\frac{\partial}{\partial \gamma_{ij}} \tilde{\mathcal{L}} = \sigma'(\gamma_{ij}) \cdot \sigma(\theta_{ij}) \cdot \mathbb{E}_{\boldsymbol{X}, C_{-ij}} \left[ \mathcal{L}_{X_i \rightarrow X_j}(X_j) - \mathcal{L}_{X_i \nrightarrow X_j}(X_j) + \lambda_{\text{sparse}} \right] \tag{3}$$

where $\mathbb{E}_{\boldsymbol{X}, C_{-ij}}$ summarizes for brevity the three expectations in Equation 2, excluding the edge $X_i \rightarrow X_j$ from $C$. Further, $\mathcal{L}_{X_i \rightarrow X_j}(X_j)$ denotes the negative log-likelihood for $X_j$, if we include the edge $X_i \rightarrow X_j$ to the adjacency matrix $C_{-ij}$, *i.e.*, $C_{ij} = 1$, and $\mathcal{L}_{X_i \nrightarrow X_j}(X_j)$ if we set $C_{ij} = 0$. The gradient in Equation 3 can be intuitively explained: if by the addition of the edge $X_i \rightarrow X_j$, the

log-likelihood estimate of $X_j$ is improved by more than $\lambda_{\text{sparse}}$, we increase the corresponding edge parameter $\gamma_{ij}$; otherwise, we decrease it.

We derive the gradients for the orientation parameters $\boldsymbol{\theta}$ similarly. As mentioned before, we only take the gradients for $\theta_{ij}$ when we perform an intervention on either $X_i$ or $X_j$. This leads us to:

$$
\begin{aligned}
\frac{\partial}{\partial \theta_{ij}} \tilde{\mathcal{L}} = \sigma'(\theta_{ij}) \Big( & p(I_{X_i}) \cdot \sigma(\gamma_{ij}) \cdot \mathbb{E}_{I_{X_i}, \boldsymbol{X}, C_{-ij}} \left[ \mathcal{L}_{X_i \to X_j}(X_j) - \mathcal{L}_{X_i \not\to X_j}(X_j) \right] - \\
& p(I_{X_j}) \cdot \sigma(\gamma_{ji}) \cdot \mathbb{E}_{I_{X_j}, \boldsymbol{X}, C_{-ij}} \left[ \mathcal{L}_{X_j \to X_i}(X_i) - \mathcal{L}_{X_j \not\to X_i}(X_i) \right] \Big)
\end{aligned} \tag{4}
$$

The probability of taking an intervention on $X_i$ is represented by $p(I_{X_i})$ (usually uniform across variables), and $\mathbb{E}_{I_{X_i}, \boldsymbol{X}, C_{-ij}}$ the same expectation as before under the intervention on $X_i$. When the oriented edge $X_i \to X_j$ improves the log-likelihood of $X_j$ under intervention on $X_i$, then the first part of the gradient increases $\theta_{ij}$. In contrast, when the true edge is $X_j \to X_i$, the correlation between $X_i$ and $X_j$ learned from observational data would yield a worse likelihood estimate of $X_j$ on interventional data on $X_i$ than without the edge $X_j \to X_i$. This is because $p(X_j|X_i, ...)$ does not stay invariant under intervening on $X_i$. The same dynamic holds for interventions on $X_j$. Lastly, for independent nodes, the expectation of the gradient is zero.

Based on Equations 3 and 4, we obtain a tractable, unbiased gradient estimator by using Monte-Carlo sampling. Luckily, samples can be shared across variables, making training efficient. We first sample an intervention, a corresponding data batch, and $K$ graphs from $p_{\boldsymbol{\gamma},\boldsymbol{\theta}}(C)$ ($K$ usually between 20 and 100). We then evaluate the log likelihoods of all variables for these graphs on the batch, and estimate $\mathcal{L}_{X_i \to X_j}(X_j)$ and $\mathcal{L}_{X_i \not\to X_j}(X_j)$ for all pairs of variables $X_i$ and $X_j$ by simply averaging the results for the two cases separately. Finally, the estimates are used to determine the gradients for $\boldsymbol{\gamma}$ and $\boldsymbol{\theta}$.

**Low variance.** Previous methods (Ke et al., 2019; Bengio et al., 2020) relied on a different REINFORCE-like estimator proposed by Bengio et al. (2020). Adjusting their estimator to our setting of the parameter $\gamma_{ij}$, for instance, the gradient looks as follows:

$$
g_{ij} = \sigma(\theta_{ij}) \cdot \mathbb{E}_{\boldsymbol{X}} \left[ \frac{\mathbb{E}_C \left[ (\sigma(\gamma_{ij}) - C_{ij}) \cdot \mathcal{L}_C(X_j) \right]}{\mathbb{E}_C \left[ \mathcal{L}_C(X_j) \right]} + \lambda_{\text{sparse}} \right] \tag{5}
$$

where $g_{ij}$ represents the gradient of $\gamma_{ij}$. Performing Monte-Carlo sampling for estimating the gradient leads to a biased estimate which becomes asymptotically unbiased with increasing number of samples (Bengio et al., 2020). The division by the expectation of $\mathcal{L}_C(X_j)$ is done for variance reduction (Mahmood et al., 2014). Equation 5, however, is still sensitive to the proportion of sampled $C_{ij}$ being one or zero. A major benefit of our gradient formulation in Equation 3, instead, is that it removes this noise by considering the difference of the two independent Monte-Carlo estimates $\mathcal{L}_{X_i \to X_j}(X_j)$ and $\mathcal{L}_{X_i \not\to X_j}(X_j)$. Hence, we can use a smaller sample size than previous methods and attain 10 times lower standard deviation, as visualized in Figure 2.

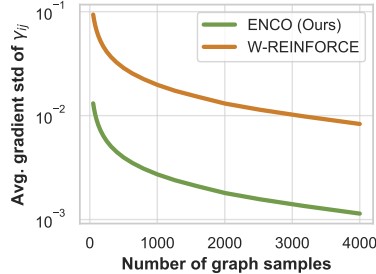

Figure 2: ENCO estimates gradients of significantly lower variance compared to (Bengio et al., 2020).

### 3.4 CONVERGENCE GUARANTEES

Next, we discuss the conditions under which ENCO convergences to the correct causal graph. We show that not only does the global optimum of Equation 2 correspond to the true graph, but also that there exist no other local minima ENCO can converge to. We outline the derivation and proof of these conditions in Appendix B, and limit our discussion here to the main assumptions and implications.

To construct a theoretical argument, we make the following assumptions. First, we assume that sparse interventions have been performed on all variables. Later, we show how to extend the algorithm to avoid this strong assumption. Further, given a CGM, we assume that its joint distribution $p(\boldsymbol{X})$ is Markovian with respect to the true graph $\mathcal{G}$. In other words, the parent set $\text{pa}(X_i)$ reflects the inputs to the causal generation mechanism of $X_i$. We assume that there exist no latent confounders in $\mathcal{G}$. Also, we assume the neural networks in ENCO are sufficiently large and sufficient observational data is provided to model the conditional distributions of the CGM up to an arbitrary small error.

Under these assumptions, we produce the following conditions for convergence:

**Theorem 3.1.** *Given a causal graph $\mathcal{G}$ with variables $X_1, ..., X_N$ and conditional observational distributions $p(X_i|...)$, the proposed method ENCO will converge to the true, causal graph $\mathcal{G}$, if the following conditions hold for all edges $X_i \rightarrow X_j$ in $\mathcal{G}$:*

1. *For all possible sets of parents of $X_j$ excluding $X_i$, by adding $X_i$ the log-likelihood estimate of $X_j$ is improved or unchanged under the intervention on $X_i$:*

$$\forall \widehat{pa}(X_j) \subseteq X_{-i,j} : \mathbb{E}_{I_{X_i}, \mathbf{X}} \left[ \log p(X_j|\widehat{pa}(X_j), X_i) - \log p(X_j|\widehat{pa}(X_j)) \right] \geq 0 \quad (6)$$

2. *For at least one set of nodes $\widehat{pa}(X_j)$, for which the probability to be sampled as parents of $X_j$ is greater than 0, Equation 6 must be strictly greater than zero.*

3. *The effect of $X_i$ on $X_j$ cannot be described by other variables up to $\lambda_{sparse}$:*

$$\min_{\hat{pa} \subseteq gpa_i(X_j)} \mathbb{E}_{\hat{I} \sim p_{I_{-j}}(I)} \mathbb{E}_{\tilde{p}_{\hat{I}}(\mathbf{X})} \left[ \log p(X_j|\hat{pa}, X_i) - \log p(X_j|\hat{pa}) \right] > \lambda_{sparse} \quad (7)$$

*where $gpa_i(X_j)$ is the set of nodes excluding $X_i$ which, according to the ground truth graph, could have an edge to $X_j$ without introducing a cycle, and $p_{I_{-j}}(I)$ refers to the distribution over interventions $p_I(I)$ excluding the intervention on variable $X_j$.*

*Further, for all other pairs $X_i, X_j$ for which $X_j$ is a descendant of $X_i$, condition 1 and 2 must hold.*

Condition 1 and 2 ensure that the orientations can be learned from interventions. Intuitively, ancestors and descendants in the graph have to be dependent when intervening on the ancestors. This aligns with the technical interpretation in Theorem 3.1 that the likelihood estimate of the child variable must improve when intervening and conditioning on its ancestor variables. Condition 3 states intuitively that the sparsity regularizer needs to be selected such that it chooses the sparsest graph among those graphs with equal joint distributions as the ground truth graph, without trading sparsity for worse distribution estimates. The specific condition in Theorem 3.1 ensures thereby that the set can be learned with a gradient-based algorithm. We emphasize that this condition only gives an upper bound for $\lambda_{\text{sparse}}$ when sufficiently large datasets are available. In practice, the graph can thus be recovered with a sufficiently small sparsity regularizer and dependencies among variables under interventions. We provide more details for various settings and further intuition in Appendix B.

**Interventions on fewer variables.** It is straightforward to extend ENCO to support interventions on fewer variables. Normally, in the graph fitting stage, we sample one intervention at a time. We can, thus, simply restrict the sampling only to the interventions that are possible (or provided in the dataset). In this case, we update the orientation parameters $\theta_{ij}$ of only those edges that connect to an intervened variable, either $X_i$ or $X_j$, as before. For all other orientation parameters, we extend the gradient estimator to include interventions on all variables. Although this estimate is more noisy and does not have convergence guarantees, it can still be informative about the edge orientations.

**Enforcing acyclicity** When the conditions are violated, e.g. by limited data, cycles can occur in the prediction. Since ENCO learns the orientations as a separate parameter, we can remove cycles by finding the global order of variables $O \in S_N$, with $S_N$ being the set of permutations, that maximizes the pairwise orientation probabilities: $\arg\max_O \prod_{i=1}^N \prod_{j=i+1}^N \sigma(\theta_{O_i, O_j})$. This utilizes the learned ancestor-descendant relations, making the algorithm more robust to noise in single interventions.

### 3.5 HANDLING LATENT CONFOUNDERS

So far, we have assumed that all variables of the graph are observable and can be intervened on. A common issue in causal discovery is the existence of latent confounders, *i.e.*, an unobserved common cause of two or more variables introducing dependencies between each other. In the presence of latent confounders, structure learning methods may predict false positive edges. Interestingly, in the context of ENCO latent confounders for two variables $X_i, X_j$ cause a unique pattern of learned parameters. When intervening on $X_i$ or $X_j$, having an edge between the two variables is disadvantageous, as in the intervened graph $X_i$ and $X_j$ are (conditionally) independent. For interventions on all other variables, however, an edge can be beneficial as $X_i$ and $X_j$ are correlated.

Exploiting this, we extend ENCO to detect latent confounders. We focus on latent confounders between two variables that do not have any direct edges with each other, and assume that the confounder is not a child of any other observed variable. For all other edges besides between $X_i$

Table 1: Comparing structure learning methods in terms of structural hamming distance (SHD) on common graph structures (lower is better), averaged over 25 graphs each. ENCO outperforms all baselines, and by enforcing acyclicity after training, can recover most graphs with minimal errors.

| Graph type | bidiag | chain | collider | full | jungle | random |
|---|---|---|---|---|---|---|
| GIES | 33.6 ($\pm$7.5) | 17.5 ($\pm$7.3) | 24.0 ($\pm$2.9) | 216.5 ($\pm$15.2) | 33.1 ($\pm$2.9) | 57.5 ($\pm$14.2) |
| IGSP | 32.7 ($\pm$5.1) | 14.6 ($\pm$2.3) | 23.7 ($\pm$2.3) | 253.8 ($\pm$12.6) | 35.9 ($\pm$5.2) | 65.4 ($\pm$8.0) |
| GES + Orientating | 14.8 ($\pm$2.6) | 0.5 ($\pm$0.7) | 20.8 ($\pm$2.4) | 282.8 ($\pm$4.2) | 14.7 ($\pm$3.1) | 60.1 ($\pm$8.9) |
| SDI | 9.0 ($\pm$2.6) | 3.9 ($\pm$2.0) | 16.1 ($\pm$2.4) | 153.9 ($\pm$10.3) | 6.9 ($\pm$2.3) | 10.8 ($\pm$3.9) |
| DCDI | 16.9 ($\pm$2.0) | 10.1 ($\pm$1.1) | 10.9 ($\pm$3.6) | 21.0 ($\pm$4.8) | 17.9 ($\pm$4.1) | 7.7 ($\pm$3.2) |
| ENCO (ours) | 2.2 ($\pm$1.4) | 1.7 ($\pm$1.3) | **1.6** ($\pm$1.6) | 9.2 ($\pm$3.4) | 1.7 ($\pm$1.3) | 4.6 ($\pm$1.9) |
| ENCO-acyclic (ours) | **0.0** ($\pm$0.0) | **0.0** ($\pm$0.0) | **1.6** ($\pm$1.6) | **5.3** ($\pm$2.3) | **0.6** ($\pm$1.1) | **0.2** ($\pm$0.5) |

and $X_j$, we can still rely on the guarantees in Section 3.4 since Equation 7 already includes the possibility of additional edges in such cases. After convergence, we score every pair of variables on how likely they share a latent confounder using a function $\text{lc}(\cdot)$ that is maximized in the scenario mentioned above. For this, we define $\gamma_{ij} = \gamma_{ij}^{(I)} + \gamma_{ij}^{(O)}$ where $\gamma_{ij}^{(I)}$ is only updated with gradients from Equation 3 under interventions on $X_i$, and $\gamma_{ij}^{(O)}$ on all others. With this separation, we define the following score function which is maximized by latent confounders:

$$\text{lc}(X_i, X_j) = \sigma\left(\gamma_{ij}^{(O)}\right) \cdot \sigma\left(\gamma_{ji}^{(O)}\right) \cdot \left(1 - \sigma\left(\gamma_{ij}^{(I)}\right)\right) \cdot \left(1 - \sigma\left(\gamma_{ji}^{(I)}\right)\right) \tag{8}$$

To converge to the mentioned values, especially of $\gamma_{ij}^{(O)}$, we need a similar condition as in Equation 7: the improvement on the log-likelihood estimate gained by the edge $X_i \to X_j$ and conditioned on all other parents of $X_j$ needs to be larger than $\lambda_{\text{sparse}}$ on interventional data excluding $X_i$ and $X_j$. If this is not the case, the sparsity regularizer will instead remove the edge between $X_i$ and $X_j$ preventing any false predictions among observed variables. For all other pairs of variables, at least one of the terms in Equation 8 converges to zero. Thus, we can detect latent confounders by checking whether the score function $\text{lc}(X_i, X_j)$ is greater than a threshold hyperparameter $\tau \in (0.0, 1.0)$. We discuss possible guarantees in Appendix B, and experimentally verify this approach in Section 4.5.

## 4 EXPERIMENTS

We evaluate ENCO on structure learning on synthetic datasets for systematic comparisons and real-world datasets for benchmarking against other methods in the literature. The experiments focus on graphs with categorical variables, and experiments on continuous data are included in Appendix D.5. Our code is publicly available at `https://github.com/phlippe/ENCO`.

### 4.1 EXPERIMENTAL SETUP

**Graphs and datasets.** Given a ground-truth causal graphical model, all methods are tasked to recover the original DAG from a set of observational and interventional data points for each variable. In case of synthetic graphs, we follow the setup of Ke et al. (2019) and create the conditional distributions from neural networks. These networks take as input the categorical values of its variable's parents, and are initialized orthogonally to output a non-trivial distribution.

**Baselines.** We compare ENCO to GIES (Hauser & Bühlmann, 2012) and IGSP (Wang et al., 2017; Yang et al., 2018) as greedy score-based approaches, and DCDI (Brouillard et al., 2020) and SDI (Ke et al., 2019) as continuous optimization methods. Further, as a common observational baseline, we apply GES (Chickering, 2002) on the observational data to obtain a graph skeleton, and orient each edge by learning the skeleton on the corresponding interventional distribution. We perform a separate hyperparameter search for all baselines, and use the same neural network setup for SDI, DCDI, and ENCO. Appendix C provides a detailed overview of the hyperparameters for all experiments.

### 4.2 CAUSAL STRUCTURE LEARNING ON COMMON GRAPH STRUCTURES

We first experiment on synthetic graphs. We pick six common graph structures and sample 5,000 observational data points and 200 per intervention. The graphs `chain` and `full` represent the

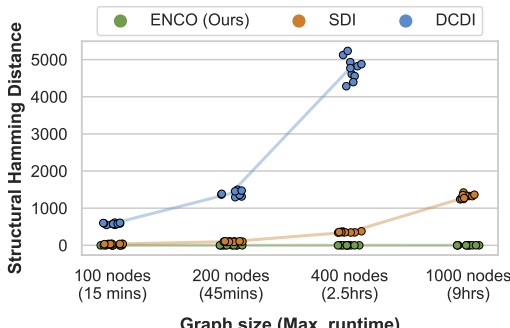

Figure 4: Experiments on graphs with interventions on fewer variables. Additional graphs are shown in Appendix D.2. ENCO outperforms DCDI on bidiag and jungle, even for very few interventions.

minimally- and maximally-connected DAGs. The graph `bidiag` is a chain with 2-hop connections, and `jungle` is a tree-like graph. In the `collider` graph, one node has all other nodes as parents. Finally, `random` has a randomly sampled graph structure with a likelihood of 0.3 of two nodes being connected by a direct edge. For each graph structure, we generate 25 graphs with 25 nodes each, on which we report the average performance and standard deviation. Following common practice, we use structural hamming distance (SHD) as evaluation metric. SHD counts the number of edges that need to be removed, added, or flipped in order to obtain the ground truth graph.

Table 1 shows that the continuous optimization methods outperform the greedy search approaches on categorical variables. SDI works reasonably well on sparse graphs, but struggles with nodes that have many parents. DCDI can recover the collider and full graph to a better degree, yet degrades for sparse graphs. ENCO performs well on all graph structures, outperforming all baselines. For sparse graphs, cycles can occur due to limited sample size. However, with enforcing acyclicity, ENCO-acyclic is able to recover four out of six graphs with less than one error on average. We further include experiments with various sample sizes in Appendix D.1. While other methods do not reliably recover the causal graph even for large sample sizes, ENCO attains low errors even with smaller sample sizes.

## 4.3 SCALABILITY

Next, we test ENCO on graphs with large sets of variables. We create `random` graphs ranging from $N = 100$ to $N = 1,000$ nodes with larger sample sizes. Every node has on average 8 edges and a maximum of 10 parents. The challenge of large graphs is that the number of possible edges grows quadratically and the number of DAGs super-exponentially, requiring efficient methods.

We compare ENCO to the two best performing baselines from Table 1, SDI and DCDI. All methods were given the same setup of neural networks and a maximum runtime which corresponds to 30 epochs for ENCO. We plot the SHD over graph size and runtime in Figure 3. ENCO recovers the causal graphs perfectly with no errors except for the 1,000-node graph, for which it misses only

Figure 3: Evaluating SDI, DCDI, and ENCO on large graphs in terms of SHD (lower is better). Dots represent single experiments, lines connect the averages. DCDI ran OOM for 1000 nodes.

one out of 1 million edges in 2 out of 10 experiments. SDI and DCDI achieve considerably worse performance. This shows that ENCO can efficiently be applied to 1,000 variables while maintaining its convergence guarantees, underlining the benefit of its low-variance gradient estimators.

## 4.4 INTERVENTIONS ON FEWER VARIABLES

We perform experiments on the same datasets as in Section 4.2, but provide interventional data only for a randomly sampled subset of the 25 variables of each graph. We compare ENCO to DCDI, which supports partial intervention sets, and plot the SHD over the number of intervened variables in Figure 4. Despite ENCO's guarantees only holding for full interventions, it is still competitive and outperforms DCDI in most settings. Importantly, enforcing acyclicity has an even greater impact on fewer interventions as more orientations are trained on non-adjacent interventions (see Appendix B.4 for detailed discussion). We conclude that ENCO works competitively with partial interventions too.

Table 3: Results on graphs from the BnLearn library measured in structural hamming distance (lower is better). Results are averaged over 5 seeds with standard deviations listed in Appendix C.5. Despite deterministic variables and rare events, ENCO can recover all graphs with almost no errors.

| Dataset | cancer (5 nodes) | asia (8 nodes) | sachs (11 nodes) | child (20 nodes) | alarm (37 nodes) | diabetes (413 nodes) | pigs (441 nodes) |
|---|---|---|---|---|---|---|---|
| SDI | 3.0 | 4.0 | 7.0 | 11.2 | 24.4 | 422.4 | 18.0 |
| DCDI | 4.0 | 5.0 | 5.4 | 8.4 | 30.0 | - | - |
| ENCO (Ours) | **0.0** | **0.0** | **0.0** | **0.0** | **1.0** | **2.0** | **0.0** |

## 4.5 DETECTING LATENT CONFOUNDERS

To test the detection of latent confounders, we create a set of 25 `random` graphs with 5 additional latent confounders. The dataset is generated in the same way as before, except that we remove the latent variable from the input data and increase the observational and interventional sample size (see Appendix C.3 for ablation studies). After training, we predict the existence of a latent confounder on any pair of variables $X_i$ and $X_j$ if $\text{lc}(X_i, X_j)$ is greater than $\tau$. We choose $\tau = 0.4$ but verify in Appendix C.3 that the method is not sensitive to the specific value of $\tau$. As shown in Table 2, ENCO detects more than $95\%$ of the latent confounders without any false positives. What is more, the few mistakes do not affect the detection of all other edges, which are recovered perfectly.

Table 2: Results of ENCO on detecting latent confounders. The missed confounders do not affect other edge predictions.

| Metrics | ENCO |
|---|---|
| SHD | $0.0\ (\pm 0.0)$ |
| Confounder recall | $96.8\%\ (\pm 9.5\%)$ |
| Confounder precision | $100.0\%\ (\pm 0.0\%)$ |

## 4.6 REAL-WORLD INSPIRED DATA

Finally, we evaluate ENCO on causal graphs from the Bayesian Network Repository (BnLearn) (Scutari, 2010). The repository contains graphs inspired by real-world applications that are used as benchmarks in literature. In comparison to the synthetic graphs, the real-world graphs are sparser with a maximum of 6 parents per node and contain nodes with strongly peaked marginal distributions. They also include deterministic variables, making the task challenging even for small graphs.

We evaluate ENCO, SDI, and DCDI on 7 graphs with increasing sizes, see Table 3. We observe that ENCO recovers almost all real-world causal graphs without errors, independent of their size. In contrast, SDI suffers from more mistakes as the graphs become larger. An exception is `pigs` (Scutari, 2010), which has a maximum of 2 parents per node, and hence is easier to learn. The most challenging graph is `diabetes` (Andreassen et al., 1991) due to its large size and many deterministic variables. ENCO makes only two mistakes, showing that it can handle deterministic variables well. We discuss results on small sample sizes in Appendix C.5, observing similar trends. We conclude that ENCO can reliably perform structure learning on a wide variety of settings, including deterministic variables.

## 5 CONCLUSION

We propose ENCO, an efficient causal structure learning method leveraging observational and interventional data. Compared to previous work, ENCO models the edge orientations as separate parameters and uses an objective unconstrained with respect to acyclicity. This allows for easier optimization and low-variance gradient estimators while having convergence guarantees. As a consequence, the algorithm can efficiently scale to graphs that are at least one order of magnitude larger graphs than what was possible. Experiments corroborate the capabilities of ENCO compared to the state-of-the-art on an extensive array of settings on graph sizes, sizes of observational and interventional data, latent confounding, as well as on both partial and full intervention sets.

**Limitations.** The convergence guarantees of ENCO require interventions on all variables, although experiments on fewer interventions have shown promising results. Future work includes investigating guarantee extensions of ENCO to this setting. A second limitation is that the orientations are missing transitivity: if $X_1 \succ X_2$ and $X_2 \succ X_3$, then $X_1 \succ X_3$ must hold. A potential direction is incorporating transitive relations for improving convergence speed and results on fewer interventions.

## ETHICS STATEMENT

Causal structure learning algorithms such as the proposed method are mainly used to uncover and understand causal mechanisms from data. The knowledge of the underlying causal mechanisms can then be applied to decide on specific actions that influence variables or factors in a desired way. For instance, by knowing that the environmental pollution in a city has an impact on the risk of cancer of its residents, one can try to reduce the pollution to decrease the risk of cancer. The applications of causal structure learning are ranging across many scientific disciplines, including computational biology (Friedman et al., 2000; Sachs et al., 2005; Opgen-Rhein & Strimmer, 2007), epidemiology (Robins et al., 2000; Vandenbroucke et al., 2016), and economics (Pearl, 2009; Hicks et al., 1980). We envision that our work can have positive impacts on those fields. One example we want to highlight is the field of genomics. Recent advances have enabled to perform gene knockdown experiments in a large scale, providing large amounts of interventional data (Dixit et al., 2016; Macosko et al., 2015). Gaining insights into how specific genes and diseases interact can lead to the development of novel pharmaceutic methods for treating current diseases. Since the number of variables in those experiments is tremendous, efficient causal structure learning algorithms are needed. The proposed method constitutes a first step towards this goal, and our work can foster future work for creating algorithms scaling beyond 10,000 variables.

Since the possible applications are fairly wide-ranging, there might be potential impacts we cannot forecast at the current time. This includes misuses of the method for unethical purposes. For instance, the method can be used to justify gender and race as causes for irrelevant variables if the output is misinterpreted, initial assumptions of the model are ignored, or the input data has been manipulated. Hence, the obligation to use this method in a correct way within ethical boundaries lies on the user. We emphasize this responsibility of the user in the license of our code.

## REPRODUCIBILITY STATEMENT

To ensure reproducibility, we have published the source code of the proposed method ENCO at `https://github.com/phlippe/ENCO`. The code includes instructions on how to download the datasets, and reproduce the experiments in Section 4 and additional experiments in Appendix D. Further, for all experiments of Section 4, we have included a detailed overview in Appendix C of (a) the used data and its generation process, (b) all hyperparameters used for all methods, and (c) additional details on the results. All experiments have been repeated with 5 to 25 seeds to obtain stable, reproducible results. Appendix C.1.2 outlines the packages that have been used for running the baselines.

The computation resources deployed for all experiments are a 24-core CPU with a single NVIDIA RTX3090 GPU. All experiments can be reproduced on a computer with a single GPU, and only the experiments on graphs larger than 100 variables require a GPU memory of about 24GB. The other experiments can be performed on smaller GPUs as well.

### ACKNOWLEDGEMENTS

This work is financially supported by Qualcomm Technologies Inc., the University of Amsterdam and the allowance Top consortia for Knowledge and Innovation (TKIs) from the Netherlands Ministry of Economic Affairs and Climate Policy. We thank Pascal Mettes, Christina Winkler, and Sara Magliacane for their valuable comments and feedback on an initial draft of this work. We thank the anonymous reviewers for the reviews, suggestions, and engaging discussions which helped to improve this work further. Finally, we thank SURFsara for the support in using the Lisa Compute Cluster.

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

## Supplementary material
## Efficient Neural Causal Discovery without Acyclicity Constraints

## Table of Contents

## A  Gradient estimators

The following section describes in detail the derivation of the gradient estimators discussed in Section 3.3. We consider the problem of causal structure learning where we parameterize the graph by edge existence parameters $\gamma$ and orientation parameters $\theta$. Our objective is to optimize $\gamma$ and $\theta$ such that we maximize the probability of interventional data, *i.e.*, data generated from the true

graphs under (arbitrary) interventions on single variables. Thereby, the likelihood estimates have been trained on observational data only. Additionally, we want to ensure that the graph is as sparse as possible to prevent unnecessary connections. Thus, an $\ell_1$ regularizer is added on top of the edge probabilities. The full objective can be written as follows:

$$\tilde{\mathcal{L}} = \mathbb{E}_{\hat{I} \sim p_I(I)} \mathbb{E}_{\tilde{p}_{\hat{I}}(\boldsymbol{X})} \mathbb{E}_{p_{\boldsymbol{\gamma},\boldsymbol{\theta}}(C)} \left[ \sum_{i=1}^{N} \mathcal{L}_C(X_i) \right] + \lambda_{\text{sparse}} \sum_{i=1}^{N} \sum_{j=1}^{N} \sigma(\gamma_{ij})\sigma(\theta_{ij}) \tag{9}$$

where:

- $N$ is the number of variables in the causal graph $(X_1, ..., X_N)$;
- $p_I(I)$ is the distribution over interventions that are performed. This distribution can be set as a hyperparameter to weight certain interventions higher than others. In our experiments, we assume it to be uniform across interventions on variables;
- $\tilde{p}_{\hat{I}}(\boldsymbol{X})$ is the joint distribution of all variables under the intervention $\hat{I}$;
- $p_{\boldsymbol{\gamma},\boldsymbol{\theta}}(C)$ is the distribution over adjacency matrices $C$, which we model as a product of independent edge probabilities: $p_{\boldsymbol{\gamma},\boldsymbol{\theta}}(C) = \prod_{i=1}^{N} \prod_{j=1,j\neq i}^{N} \sigma(\gamma_{ij}) \cdot \sigma(\theta_{ij})$;
- $\mathcal{L}_C(X_i)$ is the negative log-likelihood estimate of variable $X_i$ under sampled adjacency matrix $C$: $\mathcal{L}_C(X_i) = -\log f_{\phi_i}(X_i; C_{\cdot,i} \odot \boldsymbol{X}_{-i})$;
- $\lambda_{\text{sparse}}$ is a hyperparameter representing the regularization weight.

Based on this objective, we derive the gradient estimators for optimizing both edge existence and orientation parameters.

## A.1 LOW-VARIANCE GRADIENT ESTIMATOR FOR EDGE PARAMETERS

In order to optimize the edge parameters via SGD, we need to determine the gradient $\frac{\partial}{\partial \gamma_{ij}} \tilde{\mathcal{L}}$. Since $\mathcal{L}$ consists of a sum of two terms, *i.e.*, the log-likelihood estimate and the regularization, we can look at both parts separately. To prevent any confusion of index variables, we will use $k, l$ as indices for the parameter $\gamma_{kl}$ for which we determine the gradient, *i.e.*, $\frac{\partial}{\partial \gamma_{kl}} \tilde{\mathcal{L}}$, and $i, j$ as indices for sums.

As a first step, we determine the gradients for the regularization term. Those can be found by taking the derivative of the sigmoid:

$$\frac{\partial}{\partial \gamma_{kl}} \lambda_{\text{sparse}} \sum_{i=1}^{N} \sum_{j=1}^{N} \sigma(\gamma_{ij})\sigma(\theta_{ij}) = \sigma(\gamma_{kl}) \cdot (1 - \sigma(\gamma_{kl})) \cdot \sigma(\theta_{kl})\lambda_{\text{sparse}} \tag{10}$$

Thus, it is straight-forward to calculate for any edge parameter. In the following, we use $\sigma'(...)$ to abbreviate the derivate of the sigmoid: $\sigma'(\gamma_{kl}) = \sigma(\gamma_{kl})(1 - \sigma(\gamma_{kl}))$.

For the log-likelihood term, we start by reorganizing the expectations to simplify the gradient expression. The derivate term $\frac{\partial}{\partial \gamma_{kl}}$ can be moved inside the two expectations over interventional data since those are independent of the graph parameters. Thus, we can write:

$$\frac{\partial}{\partial \gamma_{kl}} \tilde{\mathcal{L}}' = \mathbb{E}_{\hat{I} \sim p_I(I)} \mathbb{E}_{\tilde{p}_{\hat{I}}(\boldsymbol{X})} \frac{\partial}{\partial \gamma_{kl}} \mathbb{E}_{p_{\boldsymbol{\gamma},\boldsymbol{\theta}}(C)} \left[ \sum_{i=1}^{N} \mathcal{L}_C(X_i) \right] \tag{11}$$

For readability, we denote $\tilde{\mathcal{L}}'$ to be the objective in Equation 9 without the regularizer.

Next, we take a closer look at the derivate of the expectation over adjacency matrices. Note that we have defined the adjacency matrix distribution as $p_{\boldsymbol{\gamma},\boldsymbol{\theta}}(C) = \prod_{i=1}^{N} \prod_{j=1,j\neq i}^{N} \sigma(\gamma_{ij})\sigma(\theta_{ij})$, with $C_{ij} = 1$ representing the edge $X_i \rightarrow X_j$. Since a parameter $\gamma_{ij}$ only influences the likelihood of the edge $X_i \rightarrow X_j$ and no other edges, we can reduce the expectation to a single binary variable over which we need to differentiate the expectation:

$$\frac{\partial}{\partial \gamma_{kl}} \mathbb{E}_{p_{\boldsymbol{\gamma},\boldsymbol{\theta}}(C)} \left[ \sum_{i=1}^{N} \mathcal{L}_C(X_i) \right] = \mathbb{E}_{p_{\boldsymbol{\gamma},\boldsymbol{\theta}}(C_{-kl})} \left[ \frac{\partial}{\partial \gamma_{kl}} \mathbb{E}_{p_{\boldsymbol{\gamma},\boldsymbol{\theta}}(C_{kl})} \left[ \sum_{i=1}^{N} \mathcal{L}_C(X_i) \right] \right] \tag{12}$$

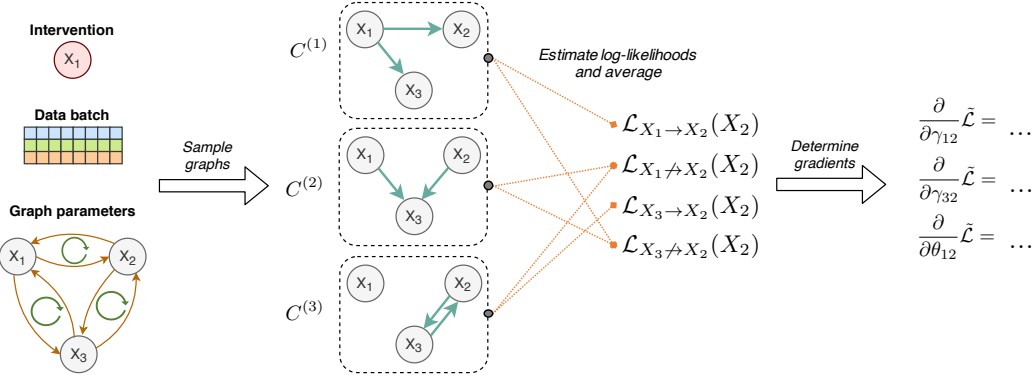

Figure 5: Visualizing the gradient calculation for the incoming edges of $X_2$ in an example graph with three variables. The intervention is being performed on $X_1$, and the data is used to calculate the log-likelihood estimates under the three randomly sampled graphs: $\mathcal{L}_{C^1}(X_2)$, $\mathcal{L}_{C^2}(X_2)$ and $\mathcal{L}_{C^3}(X_2)$. Those terms are assigned to the Monte-Carlo estimators for $\mathcal{L}_{X_i \to X_2}(X_2)$ and $\mathcal{L}_{X_i \not\to X_2}(X_2)$, and finally used to determine the gradients for $\boldsymbol{\gamma}$ and $\boldsymbol{\theta}$. The same process is performed for $X_3$ as well.

where $p_{\gamma,\theta}(C_{kl}) = \sigma(\gamma_{kl}) \cdot \sigma(\theta_{kl})$. The first expectation over $p_{\boldsymbol{\gamma},\boldsymbol{\theta}}(C_{-kl})$ is independent of $\gamma_{kl}$ as we have defined the adjacency matrix distribution to be a product of independent edge probabilities.

The log-likelihood estimate of a variable, $\mathcal{L}_C(X_i)$, depends on the adjacency matrix column $C_{\cdot,i}$ which represents the input connections to the node $X_i$. All other edges have no influence on the log-likelihood estimate of $X_i$. Hence, the parameter $\gamma_{kl}$ only influences $\mathcal{L}_C(X_l)$, and thus we can reduce the sum inside the expectation to:

$$\frac{\partial}{\partial \gamma_{kl}} \mathbb{E}_{p_{\gamma,\theta}(C_{kl})} \left[ \sum_{i=1}^{N} \mathcal{L}_C(X_i) \right] = \frac{\partial}{\partial \gamma_{kl}} \mathbb{E}_{p_{\gamma,\theta}(C_{kl})} \left[ \mathcal{L}_C(X_l) \right] \tag{13}$$

The REINFORCE trick is a simple method to move the derivative of a discrete distribution inside the expectation. Applied to our situation, we obtain:

$$\frac{\partial}{\partial \gamma_{kl}} \mathbb{E}_{p_{\gamma,\theta}(C_{kl})} \left[ \mathcal{L}_C(X_l) \right] = \mathbb{E}_{p_{\gamma,\theta}(C_{kl})} \left[ \mathcal{L}_C(X_l) \frac{\partial \log p_{\gamma,\theta}(C_{kl})}{\partial \gamma_{kl}} \right] \tag{14}$$

This leaves us with two cases in the expectation: $C_{kl} = 0$ and $C_{kl} = 1$. In other words, we need to distinguish between samples of $C$ where we have the edge $X_k \to X_l$, and where we do not have such an edge ($X_k \not\to X_l$). Thus, we can also write the expectation as a weighted sum of those two cases:

$$\mathbb{E}_{p_{\gamma,\theta}(C)} \left[ \mathcal{L}_C(X_l) \frac{\partial \log p_{\gamma,\theta}(C_{kl})}{\partial \gamma_{kl}} \right] = \sigma(\gamma_{kl}) \cdot \sigma(\theta_{kl}) \cdot \mathcal{L}_{X_k \to X_l}(X_l) \cdot \frac{\partial \log \sigma(\gamma_{kl}) \cdot \sigma(\theta_{kl})}{\partial \gamma_{kl}} +$$
$$(1 - \sigma(\gamma_{kl}) \cdot \sigma(\theta_{kl})) \cdot \mathcal{L}_{X_l \not\to X_k}(X_k) \cdot \frac{\partial \log (1 - \sigma(\gamma_{kl}) \cdot \sigma(\theta_{kl}))}{\partial \gamma_{kl}} \tag{15}$$

We use $\mathcal{L}_{X_k \to X_l}(X_l)$ to denote the (expected) negative log likelihood for $X_l$ under adjacency matrices where we have an edge from $X_k$ to $X_l$:

$$\mathcal{L}_{X_k \to X_l}(X_l) = \mathbb{E}_{p_{\gamma,\theta}(C_{-kl}), C_{kl}=1} \left[ \mathcal{L}_C(X_l) \right] \tag{16}$$
$$\mathcal{L}_{X_k \not\to X_l}(X_l) = \mathbb{E}_{p_{\gamma,\theta}(C_{-kl}), C_{kl}=0} \left[ \mathcal{L}_C(X_l) \right] \tag{17}$$

The final step is to solve the two derivative terms in Equation 14. This is done as follows:

$$\frac{\partial \log \sigma(\gamma_{kl}) \cdot \sigma(\theta_{kl})}{\partial \gamma_{kl}} = \frac{\partial \log \sigma(\gamma_{kl}) + \log \sigma(\theta_{kl})}{\partial \gamma_{kl}} = 1 - \sigma(\gamma_{kl}) \tag{18}$$
$$\frac{\partial \log (1 - \sigma(\gamma_{kl}) \cdot \sigma(\theta_{kl}))}{\partial \gamma_{kl}} = -\frac{\sigma(\gamma_{kl}) \cdot (1 - \sigma(\gamma_{kl})) \cdot \sigma(\theta_{kl})}{1 - \sigma(\gamma_{kl}) \cdot \sigma(\theta_{kl})} \tag{19}$$

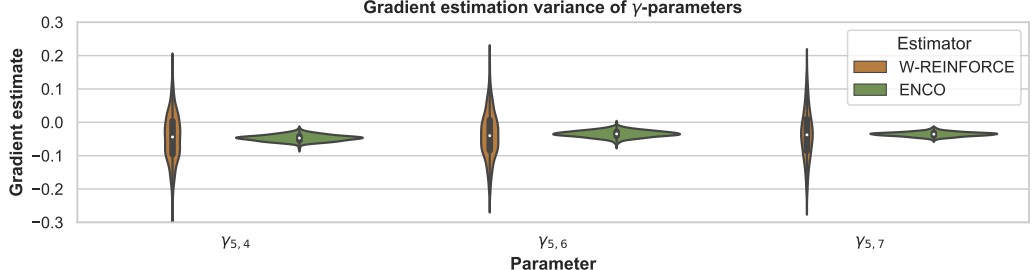

Figure 6: Plotting the gradient estimate variance of ENCO for three variables of $\gamma$ compared to previous REINFORCE-like approach by Bengio et al. (2020) on an example graph with $K = 100$. The gradients have been scaled to match in terms of averages. We can see a clear reduction in variance with the gradient estimator of ENCO allowing us to use lower sample sizes.

Putting these results back in the original equation and adding the sparsity regularizer, we get:

$$\frac{\partial}{\partial \gamma_{ij}} \tilde{\mathcal{L}} = \sigma(\gamma_{ij}) \cdot (1 - \sigma(\gamma_{ij})) \cdot \sigma(\theta_{ij}) \cdot \mathbb{E}_{\boldsymbol{X}, C_{-ij}} \left[ \mathcal{L}_{X_i \to X_j}(X_j) - \mathcal{L}_{X_i \not\to X_j}(X_j) + \lambda_{\text{sparse}} \right] \quad (20)$$

To align the result with the gradient in Section 3.3, we switch the index notation from $k, l$ to $i, j$ again. The expectation $\mathbb{E}_{\boldsymbol{X}, C_{-ij}}$ is a short form for the expectations $\mathbb{E}_{\hat{I} \sim p_I(I)} \mathbb{E}_{\tilde{p}_{\hat{I}}(\boldsymbol{X})} \mathbb{E}_{p_{\gamma, \theta}(C_{-ij})}$. From this expression, we can see that the gradients of $\gamma_{ij}$ are proportional to the difference of the expected negative log-likelihood of $X_j$ with having an edge between $X_i \to X_j$, and the cases where $X_i \not\to X_j$. The sparsity regularizer thereby biases the difference towards the no-edge case. The value of $\gamma_{ij}$ and $\theta_{ij}$ only scale the gradient, but do not influence its direction.

In order to train this objective on a dataset of interventional data, we can use Monte-Carlo sampling to obtain an unbiased gradient estimator. Note that the adjacency matrix samples to estimate $\mathcal{L}_{X_i \to X_j}(X_j)$ and $\mathcal{L}_{X_i \not\to X_j}(X_j)$ are not required to be the same. For efficiency, we instead sample $K$ adjacency matrices from $p_{\gamma, \theta}(C)$, evaluate the likelihood of a batch $\boldsymbol{X}$ under all these graphs. Afterwards, we assign the evaluated samples to one of the two cases, depending on $C_{ij}$ being zero or one. This way, we can reuse the same graph samples for all edge parameters $\gamma$. We visualize the gradient calculation in Figure 5. In the cases where we perform an intervention on $X_i$, we do not optimize $\gamma_{ij}$ for this step and set the gradients to zero. The same holds for gradient steps where we do not have any samples for one of the two log-likelihood estimates.

### A.1.1 COMPARISON TO PREVIOUS GRADIENT ESTIMATORS

As discussed in Section 3, previous work on similar structure learning methods (Bengio et al., 2020; Ke et al., 2019) relied on a different estimator. In terms of derivation, the main difference is the continuation from Equation 14 on. In our proposed method, we write the expectation as the sum of two terms that can independently be approximated via Monte-Carlo sampling. In comparison, Bengio et al. (2020) proposed to directly apply a Monte-Carlo sampler to Equation 14, and apply an importance sampling weight to reduce the variance. This estimator is also used in the method SDI (Ke et al., 2019) to which we have experimentally compared our method.

Figure 2 compared the gradient estimator in terms of standard deviation. The gradient estimator of ENCO achieves a 10 times lower standard deviation compared to Bengio et al. (2020) making it much more efficient. Since the estimator by Bengio et al. (2020) is biased and has a different mean, we have scaled both estimators to have the same mean. Specifically, we have applied ENCO to `random` graphs from our experiments on synthetic graphs (see Section 4.2), and evaluated $64,000$ sampled adjacency matrices in terms of log-likelihood estimates. These $64,000$ samples are grouped into sets of $K$ samples which we have used to estimate the gradients of $\gamma$. We evaluated different values of $K$, from $K = 20$ to $K = 4000$, and plotted the standard deviation of those estimates in Figure 2. We have also visualized three examples as violin plots in Figure 6 that demonstrate that despite both estimators having a similar mean, the variance of gradient estimates is much higher for Bengio et al. (2020).

To verify that the improvement of ENCO is not just because of the gradient estimators, we have performed an ablation study with ENCO deploying the gradient estimator of Bengio et al. (2020) in Appendix D.3.

## A.2 Low-variance gradient estimator for orientation parameters

To derive the gradients for the orientation parameters $\boldsymbol{\theta}$, we can mostly follow the same approach as for the edge existence parameters $\boldsymbol{\gamma}$. However, we have to keep in mind the constraint $\theta_{kl} = -\theta_{lk}$ which ensures that the orientation probability sums to one: $\sigma(\theta_{kl}) + \sigma(\theta_{lk}) = 1$.

To determine the gradient of the likelihood term, we can separate the two gradients of $\theta_{kl}$ and $\theta_{lk}$. This is because $\theta_{kl}$ only influences the expectation over $\mathcal{L}_C(X_l)$, while $\theta_{lk}$ concerns $\mathcal{L}_C(X_k)$. We can follow Equation 11 to Equation 20 of Section A.1 by swapping $\theta_{kl}$ and $\gamma_{kl}$. For the derivative through the expectation, we obtain the following gradient:

$$\frac{\partial}{\partial \theta_{kl}} \mathbb{E}_{p_{\boldsymbol{\gamma},\boldsymbol{\theta}}(C)} \left[ \mathcal{L}_C(X_l) \right] = \sigma'(\theta_{kl}) \cdot \sigma(\gamma_{kl}) \cdot \mathbb{E}_{p_{\boldsymbol{\gamma},\boldsymbol{\theta}}(C_{-kl})} \left[ \mathcal{L}_{X_k \to X_l}(X_l) - \mathcal{L}_{X_k \not\to X_l}(X_l) \right] \quad (21)$$

Since we have the condition that $\theta_{kl} = -\theta_{lk}$, the full gradient for $\theta_{kl}$ would therefore consist of the gradient above minus the gradient of Equation 21 with respect to $\theta_{ji}$. However, as discussed in Section 3.3, the orientation of an edge cannot be learned from observational data in this framework. Hence, we only want to use the gradients of $\theta_{kl}$ if we intervene on node $X_k$, which gives us the following gradient expression:

$$\begin{aligned}
\frac{\partial}{\partial \theta_{ij}} \tilde{\mathcal{L}} = \sigma'(\theta_{ij}) \Big( & p(I_{X_i}) \cdot \sigma(\gamma_{ij}) \cdot \mathbb{E}_{I_{X_i}, \boldsymbol{X}, C_{-ij}} \left[ \mathcal{L}_{X_i \to X_j}(X_j) - \mathcal{L}_{X_i \not\to X_j}(X_j) \right] - \\
& p(I_{X_j}) \cdot \sigma(\gamma_{ji}) \cdot \mathbb{E}_{I_{X_j}, \boldsymbol{X}, C_{-ij}} \left[ \mathcal{L}_{X_j \to X_i}(X_i) - \mathcal{L}_{X_j \not\to X_i}(X_i) \right] \Big)
\end{aligned} \quad (22)$$

To align the equation with the one in Section 3.3, we swap the indices $k, l$ with $i, j$ again. The first line represents cases where we have an intervention on the variable $X_i$, while we have it over interventions on the variable $X_j$ in the second line. The two terms are weighted based on the edge existence likelihood $\sigma(\gamma_{ij})$ and $\sigma(\gamma_{ji})$ respectively, and the likelihood of performing an intervention on $X_i$ or $X_j$. In our experiments, we use a uniform probability across interventions on variables, but emphasize that this is not strictly required. Moreover, one could design heuristics that selects the intervention to update the parameters on with the aim of increasing computational efficiency. The gradient estimators presented in Equation 22 would still be valid in such a case.

We clarify that we do not consider the gradients of $\theta_{ij}$ with respect to the edge regularizer. This is done for two reasons. Firstly, the orientation parameter models only the direction of the edge, not whether it exists or not. The regularizer would increase $\theta_{ij}$ if the edge existence for the opposite direction would be greater than for the direction from $X_i$ to $X_j$, *i.e.*$\gamma_{ij} < \gamma_{ji}$, and decrease $\theta_{ij}$ if we have $\gamma_{ij} > \gamma_{ji}$. However, the orientation should only model the causal direction of an edge. Hence, we do not gain any value from a causal perspective when adding the regularizer to the gradient. Secondly, the regularizer would require us to take additional assumptions for guaranteeing the discovery of the true graph upon convergence. In experiments with using a regularizer in the $\theta$-gradient, we did not observe any difference to the experiments without the regularizer.

We note that the orientation parameters are considered to be pairwise independent. In other words, $\theta_{ij}$ and $\theta_{kl}$ are considered independent parameters if $i \neq k, l$ and $j \neq k, l$. Global order distributions such as Plackett-Luce (Plackett, 1975; Luce, 1959) can be used to also incorporate transitive relations. However, those require high variance gradient estimators and struggled with chains in early experiments. The pairwise orientation parameters provide much easier optimization while still providing convergence guarantees for the full intervention setting.

## A.3 Training loop

Finally, we give an overview over the full training loop in Algorithm 1. The distribution over interventions $p(I)$ is set to a uniform distribution for all our experiments. However, the distribution

can also be replaced by a heuristic which selects interventions to increase computational efficiency. To keep the convergence guarantees, $p(I)$ would have to guarantee a non-zero probability to pick any variable. In experiments, we experienced that the Adam optimizer (Kingma & Ba, 2015) speeds up the convergence of the parameters $\boldsymbol{\gamma}$ and $\boldsymbol{\theta}$ while not interfering with the convergence guarantees in practice.

---

**Algorithm 1:** Learning algorithm of ENCO

---

**Data:** $N$ variables $\{X_1, ..., X_N\}$; observational dataset $D_{\text{obs}}$; interventional datasets $D_{\text{int}}(\hat{I})$ for
sparse, perfect interventions on all variables; distribution over interventions $p(I)$
**Result:** A graph structure corresponding to the cuasal relations among variables
Initialize $\boldsymbol{\gamma} = \mathbf{0}, \boldsymbol{\theta} = \mathbf{0}$
**for** *number of epochs* **do**
$\quad$ /* Distribution fitting $\qquad\qquad\qquad\qquad\qquad\qquad\qquad\qquad\qquad\qquad$ */
$\quad$ **for** *F iterations* **do**
$\qquad$ $\boldsymbol{X} \sim D_{\text{obs}}$
$\qquad$ **for** $i \leftarrow 1$ *to N* **do**
$\qquad\quad$ $M_l \sim \text{Ber}(\sigma(\gamma_{li}) \cdot \sigma(\theta_{li}))$
$\qquad\quad$ $L = -\log f_{\phi_i}(X_i; \boldsymbol{M}_{-i} \odot \boldsymbol{X}_{-i})$
$\qquad\quad$ $\phi_i \leftarrow \text{Adam}(\phi_i, \nabla_{\phi_i} L)$
$\qquad$ **end**
$\quad$ **end**
$\quad$ /* Graph fitting $\qquad\qquad\qquad\qquad\qquad\qquad\qquad\qquad\qquad\qquad\qquad$ */
$\quad$ **for** *G iterations* **do**
$\qquad$ /* Sample an intervention $\qquad\qquad\qquad\qquad\qquad\qquad\qquad\qquad$ */
$\qquad$ $\hat{I} \sim p(I)$ with intervention target $X_t$
$\qquad$ $\boldsymbol{X} \sim D_{\text{int}}(\hat{I})$
$\qquad$ /* Evaluate multiple graph samples for gradient estimator $\qquad$ */
$\qquad$ **for** $k = 1$ *to K* **do**
$\qquad\quad$ $C^{(k)} \sim \text{Ber}(\sigma(\boldsymbol{\gamma}) \cdot \sigma(\boldsymbol{\theta}))$
$\qquad\quad$ **for** $i = 1$ *to N* **do**
$\qquad\qquad$ $\mathcal{L}_{C^{(k)}}(X_i) \leftarrow -\log f_{\phi_i}(X_i; C_{:,i}^{(k)} \odot \boldsymbol{X}_{-i})$
$\qquad\quad$ **end**
$\qquad$ **end**
$\qquad$ /* Update parameters $\qquad\qquad\qquad\qquad\qquad\qquad\qquad\qquad\qquad$ */
$\qquad$ **for** $i = 1$ *to N* **do**
$\qquad\quad$ **for** $j = 1$ *to N where $j \neq i$ and $j \neq t$* **do**
$\qquad\qquad$ /* Considering edge $X_i \rightarrow X_j$ $\qquad\qquad\qquad\qquad\qquad$ */
$\qquad\qquad$ $\mathcal{L}_{X_i \rightarrow X_j}(X_j) \leftarrow \frac{\sum_{k=1}^{K} C_{ij}^{(k)} \cdot \mathcal{L}_{C^{(k)}}(X_j)}{\sum_{k=1}^{K} C_{ij}^{(k)}}$
$\qquad\qquad$ $\mathcal{L}_{X_i \nrightarrow X_j}(X_j) \leftarrow \frac{\sum_{k=1}^{K} (1 - C_{ij}^{(k)}) \cdot \mathcal{L}_{C^{(k)}}(X_j)}{\sum_{k=1}^{K} (1 - C_{ij}^{(k)})}$
$\qquad\qquad$ $\gamma_{ij} \leftarrow \gamma_{ij} - \sigma'(\gamma_{ij}) \cdot \sigma(\theta_{ij}) \cdot \left[ \mathcal{L}_{X_i \rightarrow X_j}(X_j) - \mathcal{L}_{X_i \nrightarrow X_j}(X_j) + \lambda_{\text{sparse}} \right]$
$\qquad\qquad$ **if** $i == t$ **then**
$\qquad\qquad\quad$ $\theta_{ij} \leftarrow \theta_{ij} - \sigma'(\theta_{ij}) \cdot \sigma(\gamma_{ij}) \cdot \left[ \mathcal{L}_{X_i \rightarrow X_j}(X_j) - \mathcal{L}_{X_i \nrightarrow X_j}(X_j) \right]$
$\qquad\qquad\quad$ $\theta_{ji} \leftarrow -\theta_{ij}$
$\qquad\qquad$ **end**
$\qquad\quad$ **end**
$\qquad$ **end**
$\quad$ **end**
**end**
**return** $G = (V, E)$ with $V = \boldsymbol{X}$ and
$E = \{(X_i, X_j) \mid i, j \in [1, N] \text{ and } i \neq j \text{ and } \sigma(\gamma_{ij}) > 0.5 \text{ and } \sigma(\theta_{ij}) > 0.5\}$

---

## B CONDITIONS FOR CONVERGING TO THE TRUE CAUSAL GRAPH

The following section gives an overview and proves the conditions under which ENCO converges to the correct causal graph given sufficient time and data. We emphasize that we provide conditions here for which no local optima exist, meaning that if ENCO converges, it returns the correct causal graph. This is a stronger statement than showing that the global optimum corresponds to the true graph, since a gradient-based algorithm can get stuck in a local optimum. We will discuss the conditions for the global optimum in Appendix B.2.5.

To make the proof more accessible, we will first discuss the assumptions that are needed for the guarantee, and then give a sketch of the proof. The proof will first assume that we work in the data limit, *i.e.* have given sufficient data, such that we can derive conditions that solely depend on the causal graphical model. In Appendix B.2.3, we extend the proof to the limited data setting.

### B.1 ASSUMPTIONS

**Assumption 1** We are given a dataset of observational data from the joint distribution $p(\boldsymbol{X})$. Additionally, we have $N$ interventional datasets for $N$ variables where in each intervention a different node is intervened on (the intervention size for each dataset is therefore 1).

**Assumption 2** A common assumption in causal structure learning is that the data distribution over all variables $p(\boldsymbol{X})$ is Markovian and faithful with respect to the causal graph we are trying to model. This means that the graph represents the (conditional) independence relations between variables in the data, and (conditional) independence relations in the data reflect the edges in the graph. For ENCO, faithfulness is not strictly required. This is because we work with interventional data. Instead, we rely on the Markov property and assume that for all variables, the parent set $\mathrm{pa}(X_i)$ reflects the inputs to the causal generation mechanism of $X_i$. This allows us to also handle deterministic variables.

**Assumption 3** For this proof, we assume that all variables of the graph are known and observable, and no latent confounders exist. Latent confounders can introduce dependencies between variables which are not reflected by the ground truth graph solely on the observed variables. We discuss the extension of latent confounders in Section 3.5 and Appendix B.3.

**Assumption 4** ENCO relies on neural networks to determine the conditional data distributions $p(X_i|...)$. Hence, for providing a guarantee, we assume that in the graph learning step the neural networks have been sufficiently trained such that they accurately model all possible conditional distribution $p(X_i|...)$. In practice, the neural networks might have a slight error. However, as long as enough data, network complexity, and training time is provided, it is fair to assume that the difference between the modeled distribution and the true conditional is smaller than an arbitrary constant $\epsilon$, based on the universal approximation theorem (Hornik et al., 1989). For the limited data setting, see Appendix B.2.3.

**Assumption 5** We are given a sufficiently large interventional dataset such that sampling data points from it models the exact interventional distribution under the true causal graph. This can be achieved by, for example, sampling directly from the causal graph, or having an infinitely large dataset. For the limited data setting, see Appendix B.2.3.

### B.2 CONVERGENCE CONDITIONS

The proof of the convergence conditions consists of the following three main steps:

**Step 1** We show under which conditions the orientation parameters $\theta_{ij}$ will converge to $+\infty$, i.e. $\sigma(\theta_{ij}) \to 1$, if $X_i$ is an ancestor of $X_j$. Similarly, if $X_i$ is a descendant of $X_j$, the parameter $\theta_{ij}$ will converge to $-\infty$, i.e. $\sigma(\theta_{ij}) \to 0$.

**Step 2** Under the assumption that the orientation parameters have converged as in Step 1, we show that for edges in the true graph, $\gamma_{ij}$ will converge to 1. Note that we need to take additional assumptions/conditions with respect to $\lambda_{\mathrm{sparse}}$ here.

**Step 3** Once the parameters $\gamma_{ij}$ and $\theta_{ij}$ have converged for the edges in the ground truth graph, we show that all other edges will be removed by the sparsity regularizer.

The following paragraphs provide more details for each step. Note that causal graphs that do not fulfill all parts of the convergence guarantee can still eventually be recovered. The reason is that the conditions listed in the theorems below ensure that there exists no local minima for $\theta$ and $\gamma$ to converge in. Although local minima exist, the optimization process might converge to the global minimum of the true causal graph.

**Theorem B.1.** *Consider the edge $X_i \to X_j$ in the true causal graph. The orientation parameter $\theta_{ij}$ converges to $\sigma(\theta_{ij}) = 1$ if the following two conditions are fulfilled:*

*(1) for all possible sets of parents of $X_j$ excluding $X_i$, adding $X_i$ improves the log-likelihood estimate of $X_j$ under the intervention on $X_i$, or leaves it unchanged:*

$$\forall \widehat{pa}(X_j) \subseteq X_{-i,j} : \mathbb{E}_{I_{X_i}, \mathbf{X}} \left[ \log p(X_j|\widehat{pa}(X_j), X_i) - \log p(X_j|\widehat{pa}(X_j)) \right] \geq 0 \qquad (23)$$

*(2) there exists a set of nodes $\widehat{pa}(X_j)$, for which the probability to be sampled as parents of $X_j$ is greater than 0, and the following condition holds:*

$$\exists \widehat{pa}(X_j) \subseteq X_{-i,j} : \mathbb{E}_{I_{X_i}, \mathbf{X}} \left[ \log p(X_j|\widehat{pa}(X_j), X_i) - \log p(X_j|\widehat{pa}(X_j)) \right] > 0 \qquad (24)$$

*Proof.* Based on the conditions in Equations 23 and 24, we need to show that the gradient of $\theta_{ij}$ is negative in expectation, independent of other values of $\gamma$ and $\theta$. For readability, we define the following function:

$$T(X_k, X_l) = \mathbb{E}_{I_{X_k}, \mathbf{X}, C_{-kl}} \left[ \mathcal{L}_{X_k \to X_l}(X_l) - \mathcal{L}_{X_k \nrightarrow X_l}(X_l) \right] \qquad (25)$$

Hence, the gradient of $\theta_{ij}$ can be written as:

$$\frac{\partial}{\partial \theta_{ij}} \tilde{\mathcal{L}} = \sigma'(\theta_{ij}) \cdot \left( p(I_{X_i}) \cdot \sigma(\gamma_{ij}) \cdot T(X_i, X_j) - p(I_{X_j}) \cdot \sigma(\gamma_{ji}) \cdot T(X_j, X_i) \right) \qquad (26)$$

Looking at the gradient of $\theta_{ij}$ in Equation 26, the conditions correspond to $T(X_i, X_j)$ being smaller or equals to zero. Note that the sign is flipped because in $T(X_i, X_j)$, we have negative log-likelihoods represented by $\mathcal{L}_{X_k \to X_l}(X_l)$, while in Equations 23 and 24, we have log-likelihoods. Further, the other factors of $\sigma'(\theta_{ij})$, $\sigma(\gamma_{ij})$ and $p(I_X)$ are all limited in the range of $(0, 1)$ meaning that the sign of the gradient is solely determined by $T(X_i, X_j)$ and $T(X_j, X_i)$. If $T(X_i, X_j) - T(X_j, X_i)$ is smaller than zero, then the gradient of $\theta_{ij}$ is negative, *i.e.* increasing $\theta_{ij}$.

First, we look at when $T(X_i, X_j) < 0$. The condition in Equation 23 ensures that conditioning $X_j$ on a true parent $X_i$ when intervening on $X_i$ does not lead to a worse log-likelihood estimate than without. While this condition might seem natural, there are special cases where this condition does not hold for all variables (see Section B.2.4). The second condition, Equation 24, guarantees that there is at least one parent set for which $T(X_i, X_j)$ is negative. Therefore, in expectation over all possible adjacency matrices $p_{\gamma, \theta}(C)$, $T(X_i, X_j)$ is smaller than zero if the two conditions hold.

To guarantee that the whole gradient of $\theta_{ij}$ is negative, we also need to show that for interventions on $X_j$, $T(X_j, X_i)$ can only be positive. When intervening on $X_j$, $X_i$ and $X_j$ become independent as the edge $X_i \to X_j$ is removed in the intervened graph. A distribution $p(X_i|X_j, ...)$ relying on correlations between $X_i$ and $X_j$ from observational data cannot achieve a better estimate than the same distribution when removing $X_j$. This is because the cross entropy is minimized when the sampled distribution, in this case $p(X_i)$, is equal to the log-likelihood estimator (Cover & Thomas, 2005):

$$-\sum_{x_i, x_j} p(X_i = x_i) \log p(X_i = x_i) \leq \sum_{x_i, x_j} p(X_i = x_i) \log q(X_i = x_i|X_j = x_j) \qquad (27)$$

The only situation where $X_i$ and $X_j$ can become conditionally dependent under interventions on $X_j$ is if $X_i$ and $X_j$ share a collider $X_k$, and $X_i$ is being conditioned on the collider $X_k$ and $X_j$. However, this requires that $\theta_{ki}$ has negative gradients, *i.e.* $\theta_{ki}$ increasing, when intervening on $X_k$. This cannot be the case since under interventions on $X_k$, $X_i$ and $X_k$ become conditionally independent, and the correlations learned from observational data cannot be transferred to the interventional setting. If $X_k$

and $X_i$ again share a collider, we can apply this argumentation recursively until a node $X_n$ does not share a collider with $X_i$. The recursion will always come to an end as we have a finite set of nodes, and the causal graph is assumed to be acyclic.

Thus, if the conditions in Equations 23 and 24 hold for an edge $X_i \rightarrow X_j$ in the causal graph, we can guarantee that with sufficient time and data, the corresponding orientation parameter $\theta_{ij}$ will converge to $\sigma(\theta_{ij}) = 1$. $\qquad\square$

**Theorem B.2.** *Consider a pair of variables $X_i, X_j$ for which $X_i$ is an ancestor of $X_j$ without direct edge in the true causal graph. Then, the orientation parameter of the edge $X_i \rightarrow X_j$ converges to $\sigma(\theta_{ij}) = 1$ if the same conditions as in Theorem B.1 hold for the pair of $X_i, X_j$.*

*Proof.* To show this theorem, we need to consider two cases for a pair of variables $X_i$ and $X_j$: $X_i$ and $X_j$ are conditionally independent under a sampled adjacency matrix $C$, or $X_i$ and $X_j$ are not independent. Both cases need to be considered for an intervention on $X_i$ with the log-likelihood estimate of $X_j$, and an intervention on $X_j$ with the log-likelihood estimate of $X_i$.

First, we discuss interventions on $X_i$. If under the sampled adjacency matrix $C$, $X_j$ is conditionally independent of $X_i$, the difference in the log-likelihood estimates $T(X_i, X_j)$ is zero in expectation. The variables can be independent if, for example, the parents of $X_j$ are all parents of the true causal graph. If $X_j$ is not conditionally independent of $X_i$, the conditions in Equations 23 and 24 from Theorem B.1 ensure that $X_i$ has, in expectation, a positive effect on the log-likelihood estimate of $X_j$. Thus, under interventions on $X_i$, the gradient of $\theta_{ij}$ is smaller or equals to zero, *i.e.* increases $\theta_{ij}$.

Next, we consider interventions on $X_j$. If under the sampled adjacency matrix $X_i$ is conditionally independent of $X_j$, the difference in the log-likelihood estimates $T(X_j, X_i)$ is zero. The variables can be independent if $X_i$ is conditioned on variables that d-separate $X_i$ and $X_j$ in the true causal graph. For instance, having the children of $X_i$ as parents of $X_i$ creates this scenario. However, for this scenario to take place, one or more orientation parameters of parent-child or ancestor-descendant pairs must be incorrectly converged. In case of a parent-child pair $X_i, X_k$, Theorem B.1 shows that $\sigma(\theta_{ik})$ will converge to one removing any possibility of a reversed edge to be sampled. In case of an ancestor-descendant pair $X_i, X_l$, we can apply a recursive argument: as $X_l$ d-separates $X_i$ and $X_j$, $X_l$ must come before $X_j$ in the causal order. If for the gradient $\theta_{il}$, we have a similar scenario with $X_i$ being conditionally independent of $X_j$, the same argument applies. This can be recursively applied until no more variables except direct children of $X_i$ can d-separate $X_i$ and $X_j$. In that case, $\sigma(\theta_{ik})$ will converge to one, which leads to all other orientation parameters to converge to one as well. If $X_i$ is not conditionally independent of $X_j$, we can rely back on the argumentation of Theorem B.1 when we have an edge $X_i \rightarrow X_j$: as in the intervened causal graph, $X_i$ and $X_j$ are independent, any correlation learned from observational data can only lead to a worse log-likelihood estimate. In cases of colliders, we can rely on the recursive argument from before. Thus, under interventions on $X_j$, the gradient of $\theta_{ij}$ must be smaller or equals to zero in expectation, *i.e.* increases $\theta_{ij}$.

Therefore, we can conclude that $\sigma(\theta_{ij})$ converges to one for any ancestor-descendant pairs $X_i, X_j$ under the conditions in Theorem B.1. $\qquad\square$

**Theorem B.3.** *Consider an edge $X_i \rightarrow X_j$ in the true causal graph. The parameter $\gamma_{ij}$ converges to $\sigma(\gamma_{ij}) = 1$ if the following condition holds:*

$$\min_{\hat{pa} \subseteq gpa_i(X_j)} \mathbb{E}_{\hat{I} \sim p_{I_{-j}}(I)} \mathbb{E}_{\tilde{p}_{\hat{I}}(\boldsymbol{X})} \left[ \log p(X_j | \hat{pa}, X_i) - \log p(X_j | \hat{pa}) \right] > \lambda_{sparse} \qquad (28)$$

*where $gpa_i(X_j)$ is the set of nodes excluding $X_i$ which, according to the ground truth graph, could have an edge to $X_j$ without introducing a cycle, and $p_{I_{-j}}(I)$ refers to the distribution over interventions $p_I(I)$ excluding the intervention on variable $X_j$.*

*Proof.* To show this convergence, we assume that the orientation parameters have converged corresponding to Theorem B.1 and B.2. The parameter $\gamma_{ij}$ converges to $\sigma(\gamma_{ij}) = 1$ if its gradient, $\frac{\partial}{\partial \gamma_{ij}} \tilde{\mathcal{L}}$, is negative independent of other values of $\boldsymbol{\gamma}$ and orientation parameters $\boldsymbol{\theta}$ that are not included in Theorem B.1 and B.2. The gradient of $\gamma_{ij}$ includes an expectation over adjacency matrices $p_{\boldsymbol{\gamma}, \boldsymbol{\theta}}(C)$. Based on the converged $\boldsymbol{\theta}$-values, we only need to consider sets of nodes as parents for $X_j$ that contain parents, ancestors, or (conditionally) independent nodes according to the ground truth graph. This sets of parents is represented by $gpa_i(X_j)$. Among those remaining parent sets,

we need to ensure that for any such set, the gradient is negative. The condition in Equation 28 corresponds to the inequality $\frac{\partial}{\partial \gamma_{ij}} \tilde{\mathcal{L}} < 0$ since the term on the left represents the log-likelihood difference $\mathcal{L}_{X_i \to X_j}(X_j) - \mathcal{L}_{X_i \not\to X_j}(X_j)$ in the gradients of $\gamma_{ij}$ in Equation 20 with a flipped sign. For readability and better interpretation, $\lambda_{\text{sparse}}$ has been moved on the right site of the inequality. This is possible as $\lambda_{\text{sparse}}$ is independent of the two expectations in Equation 28. If the inequality holds for all parent sets $\hat{\text{pa}}$), the gradient of $\gamma_{ij}$ can be guaranteed to be negative in expectation, independent of the other values of $\boldsymbol{\gamma}$. Since the distribution over parent sets $\hat{\text{pa}}$) depends on other values of $\boldsymbol{\gamma}$, the condition in Equation 28 ensures that even for the parent set with the lowest log-likelihood difference, it is still larger than $\lambda_{\text{sparse}}$. If this condition holds, then the gradient of $\frac{\partial}{\partial \gamma_{ij}} \tilde{\mathcal{L}}$ will be smaller than zero independent of other values of $\boldsymbol{\gamma}$. $\qquad \square$

The condition in Equation 28 introduces a dependency between convergence guarantees and the regularizer parameter $\lambda_{\text{sparse}}$. The lower we set the regularization weight $\lambda_{\text{sparse}}$, the more edges we can guarantee to recover. If the regularization weight is set too high, we can eventually obtain false negative edge predictions. If the regularization weight is set very low, we take a longer time to converge as it requires lower gradient variance or more update steps, and is more sensitive in a limited data regime. Nonetheless, if sufficient computational resources and data is provided, any value of $\lambda_{\text{sparse}} > 0$ can be used.

**Theorem B.4.** *Assume for all edges $X_i \to X_j$ in the true causal graph, $\sigma(\theta_{ij})$ and $\sigma(\gamma_{ij})$ have converged to one. Then, the likelihood of all other edges, i.e.$\sigma(\theta_{lk}) \cdot \sigma(\theta_{lk})$, will converge to zero under the condition that $\lambda_{sparse} > 0$.*

*Proof.* If all edges in the ground truth graph have converged, all other pairs of variables $X_l, X_k$ are (conditionally) independent in the graph. This statement follows from the Markov property of the graph and excludes ancestor-descendant pairs $X_i, X_j$. The possibility of having edges from descendants to ancestors has been removed by the fact that the orientation parameters $\theta_{ij}$ have converged according to Theorem B.2. Thus, for those cases, we already have the guarantee that $\sigma(\theta_{ij}) \cdot \sigma(\theta_{ij})$ converges to zero.

For a conditionally independent pair $X_l, X_k$, the difference of the log-likelihood estimate in the gradient of $\gamma_{lk}$, $i.e.\mathcal{L}_{X_l \to X_k}(X_k) - \mathcal{L}_{X_l \not\to X_k}(X_k)$, is zero in expectation since independent nodes do not share any information. Thus, the gradient remaining is:

$$\frac{\partial}{\partial \gamma_{lk}} \tilde{\mathcal{L}} = \sigma'(\gamma_{lk}) \cdot \sigma(\theta_{lk}) \cdot \lambda_{\text{sparse}} \qquad (29)$$

Since the gradient is positive independent of the values of $\gamma_{lk}$ and $\theta_{lk}$, $\gamma_{lk}$ will decrease until it converges to $\sigma(\gamma_{lk}) = 0$.

Hence, if $\gamma_{lk}$ decreases for all pairs of (conditionally) independent variables $X_l, X_k$ in the ground truth graph, and $\sigma(\theta_{lk})$ converged to zero for children and descendants, the product $\sigma(\gamma_{lk}) \cdot \sigma(\theta_{lk})$ will converge to zero for all edges not existing in the ground truth graph. $\qquad \square$

For graphs that fulfill all conditions in the Theorems B.1 and B.4, ENCO is guaranteed to converge given sufficient data and time. The conditions in the theorems ensure that there exist no local minima or saddle points in the loss surface of the objective in Equation 2 with respect to $\boldsymbol{\gamma}$ and $\boldsymbol{\theta}$.

**Summary** We can summarize the conditions discussed above as follows. Given a causal graph $\mathcal{G}$ with variables $X_1, ..., X_N$ and sparse interventions on all variables, the proposed method ENCO will converge to the true, causal graph $\mathcal{G}$, if the following three conditions hold for all edges $X_i \to X_j$ in the true causal graph $\mathcal{G}$:

1. For all possible sets of parents of $X_j$ excluding $X_i$, adding $X_i$ improves the log-likelihood estimate of $X_j$ under the intervention on $X_i$, or leaves it unchanged:

$$\forall \hat{\text{pa}}(X_j) \subseteq X_{-i,j} : \mathbb{E}_{I_{X_i}, \mathbf{x}} \left[ \log p(X_j | \hat{\text{pa}}(X_j), X_i) - \log p(X_j | \hat{\text{pa}}(X_j)) \right] \geq 0 \qquad (30)$$

2. There exists a set of nodes $\hat{\text{pa}}(X_j)$, for which the probability to be sampled as parents of $X_j$ is greater than 0, and the following condition holds:

$$\exists \hat{\text{pa}}(X_j) \subseteq X_{-i,j} : \mathbb{E}_{I_{X_i}, \mathbf{x}} \left[ \log p(X_j | \hat{\text{pa}}(X_j), X_i) - \log p(X_j | \hat{\text{pa}}(X_j)) \right] > 0 \qquad (31)$$

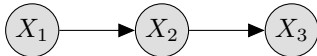

Figure 7: Example graph for which we check the convergence conditions described in Appendix B.2. The conditional distributions are given in Equation 33 to 35.

3. The effect of $X_i$ on $X_j$ cannot be described by other variables up to $\lambda_{\text{sparse}}$:

$$\min_{\hat{\text{pa}} \subseteq \text{gpa}_i(X_j)} \mathbb{E}_{\hat{I} \sim p_{I_{-j}}(I)} \mathbb{E}_{\tilde{p}_{\hat{I}}(\mathbf{x})} \big[ \log p(X_j|\hat{\text{pa}}, X_i) - \log p(X_j|\hat{\text{pa}}) \big] > \lambda_{\text{sparse}} \tag{32}$$

where $\text{gpa}_i(X_j)$ is the set of nodes excluding $X_i$ which, according to the ground truth graph, could have an edge to $X_j$ without introducing a cycle.

Further, for all other pairs $X_i$, $X_j$ for which $X_j$ is a descendant of $X_i$, conditions (1) and (2) need to hold as well.

### B.2.1 EXAMPLE FOR CHECKING CONVERGENCE CONDITIONS

In the following, we will provide a walkthrough for how the conditions above can be checked on a simple example graph. For further details on the precise calculations, we provide a Jupyter Notebook that contains all calculations in this example[1].

Suppose we have a graph with 3 binary variables, $X_1$, $X_2$, $X_3$, with the causal graph being $X_1 \to X_2 \to X_3$, i.e., a small chain. For simplicity, let us assume that the true, conditional distributions are the following:

$$p(X_1) = \text{Bern}(0.7) \tag{33}$$

$$p(X_2|X_1) = \begin{cases} X_1 & \text{with prob. } 0.6 \\ X_1 \oplus 1 & \text{with prob. } 0.4 \end{cases} \tag{34}$$

$$p(X_3|X_2) = \begin{cases} X_2 & \text{with prob. } 0.2 \\ X_2 \oplus 1 & \text{with prob. } 0.8 \end{cases} \tag{35}$$

In other words, $X_2$ is equals to the value of $X_1$ with a probability of $0.6$, and the opposite binary value otherwise. Similarly, $X_3$ is equals to the value of $X_2$ with a probability of $0.2$, and the opposite binary value with a probability of $0.8$. Therefore, the sample with the highest probability in this joint distribution would be $X_1 = 1, X_2 = 1, X_3 = 0$. Further, we assume that all interventions replace the respective conditional distribution by a uniform distribution, i.e., $p_{I_{X_i}}(X_i) = \text{Bern}(0.5)$. Next, we will check the conditions for the edges in $\mathcal{G}$, i.e., $X_1 \to X_2$ and $X_2 \to X_3$, and the remaining ancestor-descendant pair $X_1, X_3$.

**Edge $X_1 \to X_2$:**

- Condition 1: the possible parent sets that exclude $X_1$ and $X_2$ are $X_3$ and the empty set. For the empty set, we get:

$$\mathbb{E}_{I_{X_1}, \mathbf{x}} \left[ \log p(X_2|X_1) - \log p(X_2) \right] \approx 0.023 \geq 0$$

For conditioning on $X_3$, we obtain:

$$\mathbb{E}_{I_{X_1}, \mathbf{x}} \left[ \log p(X_2|X_1, X_3) - \log p(X_2|X_3) \right] \approx 0.015 \geq 0$$

Since both values are greater than zero, condition 1 is fulfilled for $X_1 \to X_2$.

- Condition 2: is already fulfilled by the equations in condition 1 since all parent sets have a difference greater than zero.

---

[1]The calculations can be found in the notebook called `convergence_guarantees_ENCO.ipynb`, see `https://github.com/phlippe/ENCO/blob/main/convergence_guarantees_ENCO.ipynb`.

- Condition 3: the set $\text{gpa}_1(X_2)$ is the empty set since $X_3$ is a descendant of $X_2$, and no other nodes exist in the graph. Thus, the parent set minimizing the expression on the left can only be the empty set, and we can calculate it as follows:

$$\mathbb{E}_{\hat{I} \sim p_{I_{-2}}(I)} \mathbb{E}_{\tilde{p}_{\hat{I}}(\mathbf{X})} \big[ \log p(X_2|X_1) - \log p(X_2) \big] \approx \underbrace{1/2 \cdot 0.023}_{I_{X_1}} + \underbrace{1/2 \cdot 0.017}_{I_{X_3}} = 0.020 > \lambda_{\text{sparse}}$$

with assuming $p_I(I)$ being the uniform distribution, and excluding $I_{X_2}$ since we do not update $\gamma_{12}$ in this case. Hence, as long as $\lambda_{\text{sparse}}$ is smaller than 0.02, the condition is fulfilled.

**Edge $X_2 \to X_3$:**

- Condition 1: the possible parent sets that exclude $X_2$ and $X_3$ are $X_1$ and the empty set. For the empty set, we get:

$$\mathbb{E}_{I_{X_2}, \mathbf{X}} \big[ \log p(X_3|X_2) - \log p(X_3) \big] \approx 0.194 \geq 0$$

For conditioning on $X_1$, we obtain:

$$\mathbb{E}_{I_{X_2}, \mathbf{X}} \big[ \log p(X_3|X_2, X_1) - \log p(X_3|X_1) \big] \approx 0.200 \geq 0$$

Since both values are greater than zero, condition 1 is fulfilled for $X_2 \to X_3$.

- Condition 2: is already fulfilled by the equations in condition 1 since all parent sets have a difference greater than zero.

- Condition 3: the set $\text{gpa}_2(X_3)$ contains the variable $X_1$ since we can introduce an edge $X_1 \to X_3$ without introducing acyclicity in the true, causal graph. Thus, we need to compare two parent sets for finding the minimum of the left-side term: $X_1$ and the empty set. First, we consider the empty set:

$$\mathbb{E}_{\hat{I} \sim p_{I_{-3}}(I)} \mathbb{E}_{\tilde{p}_{\hat{I}}(\mathbf{X})} \big[ \log p(X_3|X_2) - \log p(X_3) \big] \approx \underbrace{1/2 \cdot 0.194}_{I_{X_1}} + \underbrace{1/2 \cdot 0.194}_{I_{X_2}} = 0.194 > \lambda_{\text{sparse}}$$

Again, we exclude $I_{X_3}$ since we do not update $\gamma_{23}$ in this case. The second case considers $X_1$ as additional parent set $\hat{\text{pa}}$:

$$\mathbb{E}_{\hat{I} \sim p_{I_{-3}}(I)} \mathbb{E}_{\tilde{p}_{\hat{I}}(\mathbf{X})} \big[ \log p(X_3|X_2, X_1) - \log p(X_3|X_1) \big] \approx \underbrace{1/2 \cdot 0.186}_{I_{X_1}} + \underbrace{1/2 \cdot 0.200}_{I_{X_2}}$$
$$= 0.193 > \lambda_{\text{sparse}}$$

The minimum of both values is 0.193. Hence, the edge $X_2 \to X_3$ can be recovered if $0.193 > \lambda_{\text{sparse}}$.

**Ancestor-descendant pair $X_1, X_3$:**

- Condition 1: the possible parent sets that exclude $X_1$ and $X_3$ are $X_2$ and the empty set. For the empty set, we get:

$$\mathbb{E}_{I_{X_1}, \mathbf{X}} \big[ \log p(X_3|X_1) - \log p(X_3) \big] \approx 0.008 \geq 0$$

For conditioning on $X_2$, we obtain:

$$\mathbb{E}_{I_{X_1}, \mathbf{X}} \big[ \log p(X_3|X_1, X_2) - \log p(X_3|X_2) \big] = 0 \geq 0$$

The difference is zero because $X_3$ is independent of $X_1$ when conditioned on $X_2$: $p(X_3|X_1, X_2) = p(X_3|X_2)$ Since both values are greater or equals to zero, condition 1 is fulfilled for the pair $X_1, X_3$.

- Condition 2: from condition 1, we can see that the parent set of $\hat{\text{pa}}(X_3)$ being the empty set is the only option that fulfills the condition being greater than zero. Since we start the optimization process with an initialization that assigns a non-zero probability to all possible parent sets, it follows that $\hat{\text{pa}}(X_3)$ being the empty set has a probability greater than zero throughout the optimization process. Hence, condition 2 is fulfilled as well.

**Summary**: in conclusion, for the discussed example, we can guarantee that ENCO converges to the correct causal graph if $\lambda_{\text{sparse}} < 0.02$. To experimentally verify this results, we applied ENCO on this graph with two hyperparameter settings for the sparsity regularizer: $\lambda_{\text{sparse}} = 0.019$ and $\lambda_{\text{sparse}} = 0.021$. We considered a very large sample size, more specifically 10k per intervention and 100k observational samples, to simulate the data limit regime. For $\lambda_{\text{sparse}} = 0.019$, ENCO was able to recover the graph without errors while for $\lambda_{\text{sparse}} = 0.021$, the edge $X_1 \to X_2$ was, as expected, missed. This verifies the theoretical result above with respect to $\lambda_{\text{sparse}}$. Note that if the condition is not fulfilled by selecting a too large sparsity regularizer, this does not necessarily mean that ENCO will not be able to recover the graph. This is because we consider the 'worst-case' parent set in condition 3, while this case might not be in the true causal graph to which the other edges converge.

### B.2.2 Intuition behind Condition 1 and 2

As mentioned in the text, condition 1 and 2 of Theorem 3.1 ensure that the orientation probabilities cannot converge to any local optima. Since the conditions explicitly involve the data distributions and implicitly the gradient estimators, we provide below an assumption from a data generation mechanism perspective as an alternative, that ensures condition 1 and 2 to be satisfied.

Firstly, we assume that ancestors and descendants are not independent under interventions on the ancestors. Note that there can exist graphs where the ancestors are independent of descendants, for instance in a linear Gaussian setting when the ancestor has a weight of zero on the descendant. However, those graphs, violating faithfulness, are impossible to find for any causal discovery method since the variables are independent under any setting. In terms of condition 1 and 2, it would imply that the inequality is always zero.

Next, we show that under the previous assumption, local optima of the orientation probabilities can only occur in the following structure: for an edge $X_i \to X_j$, there exist one or more parent(s) of $X_j$ sharing a common confounder $X_k$ with $X_i$, where $X_k$ is not a direct parent of $X_j$. An example of this structure is the following: $X_1 \to X_2, X_3$; $X_2, X_3 \to X_4$ where the orientations of the edges $X_2 \to X_4$ and $X_3 \to X_4$ could have a local optimum. This statement can be proven as follows by using the do-calculus (Pearl, 2009). Suppose a graph that includes the three variables $X_1, X_2, X_3$ with $X_1 \to X_2$, $X_3 \to X_2$, and $X_2$ having no parents besides $X_1$ and $X_3$. If $X_1$ and $X_3$ do not share a confounder, then, from do-calculus, we know that $p(X_2|\text{do}(X_1 = x_1)) = p(X_2|X_1 = x_1)$ and $p(X_2|\text{do}(X_3 = x_3)) = p(X_2|X_3 = x_3)$. Furthermore, since the conditional entropy of a variable can only be smaller or equals to the marginal, i.e. $H(X) \geq H(X|Y)$, estimating $X_2$ under interventions on $X_1$ can only be improved by conditioning on $X_1$, and similarly for $X_3$. Thus, condition 1 is strictly fulfilled when parents do not share a confounder under the previous assumption of no independence in all possible settings. Now, consider the situation where $X_1$ and $X_3$ share a common confounder. Then, from do-calculus, we can state that there can exist a parameterization of the conditional distributions for which $p(X_2|\text{do}(X_1 = x_1)) \neq p(X_2|X_1 = x_1)$. Under this setting, we cannot guarantee that condition 1 is always fulfilled. However, whether this parent-confounder structure above actually leads to a local optimum or not depends on the distributions, which condition 1 models. Intuitively, this requires the mutual information between the two or more parents to be very high, and the initial edge probabilities of those edges to be very low. Further, as the results show, this combination of events is not very common in practice, meaning that as long as the ancestor and descendant are not independent under the interventions, we usually converge to a graph with the correct orientation.

Besides, if the confounder $X_j$ of $X_1$ and $X_3$ is a parent of $X_2$, then the local optimum would disappear with learning that edge since $p(X_2|\text{do}(X_1 = x_1), X_j) = p(X_2|X_1 = x_1, X_j)$. In conclusion, for many of the graph structures like chain, bidiag, collider, full and jungle, this shows that there does not exist any local optima for the orientation probabilities. Only for the certain structures of confounded parents, there may exist local optima that depend on the specific distribution parameterization.

### B.2.3 Limited data regime

Assumption (3) and (4) are taken with respect to the data limit such that the conditions derived in the next section solely depend on the given causal graphical model. However, in practice, we often have a limited data set. The proof presented for the data limit is straightforward to extend to this setting with the following modification:

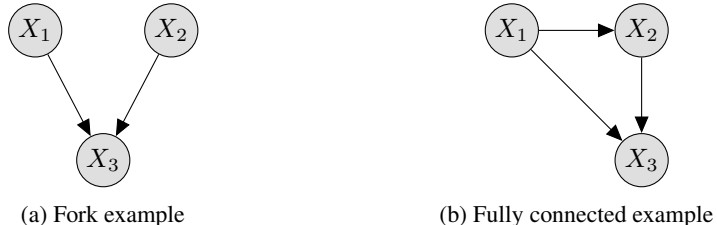

(a) Fork example    (b) Fully connected example

Figure 8: Causal graph structures for which, under specific parameterization of the conditional distributions, the conditions for guaranteeing convergence can be violated.

- The conditional distributions $p(X|...)$ are replaced by the conditional distributions that follow from the given, observational data.

- The expectations over the interventional data $\tilde{p}_{\hat{I}}(\boldsymbol{X})$ is replaced by the joint distribution over samples given for the intervention $\hat{I}$.

- Theorem B.1 and B.2 for the edge $X_i \to X_j$ are extended as follows:

  (3) *For all possible sets of parents of $X_i$ excluding $X_j$, adding $X_j$ does not improve the log-likelihood estimate of $X_i$ under the intervention on $X_j$, or leaves it unchanged:*

$$\forall \widehat{\mathrm{pa}}(X_i) \subseteq X_{-i,j} : \mathbb{E}_{I_{X_j}, \boldsymbol{X}} \left[ \log p(X_i | \widehat{\mathrm{pa}}(X_i), X_j) - \log p(X_i | \widehat{\mathrm{pa}}(X_i)) \right] \leq 0 \quad (36)$$

  This condition is the inverse statement of Equation 23, in the sense that we consider interventions on the child/descendant $X_j$. In the data limit, this naturally follows from Equation 23 and Equation 24, but in the limited data regime, we might have violations of Equation 36 due to biases in our samples. Violations of Equation 36 are the cause of ENCO predicting cyclic graphs as seen in Section 4.2.

- Finally, Theorem B.4 does not necessarily hold anymore since noise in our data can lead to an overestimation of edges. Thus, we add the following condition:

  (1) *For all pairs of variables $X_i, X_j$ for which there exists no direct causal relation in the true causal graph, and $X_j$ not being the ancestor of $X_i$, the following condition has to hold:*

$$\min_{\hat{\mathrm{pa}} \subseteq (\mathrm{gpa}_i(X_j) \setminus \mathrm{pa}(X_j))} \mathbb{E}_{\hat{I} \sim p_{I_{-j}}(I)} \mathbb{E}_{\tilde{p}_{\hat{I}}(\mathbf{X})} \left[ \log p(X_j | \mathrm{pa}(X_j), \hat{\mathrm{pa}}, X_i) - \log p(X_j | \mathrm{pa}(X_j), \hat{\mathrm{pa}}) \right] < \lambda_{\mathrm{sparse}}$$

$$(37)$$

  *where $\mathrm{gpa}_i(X_j)$ is the set of nodes excluding $X_i$ which, according to the ground truth graph, could have an edge to $X_j$ without introducing a cycle.*

  This condition ensures that no correlations due to sample biases introduce additional edges in the causal graphs.

If the conditions discussed above hold with respect to the given observational and interventional dataset, we can guarantee that ENCO will converge to the true causal graph given sufficient time.

### B.2.4 GRAPHS WITH LIMITED GUARANTEES

Most common causal graphs fulfill the conditions mentioned above, as long as a small enough value for $\lambda_{\mathrm{sparse}}$ is chosen. Still, there are situations where we cannot guarantee that ENCO convergences to the correct causal graph independent of the chosen value of $\lambda_{\mathrm{sparse}}$. Here, we want to discuss two scenarios visualized in Figure 8 under which the guarantees fail. Still, we want to emphasize that despite graphs not fulfilling the conditions, ENCO might still converge to the correct DAG for those as the guarantee conditions assume the worst-case scenarios for $\boldsymbol{\theta}$ and $\boldsymbol{\gamma}$ in all situations.

The first example we discuss is based on a fork structure where we have three binary variables, $\{X_1, X_2, X_3\}$, and the edges $X_1 \to X_3$ and $X_2 \to X_3$ (see Figure 8a). The parameterization we look at is a (noisy) XOR-gate for $X_3$ with its two input variables $X_1, X_2$ being independent of each other and uniformly distributed. The conditional probability distribution $p(X_3 | X_1, X_2)$ can be

summarized in the following probability function:

$$p(X_3 = 1|X_1, X_2) = \begin{cases} \epsilon & \text{if } X_1 = 0, X_2 = 0 \\ 1 - \epsilon & \text{if } X_1 = 1, X_2 = 0 \\ \epsilon & \text{if } X_1 = 0, X_2 = 1 \\ 1 - \epsilon & \text{if } X_1 = 1, X_2 = 1 \end{cases} \tag{38}$$

In other words, if $X_1 \neq X_2$, $X_3$ is equals 1 with a likelihood of $1 - \epsilon$. If $X_1 = X_2$, $X_3$ is equals 1 with a likelihood of $\epsilon$. The issue that this probability table creates is the following. Knowing only one out of the two variables does not improve the log likelihood estimate for the output. This is because $X_1$ and $X_2$ are independent of each other, and $p(X_3|X_1) = p(X_3)$ is a uniform distribution. Hence, the worst-case parent set in Equation 6 would be the empty set, and leads to an expected difference log-likelihood difference of zero. As $\lambda_{\text{sparse}}$ is required to be greater than zero for Theorem B.4, we cannot fulfill the condition for that graph. This means that an empty graph without any edges is a local minimum to which ENCO could converge. Yet, when the edge probabilities are non-zero, we will sample adjacency matrices with both input variables being a parent of $X_3$ with a non-zero probability. Hence, the log-likelihood difference for $X_1$ and $X_2$ to $X_3$ is unequal zero. Further, this graph is still often correctly discovered despite ENCO not having a convergence guarantee for it. We have conducted experiments on this graph with $\epsilon = \{0.1, 0.2, 0.3, 0.4, 0.45\}$ using a sparsity regularizer of $\lambda_{\text{sparse}} = 1\text{e-}4$, and in all cases, ENCO converged to the correct, acyclic graph. Note that values close to 0.5 for $\epsilon$ are most challenging, because the difference between the true conditional and marginal distribution goes against zero.

The second example we want to discuss aims at graphs that violate the condition in Theorem B.1, more specifically Equation 23. The graph we consider is a fully connected graph with three variables $X_1, X_2, X_3$ (see Figure 8b). The scenario can be described as follows: if knowing $X_2$ informs the log-likelihood estimate of $X_3$ more about $X_1$ than about $X_2$ itself, an intervention on $X_2$ and a sampled graph with the edge $X_2 \to X_3$ could lead to a worse likelihood estimate of $X_3$ than without the edge. For this scenario to happen, $p(X_2|X_1)$ must be close to deterministic. Additionally, $p(X_3|X_1, X_2)$ must be much less reliant on $X_2$ than on $X_1$, such as in the following probability density:

$$p(X_3 = 1|X_1, X_2) = \begin{cases} \epsilon_1 & \text{if } X_1 = 0, X_2 = 0 \\ 1 - \epsilon_1 & \text{if } X_1 = 1, X_2 = 0 \\ \epsilon_2 & \text{if } X_1 = 0, X_2 = 1 \\ 1 - \epsilon_2 & \text{if } X_1 = 1, X_2 = 1 \end{cases} \tag{39}$$

The two variables $\epsilon_1, \epsilon_2$ represent small constants close to zero. In this case, the graph can violate the condition in Equation 23 since intervening on $X_2$ breaks the dependency between $X_1$ and $X_2$. The conditional distribution $p(X_3|X_2)$ learned from observational data relies on the dependency between $X_1$ and $X_2$ which can make it to a worse estimate than $p(X_3)$. Note that if the edge $X_1 \to X_3$ is learned by ENCO though, this will not constitute a problem anymore since with conditioning on $X_1$, i.e. $p(X_3|X_2, X_1)$, the edge $X_2 \to X_3$ will gain a gradient towards the correct graph. Thus, when $\boldsymbol{\gamma}$ and $\boldsymbol{\theta}$ are not initialized with the worst-case values, the graph with both $X_1$ and $X_2$ as parents of $X_3$ can be sampled and provides gradients in the correct direction. Further, we did not observe any of these situations in the synthetic and real-world graphs we experimented on.

### B.2.5 CONDITIONS FOR THE GLOBAL OPTIMUM

So far, the discussion focused on proving that the optimization space does not contain any local optima with respect to the graph parameters $\boldsymbol{\gamma}$ and $\boldsymbol{\theta}$ besides the global optimum. If these conditions are violated, ENCO might still converge to the correct solution, since we are not guaranteed to find and get stuck in one of these local optima. Thus, in this section, we provide conditions under which the ground truth graph is the global optimum of the objective in Equation 2. Graphs that fulfill these conditions are very likely to be correctly identified by ENCO, but with a suboptimal choice of hyperparameters, initial starting conditions etc., we could return an incorrect graph.

The conditions and proof follow a similar structure to those as before for the local optima. We first discuss when we can guarantee that the global optima has the same orientation of edges, and then when we also find the correct parent set of the remaining variables.

**Theorem B.5.** *For every pair of variables $X_i, X_j$ where $X_i$ is a parent of $X_j$, the graph $\hat{G}$ that optimizes objective in Equation 2 models the orientation $X_i \to X_j$, if there exists an edge between $X_i$ and $X_j$ in $\hat{G}$, under the following conditions:*

- *$X_i$ and $X_j$ are not independent under observational data.*

- *Under interventions on $X_i$, $X_i$ and $X_j$ are not independent given the true parent set of $X_j$.*

*Proof.* If $X_i$ and $X_j$ are independent under observational data, the observational distributions would not identify any correlation among those two variables. Hence, transferring them for any graph to interventional data would have $p(X_i|...) = p(X_i|X_j,...)$, thus making the objective invariant to the orientation of the edge, and removing any edge between $X_i$ and $X_j$ for sparsity.

If $X_i$ and $X_j$ are dependent, we can prove the statement by showing that modeling the orientation $X_j \to X_i$ will strictly lead to a worse estimate under the intervention on $X_j$ since the orientation parameters are optimized by comparing the interventions of the two adjacent variables. Under interventions on $X_i$, the causal mechanism $p(X_j|\text{pa}(X_j))$, with $\text{pa}(X_j)$ being the parent set of the ground truth graph including $X_i$, remains invariant under interventions on $X_i$, and is strictly better than $p(X_j|\text{pa}(X_j)\backslash X_i)$ for estimating $X_j$ due to the direct causal relation. Under interventions on $X_j$, the causal mechanism $p(X_i|X_j,...)$ leads to a strictly worse estimate as discussed in Theorem B.1, since the dependency between $X_i$ and $X_j$ does not exist in the interventional regime. Hence, the inverse orientation of the edge $X_i \to X_j$, i.e. $X_j \to X_i$ cannot be part of the global optimum. $\square$

**Theorem B.6.** *For every pair of variables $X_i, X_j$ where $X_i$ is an ancestor but not direct parents of $X_j$, the graph $\hat{G}$ that optimizes objective in Equation 2 does not include the edge $X_j \to X_i$ if the conditions in Theorem B.5 hold.*

*Proof.* To show this statement, we need to consider different independence relations between $X_i$ and $X_j$. First, if $X_i$ and $X_j$ are independent in the observational dataset given any conditional set, the edge will be removed since any edge between two independent variables is removed for any $\lambda_{\text{sparse}} > 0$. The same holds if $X_i$ and $X_j$ are independent for interventions on $X_i$ and $X_j$.

If they are dependent, we can follow a similar argument as in Theorem B.2. The causal mechanism $p(X_j|X_i,...)$ transfers from observational to interventional data on $X_i$ since on interventions on $X_i$, the causal mechanism of $X_j$ is not changed. Further, when intervening on $X_j$, $X_i$ and $X_j$ become independent such that any mechanism $p(X_i|X_j,...)$ cannot transfer except if $X_i$ and $X_j$ are independent under interventions on $X_j$. In this case, the edge will be again removed by the sparsity regularizer. This shows that for any setting, the orientation of the edge $X_j \to X_i$ cannot lead to a better estimate than $X_i \to X_j$, and in case of independence, the edge $X_j \to X_i$ will be removed as well. $\square$

**Theorem B.7.** *The graph $\hat{G}$ that optimizes objective in Equation 2 models the same parent sets for each variable under the following conditions:*

- *The conditions in Theorem B.5 hold.*

- *For any variable $X_i$ with its true parent set $\text{pa}(X_i)$, there does not exist a smaller parent set $\hat{pa} \subset \boldsymbol{X} \setminus X_i, \text{descendants}(X_i)$ which approximates the log-likelihood of $X_i$ up to $\lambda_{\text{sparse}} \cdot (|\text{pa}(X_i)| - |\hat{pa}|)$ on average.*

- *The regularization parameter $\lambda_{\text{sparse}}$ is greater than zero.*

*Proof.* If the orientations for all edges are according to the ground truth graph in the global optimum following Theorem B.5 and B.6, the parent set for a variable $X_i$ is limited to those variables which are not descendants of $X_i$. From the ground truth graph, we know that, conditioned on the true parent set $\text{pa}(X_i)$, $X_i$ is independent of all other non-descendant variables. Thus, the log-likelihood estimate of $X_i$, i.e. the left part of the objective in Equation 2, is optimized by $p(X_i|\text{pa}(X_i))$. To show that this is also the global optimum when combining with the regularizer, we need to consider those parent sets of $X_i$ which obtain a lower penalty, i.e. smaller parent sets. The difference between two parent sets $\text{pa}(X_i)$ and $\hat{pa}$ in terms of the regularizer corresponds to $\lambda_{\text{sparse}} \cdot (|\text{pa}(X_i)| - |\hat{pa}|)$.

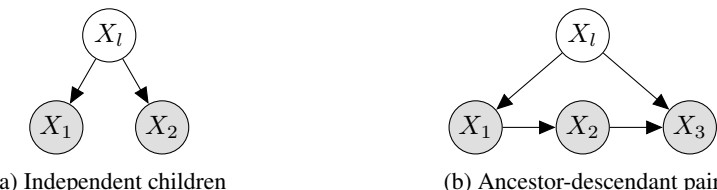

(a) Independent children         (b) Ancestor-descendant pair

Figure 9: Visualization of the different graph structures we need to consider in the guarantee discussion of latent confounder detection. Latent variables $X_l$ are shown in white, all other variables are observed. (a) The two children of $X_l$, $X_1$ and $X_2$, are independent of each other. (b) $X_1$ is an ancestor of $X_3$, and the two variables have a shared latent confounder $X_l$.

Thus, if there exists no parent set for which this difference is greater than the penalty for the worse log-likelihood estimate, the true parent set pa$(X_i)$ constitutes the global optimum. □

### B.3 CONVERGENCE CONDITIONS FOR LATENT CONFOUNDER DETECTION

In Section 3.5, we have discussed that ENCO can be extended to graph with latent confounders. For this, we have to record the gradients of $\gamma_{ij}$ for the interventional data on $X_i$ and all other interventional data separately. We define $\gamma_{ij} = \gamma_{ij}^{(I)} + \gamma_{ij}^{(O)}$ where $\gamma_{ij}^{(I)}$ is only updated with gradients from Equation 3 under interventions on $X_i$, and $\gamma_{ij}^{(O)}$ on all others. The score to detect latent confounders is:

$$\text{lc}(X_i, X_j) = \sigma\left(\gamma_{ij}^{(O)}\right) \cdot \sigma\left(\gamma_{ji}^{(O)}\right) \cdot \left(1 - \sigma\left(\gamma_{ij}^{(I)}\right)\right) \cdot \left(1 - \sigma\left(\gamma_{ji}^{(I)}\right)\right) \tag{40}$$

In this section, we show under which conditions the score $\text{lc}(X_i, X_j)$ converges to one if $X_i$ and $X_j$ share a latent confounder. We restrict our discussion to latent confounders between two variables that do not have any direct edges with each other, and assume that the confounder is not a child of any other observed variable. We assume that the causal graph based on the observed variable fulfills all conditions of Theorem B.1 to B.4 in Section B.2, meaning that without the latent confounders, the graph could have been recovered without errors. Under those conditions, we can also show that the graph among observed variables with latent confounders is also correctly recovered. This is since the latent confounders only affect Theorem B.4: if $X_i$ and $X_j$ share a latent confounder, they are not conditionally independent given their observed parents. Thus, we can rely on the fact that all edges in the true causal graph will be found according to Theorem B.1 to B.4, and the edges with latent confounders do not fulfill Theorem B.4.

For all pairs of variables that do not share a latent confounder, $\text{lc}(X_i, X_j)$ converges to zero. The edges that are removed in Theorem B.4 converge to $\sigma(\gamma_{ij}^{(O)}) = 0$ which sets $\text{lc}(X_i, X_j)$ to zero. For edges that have been recovered, we state in Equation 24 that the gradient for interventional data must be negative for interventions on the parent. Hence, $\sigma(\gamma_{ji}^{(I)})$ converges to one which brings $\text{lc}(X_i, X_j)$ to zero again.

For variables that share a latent confounder, we distinguish between two cases that are visualized in Figure 9. In the first case, we assume that $X_i$ and $X_j$ are independent in the true causal graph excluding the latent confounder. This means that an intervention on $X_i$ does not cause any change in $X_j$, and vice versa. The second case describes the situation where $X_i$ is an ancestor of $X_j$. The case of $X_i$ being a parent of $X_j$ has been excluded in earlier assumptions as in those cases, we cannot separate the causal effect of $X_i$ on $X_j$ based on its causal relation and the latent confounder.

In case that the two children of the latent confounder are not an ancestor-descendant pair, we can provide a guarantee under the following conditions.

**Theorem B.8.** *Consider a pair of variables $X_i, X_j$ that share a latent confounder $X_l$. Assume that $X_i$ and $X_j$ are conditionally independent given the latent confounder and their observed parents. Further, all other edges in the causal graph have been recovered under the conditions of Theorem B.1*

to B.4. The confounder score $lc(X_i, X_j)$ converges to one if the following two conditions hold:

$$\mathbb{E}_{\hat{I} \sim p_{I_{\cdot X_i}}(I)} \mathbb{E}_{\tilde{p}_{\hat{I}}(\boldsymbol{X})} \big[ \log p(X_j|pa(X_j), X_i) - \log p(X_j|pa(X_j)) \big] > \lambda_{sparse} \tag{41}$$

$$\mathbb{E}_{\hat{I} \sim p_{I_{\cdot X_j}}(I)} \mathbb{E}_{\tilde{p}_{\hat{I}}(\boldsymbol{X})} \big[ \log p(X_i|pa(X_i), X_j) - \log p(X_i|pa(X_i)) \big] > \lambda_{sparse} \tag{42}$$

*Proof.* We need to show that under the two conditions above, $\sigma(\gamma_{ij}^{(O)})$ and $\sigma(\gamma_{ji}^{(O)})$ are guaranteed to converge to one while $\sigma(\gamma_{ij}^{(I)})$ and $\sigma(\gamma_{ji}^{(I)})$ converge to zero. The distribution $p_{I_{\cdot X_k}}(I)$ represents the distribution over interventions excluding the ones performed on the variable $X_k$. The two conditions resemble Equation 28 with the difference that the intervention on the potential parent variable is excluded, and the parent set is the true parent set of the correct causal graph. This is because all other edges have been correctly recovered, and the two conditions are concerning $\sigma(\gamma_{ij}^{(O)})$. If the condition in Equation 41 holds, it corresponds to a negative gradient in $\gamma_{ij}^{(O)}$ following the argumentation in Theorem B.3. The same holds for $\gamma_{ji}^{(O)}$. Therefore, $\sigma(\gamma_{ij}^{(O)})$ and $\sigma(\gamma_{ji}^{(O)})$ are guaranteed to converge to one under the conditions given in Theorem B.8.

For the interventional parameters $\gamma_{ij}^{(I)}$ and $\gamma_{ji}^{(I)}$, we show that the gradient can only be positive, *i.e.* decreasing $\gamma_{ij}^{(I)}$ and $\gamma_{ji}^{(I)}$. Under the intervention on $X_i$, $X_i$ and $X_j$ become independent since we assume perfect interventions. In this case, the log-likelihood estimate of $X_j$ cannot be improved by conditioning on $X_i$. Hence, the difference $\mathcal{L}_{X_i \to X_j}(X_j) - \mathcal{L}_{X_i \nrightarrow X_j}(X_j)$ is greater or equal to zero. When further considering the sparsity regularizer $\lambda_{\text{sparse}}$, the gradient of $\gamma_{ij}$ under interventions on $X_i$ can only be positive, *i.e.* decreasing $\gamma_{ij}^{(I)}$. The same argument holds for $\gamma_{ji}^{(I)}$. Thus, we can conclude that $\sigma(\gamma_{ij}^{(I)})$ and $\sigma(\gamma_{ji}^{(I)})$ converge to zero. $\square$

If the conditions of Theorem B.8 are not fulfilled, $\sigma(\gamma_{ij}^{(O)})$ and $\sigma(\gamma_{ji}^{(O)})$ might converge to zero. This results in the score $lc(X_i, X_j)$ being zero, but also $\sigma(\gamma_{ij})$ converging to zero. Hence, we do not get any false positive edge predictions as we have seen in the experiments of Section 4.5.

For the second case where $X_i$ is an ancestor of $X_j$, we cannot give such a guarantee because of Theorem B.2. Theorem B.2 states that $\sigma(\theta_{ij})$ converges to one for ancestor-descendant pairs. However, $\sigma(\theta_{ji})$ is a factor in the gradients of $\gamma_{ji}$. This means that if $\sigma(\theta_{ji})$ converges to zero according to Theorem B.2, we cannot guarantee that $\gamma_{ji}$ converges to the desired value since its gradient becomes zero. Nevertheless, $59.2\%$ of the latent confounders in our experiments of Section 4.5 were on ancestor-descendant pairs. ENCO detects a majority of those confounders, showing that ENCO still works on such confounders despite not having guarantees. Further, we show in Section C.3 that the confounder scores $lc(X_i, X_j)$ indeed converge to one for detected confounders, and zero for all other edges.

### B.4    CONVERGENCE FOR PARTIAL INTERVENTION SETS

In Section 4.4, we have experimentally shown that ENCO works on partial intervention sets as well. Here, we will discuss convergence guarantees in the situation when interventions are not provided on all variables.

We start with discussing the case where, for a graph with $N$ variables, we are given samples of interventions on $N-1$ variables. In this case, we can rely on the previous convergence guarantees discussed in Appendix B.2 with minor modifications. Specifically, for the variable $X_i$ on which we do not have interventions, the orientation parameters $\theta_{i\cdot}$ are only updated by interventions on other variables. Hence, for this variable $X_i$, the following conditions need to hold instead of Theorem B.1:

- *For all possible sets of parents of $X_i$ excluding $X_j$, adding $X_j$ does not improve the log-likelihood estimate of $X_i$ under the intervention on $X_j$, or leaves it unchanged:*

$$\forall \widehat{pa}(X_i) \subseteq X_{-i,j} : \mathbb{E}_{I_{X_j}, \boldsymbol{X}} \big[ \log p(X_i|\widehat{pa}(X_i), X_j) - \log p(X_i|\widehat{pa}(X_i)) \big] \leq 0 \tag{43}$$

*For at least one parent set $\widehat{pa}(X_i)$, which has a probability greater than zero to be sampled, this inequality is strictly smaller than zero.*

This condition ensures that $\theta_{ij}$ converges to the correct values, where $X_i$ is the parent or ancestor of $X_j$. Thus, in conclusion, we can provide convergence guarantees if $N-1$ interventions are provided.

Next, we can consider the case of having $N-2$ interventions. With the conditions above, we can ensure that the all orientation parameters are learned, excluding $\theta_{ij}$ where $X_i$ and $X_j$ are the variables for which we have not obtained interventions. In this case, we cannot give strict convergence guarantees for the edge $X_i \leftrightarrow X_j$, especially when $X_i$ and $X_j$ have a direct causal relationship. If $X_j$ is the child of $X_i$, we might obtain the edge $X_j \rightarrow X_i$ which violates the assumptions in the second and third step of the proof. Therefore, we cannot give guarantees of correctness for incoming/outgoing edges of $X_i$ and $X_j$, and might make incorrect predictions of edges between these two variables.

When taking the next step to having $M$ interventions provided, ENCO can create more incorrect predictions. For the variables for which interventions are provided, we can use the same convergence guarantees (Theorem B.1-B.4) since all conditions are independent across variables. For variables without interventions, we cannot rely on those. While we have observed that learning the missing $\theta$'s from other interventions give reasonable results, we see a degradation of performance the further the distance is between a node and the closest intervened variable. As an example, suppose we have a chain with 5 variables, i.e. $X_1 \rightarrow X_2 \rightarrow X_3 \rightarrow X_4 \rightarrow X_5$, and we are provided with an intervention on $X_1$ only. This allows us to learn the orientation between $X_1$ and $X_2$. The orientation between $X_2$ and $X_3$ is often learned correctly as well because adding the edge $X_2 \rightarrow X_3$ instead of $X_3 \rightarrow X_2$ gives a greater decrease in overall log-likelihood, since part of the information from $X_3$ to predict $X_2$ is already included in $X_1$. However, the further we go away from $X_1$, the less information is shared between the child and the intervened variable. Moreover, the likelihood of a mistake occurring due to limited data further increases. This is why the orientation of the edge $X_4 \rightarrow X_5$ is not always learned correctly, which can also cause false positive edges.

Many scenarios of predicting false positive edges can, in theory, be solved by providing an undirected skeleton of the graph, for example, obtained from observational data. Still, one of the cornerstones of ENCO is that it does not assume faithfulness. Without faithfulness or any other assumption on the functional form of the causal mechanisms, the correct undirected graph cannot be recovered by any method. One of the future directions will be to include faithfulness in ENCO to solve the scenarios mentioned above, although this would imply that we might not be able to recover edges of deterministic variables anymore.

### B.5 EXAMPLE FOR NON-FAITHFUL GRAPHS

Below we give an example of a distribution which is not faithful with respect to its graph structure, but can yet be found by ENCO. Suppose we have a chain of three variables, $X_1 \rightarrow X_2 \rightarrow X_3$. For simplicity, we assume here that all the variables are binary, but the argument can similarly hold for any categorical data. The distribution $p(X_1)$ is an arbitrary function with $0 < p(X_1 = 1) < 1$, and the other two conditionals are deterministic functions: $p(X_2|X_1) = \delta[X_2 = X_1]$, $p(X_3|X_2) = \delta[X_3 = X_2]$. This joint distribution is not faithful to the graph, since $X_3$ is independent of $X_2$ given $X_1$, which is not implied by the graph. We will now focus our discussion on the edge $X_1 \rightarrow X_3$ to show that, despite the independence, the proposed method ENCO can identify the true parent set of $X_3$. The first step of ENCO is to fit the neural networks to the observational distributions, which include $p(X_3), p(X_3|X_1), p(X_3|X_2), p(X_3|X_1, X_2)$. Now, the update of the edge parameter $\gamma_{13}$ under interventions on $X_2$ can be summarized as follows, where we marginalize out over graph samples:

$$\frac{\partial \mathcal{L}}{\partial \gamma_{13}} = \sigma'(\gamma_{13}) \cdot \sigma(\theta_{13}) \cdot \mathbb{E}_{\mathbf{X}}\Big[ p(X_2 \rightarrow X_3) \cdot \underbrace{[-\log p(X_3|X_1, X_2) + \log p(X_3|X_2)]}_{\text{Gradients for graph samples with edge } X_2 \rightarrow X_3} +$$
$$p(X_2 \nrightarrow X_3) \cdot \underbrace{[-\log p(X_3|X_1) + \log p(X_3)]}_{\text{Gradients for graph samples without edge } X_2 \nrightarrow X_3} \Big] \tag{44}$$

where the samples $\mathbf{X}$ are sampled from the graph under interventions on $X_2$. Intuitively, the gradient points towards increasing the probability of the edge $X_1 \rightarrow X_3$ if adding $X_1$ to the conditionals improves the log-likelihood estimate of $X_3$ in both graph samples, i.e. where we have the edge $X_2 \rightarrow X_3$ or not. Thus, to show that the gradient points towards decreasing the probability of the

edge $X_1 \rightarrow X_3$, we need to show that both of the above log-likelihood differences are greater than zero (note that positive gradients lead to a decrease since we minimize the objective).

In the first difference, $\mathbb{E}_{\mathbf{X}}\left[-\log p(X_3|X_1, X_2) + \log p(X_3|X_2)\right]$, we know that $p(X_3|X_2)$ is the optimal estimator, since the ground truth data is generated via this conditional. Thus, independent of what function the network has learned for $p(X_3|X_1, X_2)$, the difference above can be only greater or equals to zero. Note that for $p(X_3|X_1, X_2)$, there are value combinations that have never been observed, i.e. $X_1 \neq X_2$, such that in practice we can't guarantee a specific distribution to be learned for such conditionals. Still, this does not constitute a problem for finding the graph as shown above.

The second difference, $\mathbb{E}_{\mathbf{X}}\left[-\log p(X_3|X_1) + \log p(X_3)\right]$, can be similarly reasoned. Since $X_1$ is independent of $X_3$ under interventions on $X_2$, $p(X_3|X_1)$ cannot lead to a better estimator than $p(X_3)$, even when trained on the interventional data. However, since $p(X_3|X_1)$ was trained on observational data, where there exist a strong correlation between $X_1$ and $X_3$, this estimator must be strictly worse than $p(X_3)$. Hence, this difference will be strictly positive.

To summarize, under interventions on $X_2$, the edge $X_1 \rightarrow X_3$ will be trained towards decreasing its probability. Further, under interventions on $X_1$, the effect of $X_1$ on $X_3$ can be fully expressed by conditioning on $X_2$, making this gradient going to zero when the edge probability $X_2 \rightarrow X_3$ goes towards one. For the edge $X_2 \rightarrow X_3$ itself, the same reasoning as here can be followed such that independent of whether the edge $X_1 \rightarrow X_3$ is included in the graph or not, conditioning $X_3$ on $X_2$ can only lead to an improvement in its estimator. Therefore, ENCO is able to find the correct graph despite it not being faithful.

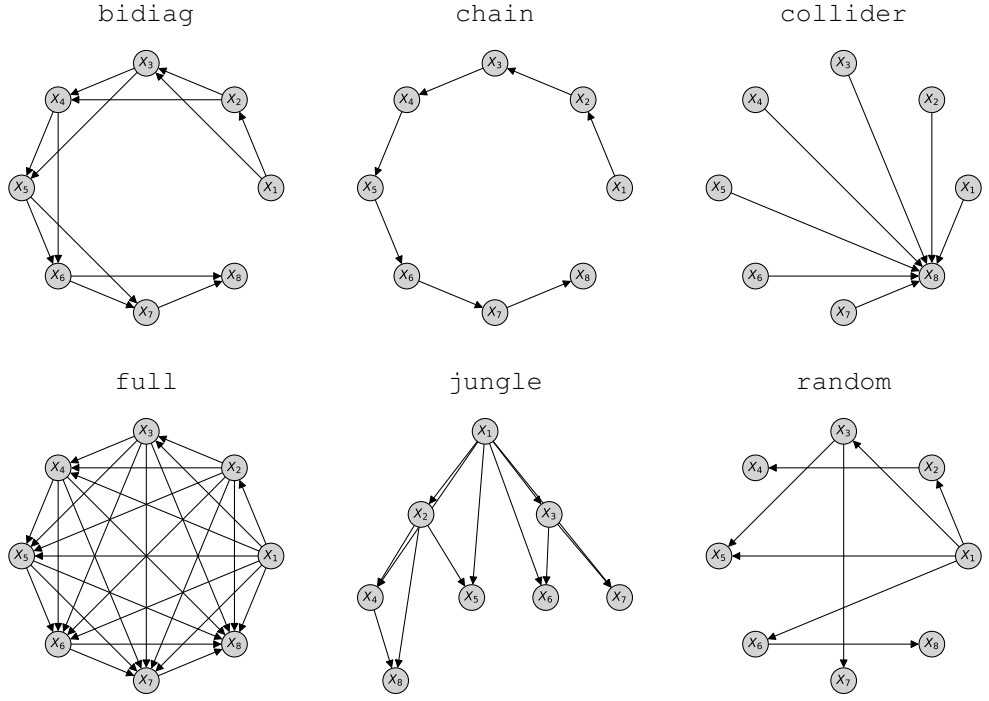

Figure 10: Visualization of the common graph structures for graphs with 8 nodes. The graphs used in the experiments had 25 nodes. Note that the graph random is more densely connected for larger graphs, as the number of possible edges scales quadractically with the number of nodes.

## C   EXPERIMENTAL DETAILS

The following section gives an overview of the hyperparameters used across experiments. Additionally, we discuss details of the graph generation and the learning process of different algorithms.

### C.1   COMMON GRAPH STRUCTURE EXPERIMENTS

#### C.1.1   DATASETS

**Graph generation**   The six common graph structures we have used for the experiments in Table 1 are visualized in Figure 10. In the graph bidiag, a variable $X_i$ has $X_{i-2}$ and $X_{i-1}$ as parents, and consequently $X_{i+1}$ and $X_{i+2}$ as children. Hence, this graph represents a chain with 2-hop connections. The graph chain is a bidiag with a single hop, meaning that $X_i$ is the parent of $X_{i-1}$ but not $X_{i-2}$. In the graph collider, the variable $X_N$ has all other variables, $\boldsymbol{X}_{-i}$, as parents. In the graph full, the parents of a variable $X_i$ are all previous variables: $\text{pa}(X_i) = \{X_1, X_2, ..., X_{i-1}\}$. Hence, it is the densest connected graph possible. The graph jungle represents a binary tree where a node is also connected to its parent's parent. Finally, the graph random follows a randomly sampled adjacency matrix. For every possible pair of variables $X_i, X_j$, we sample an edge with a likelihood of $0.3$. To determine the orientation of the edges, we assume the causal ordering of $X_i \succ X_{i+1}$. In other words, if we have an edge between $X_i$ and $X_j$, it is oriented $X_i \rightarrow X_j$ is $i < j$ else $X_j \rightarrow X_i$.

**Conditional distributions**   In the generated graphs, we use categorical variables that each have 10 categories. To model the ground-truth conditional distributions, we use randomly initialized neural networks. Specifically, we use MLPs of two layers where the categorical inputs are represented by embedding vectors. For a variable $X_i$ with $M$ parents, we stack the $M$ embedding vectors to form the input to the following MLPs. Each embedding has a dimensionality of 4, hence the input size to the first linear layer is $4M$. The hidden size of the layers is 48, and we use a LeakyReLU activation function in between the two linear layers. Finally, a softmax activation function is used on

Table 4: Hyperparameter overview for the simulated graphs dataset experiments presented in Table 1.

| Hyperparameters | SDI | ENCO |
|---|---|---|
| Sparsity regularizer $\lambda_{\text{sparse}}$ | {0.01, 0.02, 0.05, 0.1, 0.2} | {0.002, 0.004, 0.01} |
| DAG regularizer | {0.2, 0.5, 1.0, 2.0, 5.0} | - |
| Distribution model | 2 layers, hidden size 64, LeakyReLU($\alpha = 0.1$) | |
| Batch size | 128 | |
| Learning rate - model | {2e-3, 5e-3, 2e-2, 5e-2} | |
| Weight decay - model | {1e-5, 1e-4, 1e-5} | |
| Distribution fitting iterations $F$ | 1000 | |
| Graph fitting iterations $G$ | 100 | |
| Graph samples $K$ | 100 | |
| Epochs | 50 | 30 |
| Learning rate - $\gamma$ | {5e-3, 2e-2, 5e-2} | {5e-3, 2e-2, 5e-2} |
| Learning rate - $\theta$ | - | {5e-3, 2e-2, 5e-2, 1e-1} |

the output to obtain a distribution over the 10 categories. The MLP and hyperparameters have been chosen based on the design of networks used in ENCO, SDI (Ke et al., 2019) and DCDI (Brouillard et al., 2020). For the initialization of the networks, we follow Ke et al. (2019) and use the orthogonal initialize with a gain of 2.5. The biases are thereby initialized uniformly between $-0.5$ and $0.5$. This way, we obtain non-trivial, random distributions. Experiments with different synthetic distributions are provided in Appendix D.7.

### C.1.2 METHODS AND HYPERPARAMETERS

**Baseline implementation**    We used existing implementations to run the baselines GIES (Hauser & Bühlmann, 2012), IGSP (Wang et al., 2017), GES (Chickering, 2002) and DCDI (Brouillard et al., 2020). For GIES, we used the implementation from the R package `pcalg`[2]. To run categorical data, we used the `GaussL0penIntScore` score function. For IGSP, we used the implementation of the python package `causaldag`[3]. As IGSP uses conditional independence tests in its score function, we cast the categorical data into continuous space first and experiment with different kernel-based independence tests. Due to its long runtime for large dataset sizes, we limit the interventional and observational data set size to 25k. Larger dataset sizes did not show any significant improvements. For details on the observational GES experiments, see Section D.6. Finally, we have used the original python code for DCDI published by the authors[4]. We have added the same neural networks used by ENCO into the framework to perform structure learning on categorical data. Bugs in the original code were corrected to the best of our knowledge. Since SDI (Ke et al., 2019) has a similar learning structure as ENCO, we have implemented it in the same code base as ENCO. This allows us to compare the learning algorithms under exact same perquisites. Further, all methods with neural networks used the deep learning framework PyTorch (Paszke et al., 2019) which ensures a fair run time comparison across methods.

**Hyperparameters**    To ensure a fair comparison, we performed a hyperparameter search for all methods. The hyperparameter search was performed on a hold-out set of graphs containing two of each graph structure.

**GIES**    We performed a hyperparameter search over the regularizer values $\lambda \in \{0.01, 0.02, 0.05, 0.1, 0.2, 0.5, 1, 2, 5, 10, 20, 50, 100, 200\}$. The value obtaining the best results in terms of structural hamming distance (SHD) was $\lambda = 20$. The average run time of GIES was 2mins per graph.

---

[2]`https://cran.r-project.org/web/packages/pcalg/index.html`
[3]`https://github.com/uhlerlab/causaldag`
[4]`https://github.com/slachapelle/dcdi`

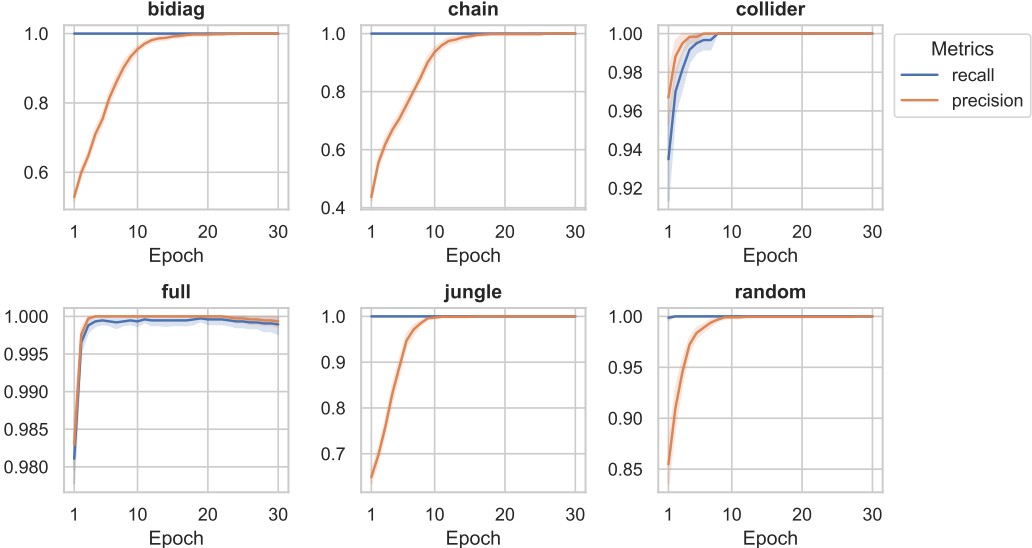

Figure 11: Learning curves of ENCO in terms of recall and precision on edge predictions for synthetic graph structures. The orientations for the ground-truth edges are not plotted as they have usually been correctly learned after the first epoch except for the graph `full`. Overall, we see that the edge recall starts very high for all graphs, and the precision catches up over the epochs. This is in line the steps in the convergence proof in Section B.

**IGSP** We experimented with two different conditional independence tests, `kci` and `hsic`, and different cutoff values $\alpha = \{1\text{e-}5, 1\text{e-}4, 1\text{e-}3, 1\text{e-}2\}$. The best hyperparameter setting was `kci` with $\alpha = 1\text{e-}3$. The average run time of IGSP was 13mins.

**SDI** We focused the hyperparameter search for SDI on its two regularizers, $\lambda_{\text{sparse}}$ and $\lambda_{\text{DAG}}$, as well as its learning rate for $\gamma$. The other hyperparameters with respect to the neural networks were kept the same as ENCO for a fair comparison. We show all details of the hyperparameter search in Table 4. The best combination of regularizers found was $\lambda_{\text{sparse}} = 0.02$ and $\lambda_{\text{DAG}} = 0.5$. Lower values of $\lambda_{\text{sparse}}$ lead to more false positives, especially in sparse graphs, while a lower value of $\lambda_{\text{DAG}}$ caused many two-variable loops. Compared to the reported hyperparameter by Ke et al. (2019) ($\lambda_{\text{DAG}} = 0.5, \lambda_{\text{sparse}} = 0.1$), we found a lower sparsity regularizer to work better. This is likely because of testing SDI on larger graphs. In contrast to ENCO, SDI needed a lower learning rate for $\gamma$ due to its higher variance gradient estimators. To compensate for it, we ran it for 50 instead of 30 epochs. In general, SDI achieved lower scores than in the original experiments by Ke et al. (2019) which was because of the larger graph size and smaller dataset size. The average run time of SDI was 4mins per graph.

**DCDI** The most crucial hyperparameter in DCDI was the initialization of the constraint factor $\mu_0$. We experimented with a range of $\mu_0 \in \{1\text{e-}10, 1\text{e-}9, 1\text{e-}8, 1\text{e-}7, 1\text{e-}6, 1\text{e-}5\}$ and found $\mu_0 = 1\text{e-}9$ to work best. This is close to the reported value of 1e-8 by Brouillard et al. (2020). Higher values lead to empty graphs, while lower values slowed down the optimization. Additionally, we search over the regularizer hyperparameter $\lambda \in \{1\text{e-}3, 1\text{e-}2, 1\text{e-}1, 1.0, 10.0\}$ where we found $\lambda = 0.1$, which is the same value used by Brouillard et al. (2020). We stop the search after the Lagrangian constraint is below 1e-7, following Brouillard et al. (2020), or 50k iterations have been used which was sufficient to converge on all graphs. We have experimented with using weight decay to prevent overfitting, but did not find it to provide any gain in terms of structure learning performance. The average run time of DCDI was 16 minutes.

**ENCO** We outline the hyperparameters of ENCO in Table 4. As discussed in Section 3.4, the most crucial hyperparameter in ENCO is the sparsity regularizer $\lambda_{\text{sparse}}$. The larger it is, the faster ENCO converges, but at the same time might miss edges in the ground-truth graph.

Table 5: Extension of Table 1 with the metric *structural intervention distance* (SID) (lower is better), averaged over 25 graphs each. The conclusion is the same as for SHD, namely that ENCO outperforms all baselines, while the acyclic heuristic has an even greater impact.

| Graph type | bidiag | chain | collider | full | jungle | random |
|---|---|---|---|---|---|---|
| GIES | 460.0 ($\pm$60.1) | 224.2 ($\pm$87.3) | 83.6 ($\pm$143.5) | 527.9 ($\pm$35.7) | 441.4 ($\pm$26.1) | 471.9 ($\pm$33.7) |
| IGSP | 423.3 ($\pm$48.2) | 240.1 ($\pm$78.8) | 120.7 ($\pm$51.4) | 554.8 ($\pm$26.4) | 394.8 ($\pm$73.5) | 524.0 ($\pm$18.8) |
| SDI | 243.7 ($\pm$55.1) | 70.2 ($\pm$46.8) | 24.0 ($\pm$0.0) | 537.1 ($\pm$29.2) | 180.0 ($\pm$56.4) | 317.9 ($\pm$62.7) |
| DCDI | 369.3 ($\pm$47.5) | 233.4 ($\pm$24.8) | 10.9 ($\pm$3.6) | 183.6 ($\pm$54.5) | 339.4 ($\pm$59.1) | 158.6 ($\pm$69.1) |
| ENCO (ours) | 28.2 ($\pm$18.8) | 19.9 ($\pm$17.9) | **7.4** ($\pm$13.3) | 63.2 ($\pm$22.6) | 16.3 ($\pm$9.8) | 77.6 ($\pm$27.2) |
| ENCO-acyclic (ours) | **0.0** ($\pm$0.0) | **0.0** ($\pm$0.0) | **7.4** ($\pm$13.3) | **17.7** ($\pm$8.9) | **4.6** ($\pm$8.1) | **5.3** ($\pm$11.8) |

Lower values allow the detection of more edges for the price of longer training times. We have found that for the graph structures given, only the graph `full` was sensitive to the value of $\lambda_{\text{sparse}}$ where $\lambda_{\text{sparse}} = 0.002$ and $\lambda_{\text{sparse}} = 0.004$ performed almost equally well. In comparison to SDI, ENCO can make use of larger learning rates due to lower variance gradient estimators. Especially for $\boldsymbol{\theta}$, we have noticed that high learning rates are beneficial. This is in line with our theoretical guarantees which require the orientation parameters to converge first. We use the Adam optimizer for $\boldsymbol{\gamma}$ and $\boldsymbol{\theta}$ with the $\beta$-hyperparameters $(0.9, 0.9)$ and $(0.9, 0.999)$ respectively. A lower $\beta_2$ hyperparameter for $\boldsymbol{\gamma}$ allows the second momentum to adapt faster to a change of gradient scale which happens for initial false positive predictions.

The average run time of ENCO was 2mins per graph. The algorithm could be sped up even more by reducing the number of graph samples $K$ and model fitting iterations. However, for graphs of larger than 100 nodes, $K = 100$ and longer model fitting times showed to be beneficial. The learning curves in terms of recall and precision are shown in Figure 10.

**Enforcing acyclicity** ENCO is guaranteed to converge to acyclic graphs in the data limit; arguably, an assumption that does not always hold. In the presence of cycles, which can occur especially when low data is available, a simple heuristic is to keep the graph, which maximizes the orientation probabilities. Specifically, we aim to find the order $O \in S_N$, where $S_N$ represents the set of all permutations from 1 to the number of variables $N$, for which we maximize the following objective:

$$\hat{O} = \arg \max_O \left[ \prod_{i=1}^{N} \prod_{j=i+1}^{N} \sigma(\theta_{O_i, O_j}) \right] \tag{45}$$

For small cycles it is easy to do this exhaustively by checking all permutations. For larger cycles, we apply a simple greedy search that works just as well. Once the order $\hat{O}$ has been found, we remove all edges $X_i \rightarrow X_j$ where $i$ comes after $j$ in $\hat{O}$. This guarantees to result in an acyclic graph.

The intuition behind this heuristic is the following. Cycles are often caused by a single orientation pair being incorrect due to noise in the interventional data. For example, in a chain $X_1 \rightarrow X_2 \rightarrow X_3 \rightarrow X_4 \rightarrow X_5$, it can happen that the orientation parameter $\theta_{14}$ is incorrectly learned as orientation of the edge between $X_1$ and $X_4$ as $X_4 \rightarrow X_1$ if the interventional data on $X_4$ does not show the independence of $X_1$ and $X_4$. However, most other orientation parameters, e.g. $\theta_{12}, \theta_{13}, \theta_{24}, \theta_{34}$, etc., have been likely learned correctly. Thus, it is easy to spot that $\theta_{14}$ is an outlier, and this is what the simple heuristic above implements.

**Results** Besides the structural hamming distance, a common alternative metric is *structural intervention distance* (SID) (Peters & Bühlmann, 2015). In contrast to SHD, SID quantifies the closeness between two DAGs in terms of their corresponding causal inference statements. Hence, it is suited for comparing causal discovery methods. The results of the experiments on the synthetic graphs in terms of SID are shown in Table 5, and show a similar trend as before, namely that ENCO is outperforming all baselines.

## C.2 SCALABILITY EXPERIMENTS

**Graph generation** For generating the graphs, we use the same strategy as for the graph `random` in the previous experiments. The probability of sampling an edge is set to $8/N$, meaning that on

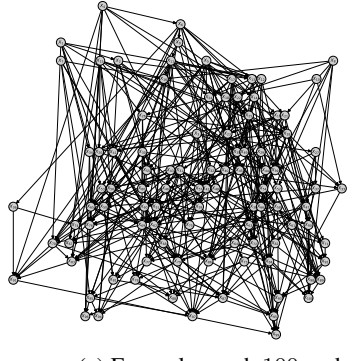

(a) Example graph 100 nodes

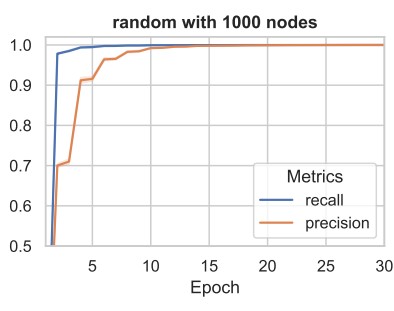

(b) Learning curve $1,000$ nodes

Figure 12: (a) Example graph of 100 variables (best viewed electronically). Every node has on average 8 edges and a maximum of 10 parents. (b) Plotting recall and precision of the edge predictions for the training on graphs with $1,000$ nodes. The small standard deviation across graphs shows that ENCO can reliably recover large graphs.

average, every node has 8 in- and outgoing edges. We limit the number of parents to 10 per node since, otherwise, we cannot guarantee that the randomly sampled distributions take all parents faithfully into account. This is also in line with the real-world inspired graphs of the BnLearn repository, which have a maximum of 6 parents. To give an intuition on the complexity of such graphs, we show an example graph of 100 nodes in Figure 12a. Accordingly to the number of variables, we have increased the data set size to 4096 samples per intervention, and 100k observational samples. We did not apply the order heuristic on the predictions, since ENCO was able to recover acyclic graphs by itself with the given data.

$\gamma$-**freezing stage** For ENCO, one challenge of large graphs is that the orientation parameters $\boldsymbol{\theta}$ are updated very sparsely. The gradients for $\theta_{ij}$ require data from an intervention on one of its adjacent nodes $X_i$ or $X_j$, which we evaluate less frequently with increasing $N$ as we iterate over interventions on all $N$ nodes. Hence, we require more iterations/epochs just for training the orientation parameters while wasting a lot of computational resources. To accelerate training of large graphs, we freeze $\boldsymbol{\gamma}$ in every second graph fitting stage. Updating only $\boldsymbol{\theta}$ allows us to use the same graph sample $C_{-ij}$ for both $\mathcal{L}_{X_i \to X_j}(X_j)$ and $\mathcal{L}_{X_i \not\to X_j}(X_j)$ since the log-likelihood estimate of $X_j$ only needs to be evaluated for $\theta_{ij}$. With this gradient estimator, we experience that as little as 4 graph samples are sufficient to obtain a reasonable gradient variance. Hence, it is possible to perform more gradient updates of $\boldsymbol{\theta}$ in the same computation time. Note that this is estimator not efficient when training $\boldsymbol{\gamma}$ as we require different $C_{-ij}$ samples for every $i$. In experiments, we alternate the standard graph fitting step with this pure $\theta$-training stage. We want to emphasize that this approach can also be used for small graphs obtaining similar results as in Table 1. However, it is less needed because the orientation parameters are more frequently updated in the first place. Such an approach is not possible for the baselines, SDI and DCDI, because they do not model the orientation as a separate variable.

**Hyperparameters** To experiment with large graphs, we mostly keep to the same hyperparameters as reported in Section C.1. However, all methods showed to gain by a small hyperparameter search. For SDI and ENCO, we increase the number of distribution fitting iterations as the neural networks need to model a larger set of possible parents. We also increase the learning rate of $\boldsymbol{\gamma}$ to 2e-2. However, while SDI reaches better performance with the increased learning rate at epoch 30, it showed to perform worse when training for longer. This indicates that high learning rates can lead to local minima in SDI. Additionally, we noticed that a slightly higher sparsity regularizer improved convergence speed for ENCO while SDI did not improve with a higher sparsity regularizer. Table 6 shows a hyperparameter overview of ENCO on large-scale graphs, and Figure 12b the learning curve on graphs of $1,000$ nodes.

For DCDI, we noticed that the hyperparameters around the Lagrangian constraint needed to be carefully fine-tuned. The Lagrangian constraint can reach values larger than possible to represent with double, and starts with 8e216 for graphs of $1,000$ nodes. Following Brouillard et al. (2020), we

normalize the constraint by the value after initialization, which gives us a more reasonable value to start learning. We performed another hyperparameter search on $\mu_0$, noticed however that it did not have a major impact. In the run time of ENCO, DCDI just starts to increase the weighting factor of the augmented Lagrangian while the DAG constraint is lower than 1e-10 for the smallest graph. The best value found was $\mu_0 = 1\mathrm{e}\text{-}7$.

Table 6: Hyperparameter overview of ENCO for the scalability experiments presented in Table 7. Numbers in the center represent that we use the same value for all graphs.

| **Hyperparameters** | 100 nodes | 200 nodes | 400 nodes | 1000 nodes |
|---|---|---|---|---|
| Distribution model | 2 layers, hidden size 64, LeakyReLU($\alpha = 0.1$) | | | |
| Batch size - model | 128 | 128 | 128 | 64 |
| Learning rate - model | 5e-3 | | | |
| Distribution fitting iterations $F$ | 2000 | 2000 | 4000 | 4000 |
| Graph fitting iterations $G$ | 100 | | | |
| Graph samples | 100 | | | |
| Learning rate - $\gamma$ | 2e-2 | | | |
| Learning rate - $\theta$ | 1e-1 | | | |
| $\theta$-training iterations | 1000 | 1000 | 2000 | 2000 |
| $\theta$-training graph samples | 2 + 2 | | | |
| Sparsity regularizer $\lambda_{\text{sparse}}$ | {0.002, 0.004, 0.01} | | | |
| Number of epochs (max.) | 30 | | | |

**Results**    For clarity, we report the results of all methods below. The exact values might not be easily readable in Figure 3 due to large differences in performance.

Table 7: Results for graphs between 100 and 1000 nodes. We report the average and standard deviation of the structural hamming distance (SHD) over 10 randomly sampled graphs. [†]The maximum runtime of ENCO was measured on an NVIDIA RTX3090. Baselines were executed on the same hardware.

| **Graph size** **Max Runtime**[†] | 100 15mins | 200 45mins | 400 2.5hrs | 1000 9hrs |
|---|---|---|---|---|
| DCDI (Brouillard et al., 2020) | 583.1 ($\pm$21.8) | 1399.0 ($\pm$67.5) | 4761.2 ($\pm$303.4) | OOM |
| SDI (Ke et al., 2019) | 35.7 ($\pm$2.9) | 100.9 ($\pm$7.6) | 356.3 ($\pm$11.5) | 1314.4 ($\pm$58.5) |
| ENCO (Ours) | 0.0 ($\pm$0.0) | 0.0 ($\pm$0.0) | 0.0 ($\pm$0.0) | 0.2 ($\pm$0.42) |

## C.3    LATENT CONFOUNDER EXPERIMENTS

**Graph generation**    The graphs used for testing the latent confounding strategy are based on the `random` graphs from Section 4.2. We use graphs of 25 nodes, and add 5 extra nodes that represent latent confounders. Each latent confounder $X_l$ is connected to two randomly sampled nodes $X_i$, $X_j$ that do not have a direct connection. However, $X_i$ and $X_j$ can be an ancestor-descendant pair and have any other (shared) parent set (see Figure 13a). In the adjacency matrix, we add the edges $X_l \rightarrow X_i$ and $X_l \rightarrow X_j$, and perform the data generation as for the previous graphs. After data generation, we remove the 5 latent confounders from both observational and interventional data. The task is to learn the graph structure of the remaining 25 observable variables, as well as detecting whether there exists a latent confounder between any pair of variables. We use the same setup in terms of dataset size as before for the observational samples, namely 5k, but increased the samples per intervention to 512. Little interventional data showed to cause a high variance in the interventional gradients, $\gamma_{ij}^{(I)}$, which is why more false positives occured. The results in for the limited data with 200 interventions, and results in the data limit, *i.e.* for 10k interventional samples and 100k observational samples, are shown in Table 8.

Table 8: Results of ENCO on detecting latent confounders averaged over 25 graphs with 25 nodes in the data limit (10k samples per intervention, 100k observational samples) and limited data (200 samples per intervention, 5k observational samples). In the data limit, only false negative predictions of latent confounders occured which did not affect other edge predictions. With little interventional data, more false positives occur reducing the precision.

| Metrics | ENCO (10k interv./100k observ.) | ENCO (512 interv./5k observ.) | ENCO (200 interv./5k observ.) |
|---|---|---|---|
| SHD | 0.0 ($\pm$0.0) | 1.24 ($\pm$1.33) | 4.12 ($\pm$1.86) |
| Confounder recall | 96.8% ($\pm$9.5%) | 93.6% ($\pm$13.8%) | 92.0% ($\pm$11.5%) |
| Confounder precision | 100.0% ($\pm$0.0%) | 96.5% ($\pm$7.1%) | 83.8% ($\pm$16.4%) |

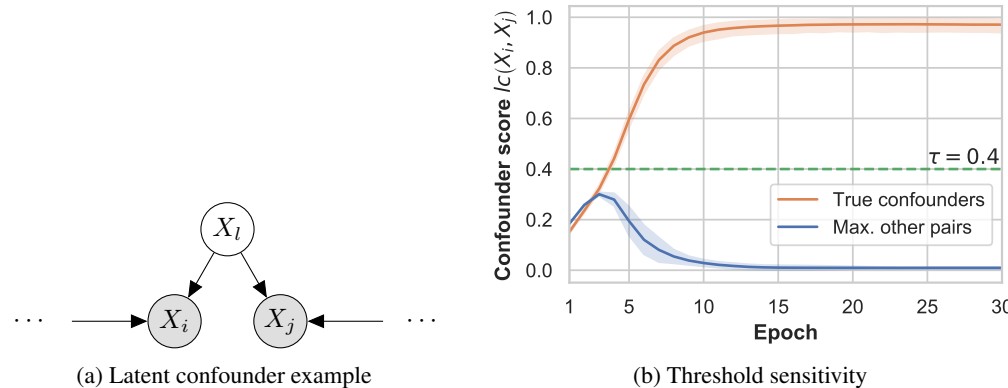

(a) Latent confounder example          (b) Threshold sensitivity

Figure 13: **Left**: Example of a latent confounder scenario, where $X_l$ is not observed and introduces a dependency between $X_i$ and $X_j$ on observational data. The dots on the left and right represent eventual (observed) parents of $X_i$ and $X_j$. **Right**: Plotting the average score lc($X_i, X_j$) for confounders $X_i \leftarrow X_l \rightarrow X_j$ in the true causal graph (orange) and maximum score of any other node pair (blue). The plot shows the detection of latent confounders in ENCO is not sensitive to the specific value of $\tau$.

**Baselines**   None of our previous continuous optimization baselines, *i.e.*, SDI and DCDI, are able to deal with latent confounders. To the best of our knowledge, other methods that are able to handle latent confounders commonly take assumptions that do not hold in our experimental setup. Further, most methods are able to deal with latent confounders in the sense that they obtain the correct results despite latent confounding being present. However, in our case, we explicitly predict latent confounders which is a different task by itself.

**Hyperparameters**   To show that we can perform latent confounder detection without specific hyperparameters, we use the same hyperparameters as for the experiment on the previous graph structures (see Appendix C.3). To record $\gamma_{ij}^{(I)}$ and $\gamma_{ij}^{(O)}$ separately, we use separate first and second order momentum parameters in the Adam optimizer. We plot in Figure 13b the latent confounder scores lc($X_i, X_j$) calculated based on Equation 8. We see that the score converges close to 1 for pairs with a latent confounder, and for all other, it converges to 0. This verifies our motivation of the score function discussed in Section 3.5, and also shows that the method is not sensitive to the threshold hyperparameter $\tau$. We choose $\tau = 0.4$ which was slightly higher than the highest value recorded for any other pair at early stages of training.

## C.4   INTERVENTIONS ON FEWER VARIABLES

**Datasets**   We perform the experiments of interventions on fewer variables on the same graphs and datasets as used for the initial synthetic graphs (see Section C.1). To simulate having interventions on fewer variables, we randomly sample a subset of variables for which we include the interventional data, and remove for all others. The sampled variables are the same for both ENCO and DCDI, and

differ across graphs. The dataset size is the same as before, namely 200 samples per intervention and 5k observational datasets.

**ENCO for partial intervention sets**    While the theoretical guarantees for convergence to an acyclic graph apply when interventions on all variables are possible, it is straightforward to extend the ENCO algorithm to support partial interventions as well. Normally, in the graph fitting stage, we sample one intervention at a time. We can, thus, simply restrict the sampling only to the interventions that are possible (or provided in the dataset). In this case, we update the orientation parameters $\theta_{ij}$ of only those edges that connect to an intervened variable, either $X_i$ or $X_j$, as before. All other orientation parameters would remain unchanged throughout the training, since their gradients rely on interventions missing from the dataset. Instead, we extend the gradient estimator in Equation 4 to not be exclusive to adjacent interventions, but include interventions on all variables. Specifically, for the orientation parameter $\theta_{ij}$ without any interventions on $X_i$ or $X_j$, we use the following gradient estimator:

$$
\begin{aligned}
\frac{\partial}{\partial \theta_{ij}} \tilde{\mathcal{L}} = \sigma'(\theta_{ij}) \Big( & \sigma(\gamma_{ij}) \cdot \mathbb{E}_{I_{X_k}, \boldsymbol{X}, C_{-ij}} \left[ \mathcal{L}_{X_i \to X_j}(X_j) - \mathcal{L}_{X_i \not\to X_j}(X_j) \right] - \\
& \sigma(\gamma_{ji}) \cdot \mathbb{E}_{I_{X_k}, \boldsymbol{X}, C_{-ij}} \left[ \mathcal{L}_{X_j \to X_i}(X_i) - \mathcal{L}_{X_j \not\to X_i}(X_i) \right] \Big)
\end{aligned}
\tag{46}
$$

where we have an intervention on an arbitrary variable $X_k$ with $k \neq i, k \neq j$. This still represents an unbiased gradient estimator since in the derivation of the estimator, we excluded interventions on other variables only to reduce noise.

ENCO has been designed under the assumption that interventional data is provided. When we have interventional data on only a very small subset of variables, we might not optimally use the information that is provided by the observational data. To overcome this issue, we can run a causal discovery method that solely work on observational data and return an undirected graph. This skeleton can be used as a prior, and prevents false positive edges between conditionally independent variables.

**Hyperparameters**    We reuse the hyperparameters of the experiments on the synthetic graph except that we use a slighly smaller sparsity regularizer, $\lambda_{\text{sparse}} = 0.002$, and a weight decay of $4e$-5. For the orientation parameters without adjacent intervention, we use a learning rate of $0.1 \cdot \text{lr}_\theta$ which is $1e$-3 for this experiment. For DCDI, we observed that a higher regularization term of $\lambda = 1.0$ obtained best performance. All other hyperparameters are the same as in Section C.1.

**Results**    For additional experimental results, see Section D.2.

## C.5    REAL-WORLD INSPIRED EXPERIMENTS

**Datasets**    We perform experiments on a collection of causal graphs from the Bayesian Network Repository (BnLearn) (Scutari, 2010). The repository contains graphs inspired by real-world applications that are used as benchmarks in literature. We chose the graphs to reflect a variety of sizes and different challenges (rare events, deterministic variables, etc.). The chosen graphs are cancer (Korb & Nicholson, 2010), earthquake (Korb & Nicholson, 2010), asia (Lauritzen & Spiegelhalter, 1988), sachs (Sachs et al., 2005), child (Spiegelhalter & Cowell, 1992), alarm (Beinlich et al., 1989), diabetes (Andreassen et al., 1991), and pigs (Scutari, 2010). The graphs have been downloaded from the BnLearn website[5]. For the small graphs, we have used a dataset size of 50k observational samples and 512 samples per intervention. This is a larger dataset size than for the synthetic graph because many edges in the real-world graphs have very small causal effects that cannot be recovered from limited data, and the goal of the experiment was to show that the convergence conditions also hold on real-world graphs. Hence, we need more observational and interventional samples. The results with a smaller dataset size, i.e. 5k observational and 200 interventional samples as before, are shown in Table 9. For the large graphs, we follow the dataset size for the scalability experiments (see Section C.2).

**Hyperparameters**    We reuse most of the hyperparameters of the previous experiments. For all graphs less than 100 nodes, we use the hyperparameters of Appendix C.1, *i.e.* the synthetic graphs of

---

[5]https://www.bnlearn.com/bnrepository/

Table 9: Results on graphs from the BnLearn library measured in structural hamming distance (lower is better). Results are averaged over 5 seeds with standard deviations.

| Dataset | cancer (5 nodes) | earthquake (5 nodes) | asia (8 nodes) | sachs (11 nodes) | child (20 nodes) | alarm (37 nodes) | diabetes (413 nodes) | pigs (441 nodes) |
|---|---|---|---|---|---|---|---|---|
| SDI | 3.0 ($\pm$0.0) | 0.4 ($\pm$0.5) | 4.0 ($\pm$0.0) | 7.0 ($\pm$0.0) | 11.2 ($\pm$0.4) | 24.4 ($\pm$1.7) | 422.4 ($\pm$8.7) | 18.0 ($\pm$1.6) |
| DCDI | 4.0 ($\pm$0.0) | 2.0 ($\pm$0.0) | 5.0 ($\pm$0.0) | 5.4 ($\pm$2.1) | 8.4 ($\pm$0.7) | 30.0 ($\pm$4.2) | - | - |
| ENCO | **0.0** ($\pm$0.0) | **0.0** ($\pm$0.0) | **0.0** ($\pm$0.0) | **0.0** ($\pm$0.0) | **0.0** ($\pm$0.0) | **1.0** ($\pm$0.0) | **2.0** ($\pm$0.0) | **0.0** ($\pm$0.0) |

Table 10: Results on graphs from the BnLearn library measured in structural hamming distance (lower is better), using 5k observational and 200 interventional samples.

| Dataset | cancer (5 nodes) | earthquake (5 nodes) | asia (8 nodes) | sachs (11 nodes) | child (20 nodes) | alarm (37 nodes) |
|---|---|---|---|---|---|---|
| SDI | 3.0 ($\pm$0.0) | **0.4** ($\pm$0.5) | 4.6 ($\pm$0.5) | 8.4 ($\pm$0.5) | 12.4 ($\pm$0.9) | 26.6 ($\pm$1.1) |
| DCDI | 4.0 ($\pm$0.0) | 2.0 ($\pm$0.0) | 4.4 ($\pm$0.7) | 7.2 ($\pm$2.4) | 9.8 ($\pm$0.6) | 31.4 ($\pm$0.7) |
| ENCO | **1.2** ($\pm$0.4) | **0.4** ($\pm$0.9) | **1.4** ($\pm$0.5) | **0.4** ($\pm$0.5) | **0.8** ($\pm$1.1) | **11.4** ($\pm$1.8) |

25 nodes. For all graphs larger than 100 nodes, we use the hyperparameters of Appendix C.2, *i.e.* the large-scale graphs. One exception is that we allow the fine-tuning of the regularizer parameter for both sets. For ENCO, we used a slightly smaller regularizer, $\lambda_{\text{sparse}} = 0.002$, for the small graphs, and a larger one, $\lambda_{\text{sparse}} = 0.02$, for the large graphs. Due to the large amount of deterministic variables, ENCO tends to predict more false positives in the beginning before removing them one by one. For SDI, we also found a smaller regularizer, $\lambda_{\text{sparse}} = 0.01$, to work best for the small graphs. However, in line with the results of Ke et al. (2019), SDI was not able to detect all edges. Even lower regularizers showed to perform considerably worse on the child dataset, while minor improvements were made on the small graphs. Hence, we settled for $\lambda_{\text{sparse}} = 0.01$. In terms of run time, both methods used 100 epochs for the small graphs and 50 for the large graphs.

**Results** The results including standard deviations can be found in Table 9. The low standard deviation for ENCO shows that the approach is stable across seeds, even for large graphs. SDI has a zero standard deviation for a few graphs. In those cases, SDI converged to the same graph across seeds, but not necessarily the correct graph. We have also applied DCDI (Brouillard et al., 2020) to the real-world datasets and report the results in Table 9 and 10. DCDI performs relatively similar to SDI, making a few more mistakes on the very small graphs ($< 10$ nodes) while being slightly better on sachs and child. Nonetheless, ENCO outperforms DCDI on all graphs. We do not report results of DCDI on the largest graphs, diabetes and pigs, because it ran out of memory for diabetes (larger number of max. categories per variable) and did not converge within the same time limitations as SDI and ENCO (see Section 4.3 for a comparison on scalability).

# D  ADDITIONAL EXPERIMENTS

In this section, we show additional experiments performed as ablation studies of ENCO. First, we discuss further experiments We then discuss the effect of using our gradient estimators proposed in Section 3.4 compared to Bengio et al. (2020). Next, we show experiments on synthetic graphs with deterministic variables violating faithfulness, and experiments on continuous data with Normalizing Flows. Finally, we discuss experiments with different causal mechanism functions for generating synthetic, conditional categorical distributions besides neural networks.

## D.1  EFFECT OF THE SAMPLE SIZE

The number of samples provided as observational and interventional data is crucial for causal structure learning methods since the more data we have, the better we can estimate the underlying causal mechanisms. To gain further insights in the effect of the sample size on ENCO and the compared baselines, we repeat the experiments of Section 4.2 with different sample sizes.

**Large sample size**  First, we use very large sample sizes to find the upper bound performance level that we can expect from each method. For this, we sample 100k observational samples per graph, and 10k samples per intervention. We observed that this is sufficient to model most conditional probabilities up to a negligible error. The results are shown in Table 11. We find that, in line with the theoretical guarantees, ENCO can reliably recover most graphs, only making 0.3 mistakes on average on the full graph. Of the baselines, only DCDI is able to recover the collider graph without errors since its edges can be independently orientated. For all other graphs, DCDI converges to acyclic graphs, but incorrectly orients some edges and predicts false positive edges, while being 8 times slower than ENCO on the same hardware. All other baselines show improved SHD scores than in Table 1 as well, but are not able to match ENCO's performance. This shows that, even in the data limit, ENCO achieves notably better results than concurrent methods.

Next, we consider situations where data is very limited. Thereby, we consider two data sample axes: observational and interventional data.

**Limited interventional data sample sizes**  We repeat the experiments of Table 1 for ENCO while limiting the sample size per intervention to 20, 50, and 100 (200 before). The observational dataset size of 5000 samples is thereby kept constant. We plot the performance for all graph structures in Figure 14. Overall, the decrease of performance with lower interventional sample size is consistent across graph structures. With only 20 samples per intervention, it becomes especially hard to reason about variables with many parents, since the variable's distribution is determined by many other parents as well. Yet, for four out of the six graphs, we obtain an SHD of less than 1 with 100 interventional samples, and less than 6 when only 20 samples are available. In conclusion, ENCO works well with little interventional data if most variables have a small parent set.

**Limited observational data sample sizes**  Similarly as above, we repeat the experiments of Table 1 for ENCO but limit the observational sample size to 1000 and 2000 (5000 before) while keeping 200 samples per interventions. Observational data is important in ENCO for learning the conditional distributions. For variables with many parents, this becomes more difficult when fewer samples are available, because the input space grows exponentially with the number of parents. Thus, we

Table 11: Repeating experiments of Table 1 with large sample sizes (10k samples per intervention, 100k observational samples). In line with the theoretical guarantees, ENCO can reliably recover five out of the six graph structures without errors.

| Graph type | bidiag | chain | collider | full | jungle | random |
|---|---|---|---|---|---|---|
| GIES | 47.4 (±5.2) | 22.3 (±3.5) | 13.3 (±3.0) | 152.7 (±12.0) | 53.9 (±8.9) | 86.1 (±12.0) |
| IGSP | 33.0 (±4.2) | 12.0 (±1.9) | 23.4 (±2.2) | 264.6 (±7.4) | 38.6 (±5.7) | 76.3 (±7.7) |
| SDI | 2.1 (±1.5) | 0.8 (±0.9) | 14.7 (±4.0) | 121.6 (±18.4) | 1.8 (±1.6) | 1.8 (±1.9) |
| DCDI | 3.7 (±1.5) | 4.0 (±1.3) | **0.0** (±0.0) | 2.8 (±2.1) | 1.2 (±1.5) | 2.2 (±1.5) |
| ENCO (Ours) | **0.0** (±0.0) | **0.0** (±0.0) | **0.0** (±0.0) | **0.3** (±0.9) | **0.0** (±0.0) | **0.0** (±0.0) |

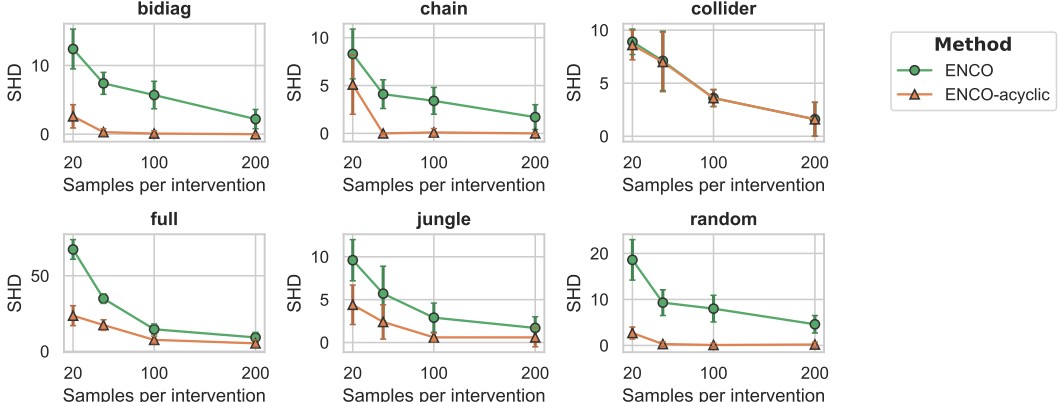

Figure 14: Results of ENCO for different graph structures under limited interventional data sample size. Note the different scale of the y-axis for the six graphs. While the general trend is the same for all graphs, *i.e.* decreasing performance with fewer samples, the order heuristic can reduce the SHD error by a considerable margin for most graphs.

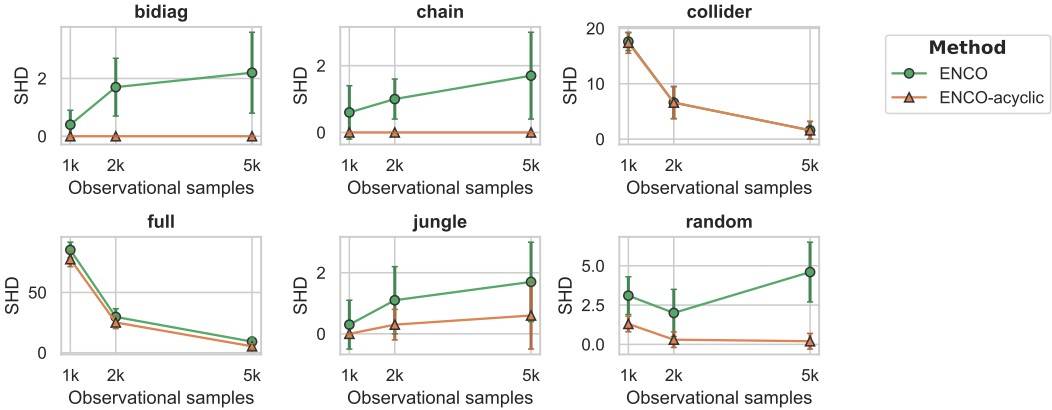

Figure 15: Results of ENCO for different graph structures under limited observational data sample size. Note the different scale of the y-axis for the six graphs. The structure learning performance remains good for sparse graphs, and suffers for graphs with larger parent sets.

would expect the collider and full graph suffer the most from having less observational data, and this is indeed the case as shown by the results in Figure 15. The results of all other graphs are less affected, although interestingly, some become even better with less observational data. For the chain $X_1 \to X_2 \to ...X_N$, for instance, we observed that the learned conditional distributions picked up spurious correlations among variables, e.g., between $X_1$ and $X_3$ when modeling $p(X_3|X_1, X_2)$ which are, in the data limit, independent given $X_2$. Since those correlations do not necessarily transfer to the interventional setting, it is easier to spot false positive edges, and we can obtain even better results than for the larger sample sizes. In conclusion, having sufficient observational data is crucial in ENCO for graphs with variables that have larger parent sets, while being less important for sparser graphs.

**Limited interventional and observational data sample sizes** Finally, we combine the smallest interventional and observational data sample sizes, and also include the results of the previously best baselines, SDI and DCDI, in Table 12. The results of ENCO show the combination of the previous two effects: graphs consisting of variables with small parent sets can still be recovered well by ENCO, while errors increase for the collider and full graph. Similar trends are observed for SDI, while DCDI showed a considerable decrease in performance for all graphs. In conclusion, ENCO still works well for graphs with smaller parent sets under a small observational and interventional data regime, and outperforms related baselines in this setting.

Table 12: Repeating experiments of Table 1 with very small sample sizes (20 samples per intervention, 1k observational samples). Despite the limited data, ENCO can recover graphs with small parent sets reasonably well, while the graphs collider and full suffer for all methods.

| Graph type | bidiag | chain | collider | full | jungle | random |
|---|---|---|---|---|---|---|
| SDI | 10.9 ($\pm$2.7) | 6.1 ($\pm$1.5) | 22.1 ($\pm$1.9) | 211.0 ($\pm$6.2) | 10.4 ($\pm$2.7) | 22.7 ($\pm$7.4) |
| DCDI | 30.0 ($\pm$4.2) | 22.0 ($\pm$1.5) | 23.2 ($\pm$1.3) | 185.2 ($\pm$4.5) | 25.8 ($\pm$2.7) | 40.2 ($\pm$8.4) |
| ENCO (Ours) | 9.7 ($\pm$3.6) | 5.6 ($\pm$1.7) | 22.7 ($\pm$2.1) | 132.6 ($\pm$8.0) | 8.1 ($\pm$2.3) | 18.4 ($\pm$4.8) |
| ENCO-acyclic (Ours) | **2.0** ($\pm$2.3) | **2.7** ($\pm$2.1) | 22.9 ($\pm$2.3) | **88.4** ($\pm$6.6) | **4.1** ($\pm$2.0) | **5.3** ($\pm$2.5) |

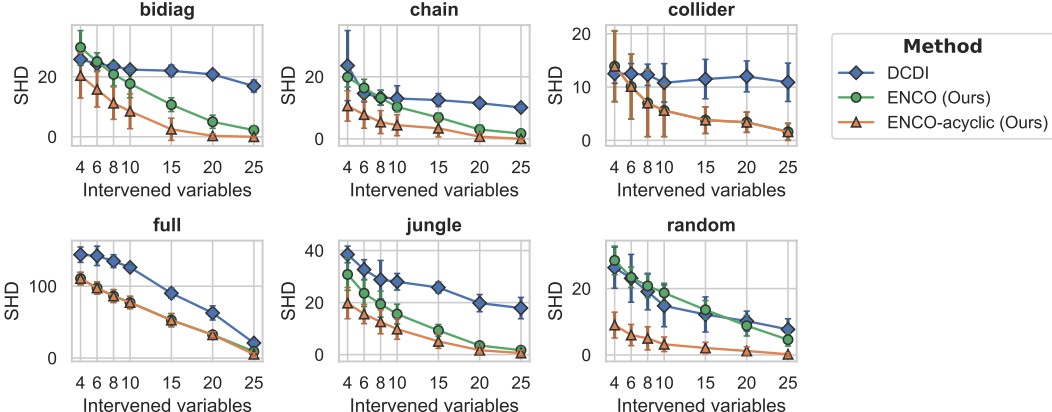

Figure 16: Results of ENCO and DCDI for different graph structures under fewer interventions provided. Note the different scale of the y-axis for the six graphs. For four out of six graphs, ENCO outperforms DCDI even for as few as 4 interventions, especially when enforcing acyclicity. The detailed numbers of the results are listed in Table 13.

## D.2 INTERVENTIONS ON FEWER VARIABLES

We have performed the experiments in Section 4.4 using fewer interventions for all six synthetic graph structures. The results are visualized in Figure 16, and presented in table-form in Table 13. From the experiments, we can see that ENCO with interventions on only 4 variables matches or outperforms DCDI with 10 interventions for 4 out of the 6 graph structures (bidiag, chain, full, jungle) when enforcing acyclicity. Especially on chain-like graphs such as jungle, ENCO achieves lower SHD scores for the same number of interventions on variables, while DCDI incorrectly orientates many edges and predicts false positive edges between children. On the graph collider, we observed a high variance for settings with very few interventions. This is because when we intervene on the collider node itself, ENCO can deduce the orientation for all edges. Finally, on the graph random, we observe that enforcing acyclicity for ENCO reduces the error a lot. This is because incorrectly orientated edges cause more false positives in this densely connected graph, which are removed with the cycles. We include a longer discussion of the limitations of ENCO on fewer interventions in Appendix B.4. Still, we conclude that, in practice, ENCO performs competitively to DCDI, even when very few interventions are provided, and scales better to more interventions.

## D.3 ABLATION STUDY ON GRADIENT ESTIMATORS

To analyze the importance of the low-variance gradient estimators in ENCO, we repeat the experiments on synthetic graph structure from Section 4.2 where the gradient estimators of ENCO have been replaced by those from Bengio et al. (2020). The results are shown in Table 14. Overall, the scores are very similar with minor differences for the graphs full and bidiag. In comparisons to the learning curves in Figure 11, the curves with the gradient estimator of Bengio et al. (2020) are more noisy, with recall and precision jumping up and down. While ENCO easily converged early to the correct graphs for all graph types, this model often required the full 30 iterations to reach the optimal recovery.

Table 13: Detailed results of the experiments with fewer interventions. See Figure 16 for a visualization and discussion.

| Graph type | bidiag | chain | collider | full | jungle | random |
|---|---|---|---|---|---|---|
| DCDI 4 vars | 25.8 ($\pm$2.0) | 23.6 ($\pm$11.3) | 12.5 ($\pm$1.8) | 143.8 ($\pm$10.7) | 38.5 ($\pm$3.2) | 26.3 ($\pm$6.2) |
| DCDI 6 vars | 24.3 ($\pm$2.2) | 14.6 ($\pm$2.7) | 12.5 ($\pm$1.9) | 142.2 ($\pm$13.5) | 32.7 ($\pm$3.8) | 23.1 ($\pm$7.2) |
| DCDI 8 vars | 23.5 ($\pm$1.4) | 13.3 ($\pm$2.4) | 12.3 ($\pm$2.0) | 134.8 ($\pm$8.9) | 28.8 ($\pm$7.4) | 19.1 ($\pm$5.5) |
| DCDI 10 vars | 22.4 ($\pm$1.1) | 13.0 ($\pm$4.1) | 10.8 ($\pm$3.6) | 126.2 ($\pm$4.2) | 28.0 ($\pm$3.2) | 14.8 ($\pm$6.3) |
| DCDI 15 vars | 22.0 ($\pm$1.9) | 12.5 ($\pm$2.1) | 11.5 ($\pm$3.7) | 90.2 ($\pm$7.1) | 25.8 ($\pm$2.1) | 12.2 ($\pm$5.3) |
| DCDI 20 vars | 20.8 ($\pm$1.4) | 11.5 ($\pm$1.3) | 12.0 ($\pm$2.9) | 62.8 ($\pm$9.8) | 19.8 ($\pm$3.3) | 10.2 ($\pm$3.0) |
| DCDI 25 vars | 16.9 ($\pm$2.0) | 10.1 ($\pm$1.1) | 10.9 ($\pm$3.6) | 21.0 ($\pm$4.8) | 17.9 ($\pm$4.1) | 7.7 ($\pm$3.2) |
| ENCO 4 vars | 29.8 ($\pm$5.6) | 19.9 ($\pm$3.2) | 13.9 ($\pm$6.6) | 110.6 ($\pm$8.6) | 30.8 ($\pm$6.1) | 28.5 ($\pm$4.3) |
| ENCO 6 vars | 25.0 ($\pm$3.0) | 16.4 ($\pm$2.8) | 10.1 ($\pm$6.1) | 97.5 ($\pm$8.5) | 23.6 ($\pm$5.1) | 23.4 ($\pm$3.1) |
| ENCO 8 vars | 20.8 ($\pm$3.8) | 13.2 ($\pm$2.4) | 7.0 ($\pm$6.3) | 86.3 ($\pm$8.9) | 19.4 ($\pm$5.0) | 20.8 ($\pm$3.5) |
| ENCO 10 vars | 17.7 ($\pm$4.7) | 10.3 ($\pm$1.6) | 5.6 ($\pm$4.9) | 77.3 ($\pm$8.5) | 15.6 ($\pm$3.8) | 18.7 ($\pm$2.9) |
| ENCO 15 vars | 10.7 ($\pm$2.3) | 6.9 ($\pm$1.5) | 3.8 ($\pm$2.5) | 52.8 ($\pm$9.1) | 9.3 ($\pm$2.3) | 13.6 ($\pm$2.8) |
| ENCO 20 vars | 5.0 ($\pm$2.2) | 3.0 ($\pm$1.5) | 3.4 ($\pm$1.9) | 32.4 ($\pm$4.2) | 3.5 ($\pm$1.1) | 8.8 ($\pm$3.1) |
| ENCO 25 vars | 2.2 ($\pm$1.4) | 1.7 ($\pm$1.3) | 1.6 ($\pm$1.6) | 9.2 ($\pm$3.4) | 1.7 ($\pm$1.3) | 4.6 ($\pm$1.9) |
| ENCO-acyclic 4 vars | 20.4 ($\pm$7.5) | 10.6 ($\pm$4.9) | 13.9 ($\pm$6.7) | 110.6 ($\pm$8.6) | 19.8 ($\pm$6.0) | 9.0 ($\pm$3.9) |
| ENCO-acyclic 6 vars | 15.8 ($\pm$5.9) | 7.8 ($\pm$4.4) | 10.1 ($\pm$6.1) | 97.5 ($\pm$8.5) | 15.6 ($\pm$3.7) | 6.0 ($\pm$3.2) |
| ENCO-acyclic 8 vars | 11.2 ($\pm$5.4) | 5.4 ($\pm$3.7) | 7.0 ($\pm$6.3) | 86.3 ($\pm$8.9) | 12.6 ($\pm$4.6) | 5.0 ($\pm$3.5) |
| ENCO-acyclic 10 vars | 8.5 ($\pm$5.8) | 4.4 ($\pm$3.4) | 5.6 ($\pm$4.9) | 77.3 ($\pm$8.5) | 9.8 ($\pm$3.9) | 3.2 ($\pm$2.2) |
| ENCO-acyclic 15 vars | 2.5 ($\pm$3.7) | 3.4 ($\pm$2.8) | 3.8 ($\pm$2.5) | 52.8 ($\pm$9.1) | 5.1 ($\pm$2.7) | 2.1 ($\pm$1.7) |
| ENCO-acyclic 20 vars | 0.3 ($\pm$0.7) | 0.7 ($\pm$1.2) | 3.4 ($\pm$1.9) | 32.4 ($\pm$4.2) | 1.7 ($\pm$1.5) | 1.2 ($\pm$1.2) |
| ENCO-acyclic 25 vars | 0.0 ($\pm$0.0) | 0.0 ($\pm$0.0) | 1.6 ($\pm$1.6) | 5.3 ($\pm$2.3) | 0.6 ($\pm$1.1) | 0.2 ($\pm$0.5) |

Table 14: Extension of Table 11 with ablation study of using Bengio et al. (2020) gradients with ENCO.

| Graph type | bidiag | chain | collider | full | jungle | random |
|---|---|---|---|---|---|---|
| ENCO (Ours) | | | | | | |
| - Bengio et al. (2020) grads | 0.1 ($\pm$0.4) | **0.0** ($\pm$0.0) | **0.0** ($\pm$0.0) | 1.9 ($\pm$1.5) | **0.0** ($\pm$0.0) | **0.0** ($\pm$0.0) |
| - Our gradient estimator | **0.0** ($\pm$0.0) | **0.0** ($\pm$0.0) | **0.0** ($\pm$0.0) | **0.3** ($\pm$0.9) | **0.0** ($\pm$0.0) | **0.0** ($\pm$0.0) |

The difference between the two gradient estimators becomes more apparent on large graphs. We repeated the experiments of Section 4.3 on the graphs with 100 nodes using the gradient estimator of Bengio et al. (2020). Within the 30 epochs, the model obtained an SHD of 15.4 on average over 5 experiments, which is considerably higher than ENCO with the proposed gradient estimators (0.0). Still, this is only half of the errors that SDI (Ke et al., 2019) with the same gradient estimator achieved. Hence, we can conclude that the proposed gradient estimators are beneficial for ENCO but not strictly necessary for small graphs. For large graphs, the low variance of the estimator becomes much more important.

## D.4 Deterministic variables

In contrast to algorithms working on observational data, ENCO does not strictly require the faithfulness assumption. Hence, we can apply ENCO to graphs with deterministic variables. Deterministic variables have a distribution that is defined by a one-to-one mapping of its parents' inputs to an output value. In other words, we have the following distribution:

$$p(X_i|\text{pa}(X_i)) = \mathbb{1}\left[X_i = f(\text{pa}(X_i))\right] \tag{47}$$

where $f$ is an arbitrary function. The difficulty of deterministic variables is that a variable $X_i$ can be fully replaced by its parents $\text{pa}(X_i)$ in any conditional distribution. The only way we can identify deterministic variables is from interventional data, where an intervention on $X_i$ breaks the dependency to its parents.

We have already tested ENCO on deterministic variables in the context of the real-world inspired graphs of Section 4.6. To have a more detailed analysis, we created synthetic graphs following the random graph setup with an edge probability of 0.1 and an average of two parents, and maximum of three parents. An example graph is shown in Figure 17. All variables except the leaf nodes have deterministic distributions, where the function $f(\text{pa}(X_i))$ is randomly created by picking a

Table 15: Experiments on graphs with deterministic variables. The performance over 10 experiments is reported in terms of SHD with standard deviation in brackets. ENCO can recover most of the graphs with less than two errors.

| Graph type | deterministic |
|---|---|
| SDI (Ke et al., 2019) | 20.6 ($\pm 3.8$) |
| ENCO (Ours) | **1.4** ($\pm 1.3$) |

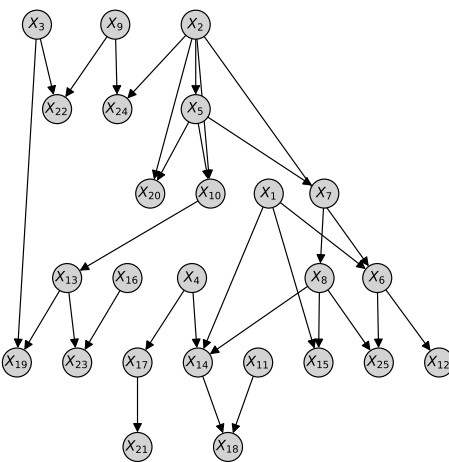

Figure 17: Example graph for the deterministic setup. We use the `random` setup with edge probability 0.1 and limit number of parents to 3. All variables except the leaf nodes have deterministic distributions.

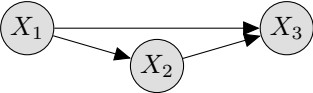

Figure 18: Example graph for cancelling paths $X_1 \rightarrow X_2 \rightarrow X_3$ and $X_1 \rightarrow X_3$. This can, for instance, occur when the conditional distribution of $X_2$ is a Dirac delta around $X_1$: $p(X_2|X_1) = \delta[X_2 = X_1]$.

random output category for any pair of input values. We create 10 such graphs and use the same hyperparameter setup as for the synthetic graphs, except that we increase the sparsity regularizer to $\lambda_{\text{sparse}} = 0.02$. We report the results in Table 15. In line with the results on the real-world graphs, ENCO is able to recover most graphs with less than two errors. As a baseline, we apply SDI (Ke et al., 2019) and see a significant higher error rate. The method predicts many false positives, including two-variable loops, but was also missing out some true positive edges. We conclude that ENCO also works well on deterministic graphs.

**Cancellation of paths** Besides deterministic nodes, a common example for faithfulness violation is the cancellation of two paths. For instance, consider the causal graph with the three variables $X_1, X_2, X_3$ shown in Figure 18, and the conditional distribution $p(X_2|X_1) = \delta[X_2 = X_1]$. In this case, the two paths $X_1 \rightarrow X_2 \rightarrow X_3$ and $X_1 \rightarrow X_3$ cancel each other, i.e. $X_3$ is independent of $X_2$ when conditioned on $X_1$, and independent of $X_1$ when conditioned on $X_2$. This implies that only one of the two graphs is necessary for describing the relations. Yet, ENCO can find the edge $X_1 \rightarrow X_3$ by observing interventions on $X_2$, since in this case, $X_1 \perp\!\!\!\perp X_2$ and $X_1 \not\!\perp\!\!\!\perp X_3|X_2$. The remaining edges can be learned in the same manner. We also emperically verify this by running ENCO on the graph structure of Figure 18 with the three variables being binary. We set $p(X_2|X_1) = \delta[X_2 = X_1]$ for

Table 16: Experiments on graph with continuous data from Brouillard et al. (2020). The suffix "*-G*" denotes that the neural networks model a Gaussian density, and "*-DSF*" a two-layer deep sigmoidal flow. ENCO outperforms all baselines in this scenario, verifying that ENCO also works on continuous data well.

| Graph type | Linear | Nonlinear with additive noise | Nonlinear with non-additive noise |
|---|---|---|---|
| GIES | 0.6 ($\pm$1.3) | 9.1 ($\pm$5.3) | 4.4 ($\pm$6.1) |
| IGSP (best) | 1.9 ($\pm$1.8) | 5.3 ($\pm$3.0) | 4.1 ($\pm$2.8) |
| DCDI-G | 1.3 ($\pm$1.9) | 5.2 ($\pm$7.5) | 2.3 ($\pm$3.6) |
| DCDI-DSF | 0.9 ($\pm$1.3) | 4.2 ($\pm$5.6) | 7.0 ($\pm$10.7) |
| GES + Orientating | 2.5 ($\pm$4.3) | 12.5 ($\pm$9.2) | 7.9 ($\pm$7.4) |
| ENCO-G (ours) | **0.0** ($\pm$0.0) | **0.2** ($\pm$0.4) | **0.1** ($\pm$0.3) |
| ENCO-DSF (ours) | 0.3 ($\pm$0.7) | 1.4 ($\pm$1.4) | 1.2 ($\pm$1.5) |

canceling the two paths, and the remaining distributions are randomly initialized. ENCO reconstructs the graph without errors, showing that it also works in practice.

### D.5 CONTINUOUS DATA

We verify that ENCO works just as well with continuous data by performing the experiments on datasets from Brouillard et al. (2020) that contained interventions on all variables. In these datasets, the graphs consist of 10 variables with an average of one edge per variable, and deploy three different causal mechanisms: linear, nonlinear additive noise models, and nonlinear models with non-additive noise using neural networks. The datasets contain 909 observational samples and 909 samples per intervention. All results of GIES, IGSP, and DCDI have been taken from Brouillard et al. (2020) (Appendix C.7, Table 22-24). We follow the setup of Brouillard et al. (2020) and compare two different neural network setups. First, we use MLPs that model a Gaussian density by predicting a mean and variance variable (denoted by suffix *G*). The second setup uses normalizing flows, more specifically a two-layer deep sigmoidal flow (Huang et al., 2018), which is flexible enough to model more complex distributions (denoted by suffix *DSF*). The rest of the experimental setup in ENCO is identical to the categorical case.

Results are shown in Table 16, and the observations are the same as with categorical data. ENCO outperforms all other methods in all settings, especially for the more complex distributions. The higher error rate for the DSF setup is mostly due to overfitting of the flow models. We conclude that ENCO works as accurately for both continuous and categorical data.

### D.6 SKELETON LEARNING WITH OBSERVATIONAL BASELINE

To show the benefit of learning a graph from observational and interventional data jointly, we compare ENCO to a simple observational baseline. This baseline first learns the skeleton of the graph by applying greedy equivalence search (GES) (Chickering, 2002) on the observational data. Then, for each interventional dataset, we apply GES as well and use those skeletons to orientate the edges of the original one. This can be done by checking for each undirected edge $X - Y$ whether $X \rightarrow Y$ is in the skeleton of interventions on $X$ or not. As a reference implementation of GES, we have used the one provided in the Causal Discovery Toolbox (Kalainathan et al., 2020).

The results on continuous data are shown in Table 16. Since GES assumes linear mechanisms and gaussianity of the data, it is unsurprising that it performs better on the linear Gaussian dataset than on the non-linear datasets. However, on all the three datasets, it constitutes the lowest performance compared to the other methods, including ENCO. This highlights the benefits of incorporating interventional data in the learning of the skeleton and graph structure. To gain further insights in comparison to the constraint-based baseline, we repeat the experiments with smaller sample sizes. The original dataset has 909 samples for observational data and per intervention, and we sub-sample 500 and 100 of those respectively for simulating smaller dataset sizes. The results of those experiments can be found in Table 17. It is apparent that the results of GES on the linear dataset get considerably worse with fewer data samples being available, while ENCO-G is able to reconstruct most graphs still without errors. Especially for the small dataset of 100 samples, we noticed that the skeletons

Table 17: Experiments on graph with continuous data from Brouillard et al. (2020) with smaller sample sizes for both observational and interventional datasets (in brackets). ENCO shows to perform much better in smaller sample sizes than a skeleton+orientation method, underlining the benefit of learning the whole graph from observational and interventional data jointly.

| Graph type | Linear Gaussian | Nonlinear with additive noise | Nonlinear with non-additive noise |
|---|---|---|---|
| GES + Orientating (909) | 2.5 ($\pm$4.3) | 12.5 ($\pm$9.2) | 7.9 ($\pm$7.4) |
| ENCO-G (ours) (909) | **0.0** ($\pm$0.0) | **0.2** ($\pm$0.4) | **0.1** ($\pm$0.3) |
| GES + Orientating (500) | 3.2 ($\pm$4.3) | 12.0 ($\pm$9.2) | 9.2 ($\pm$7.5) |
| ENCO-G (ours) (500) | **0.1** ($\pm$0.3) | **0.5** ($\pm$0.7) | **0.0** ($\pm$0.0) |
| GES + Orientating (100) | 6.1 ($\pm$5.3) | 10.2 ($\pm$6.7) | 9.0 ($\pm$6.5) |
| ENCO-G (ours) (100) | **0.2** ($\pm$0.4) | **0.9** ($\pm$1.1) | **1.3** ($\pm$0.8) |

Table 18: Comparing structure learning methods in terms of structural hamming distance (SHD) on common graph structures (lower is better), averaged over 25 graphs each. ENCO outperforms all baselines, and by enforcing acyclicity after training, can recover most graphs with minimal errors.

| Graph type | bidiag | chain | collider | full | jungle | random |
|---|---|---|---|---|---|---|
| GIES | 33.6 ($\pm$7.5) | 17.5 ($\pm$7.3) | 24.0 ($\pm$2.9) | 216.5 ($\pm$15.2) | 33.1 ($\pm$2.9) | 57.5 ($\pm$14.2) |
| IGSP | 32.7 ($\pm$5.1) | 14.6 ($\pm$2.3) | 23.7 ($\pm$2.3) | 253.8 ($\pm$12.6) | 35.9 ($\pm$5.2) | 65.4 ($\pm$8.0) |
| SDI | 9.0 ($\pm$2.6) | 3.9 ($\pm$2.0) | 16.1 ($\pm$2.4) | 153.9 ($\pm$10.3) | 6.9 ($\pm$2.3) | 10.8 ($\pm$3.9) |
| DCDI | 16.9 ($\pm$2.0) | 10.1 ($\pm$1.1) | 10.9 ($\pm$3.6) | 21.0 ($\pm$4.8) | 17.9 ($\pm$4.1) | 7.7 ($\pm$3.2) |
| GES + Orientating | 14.8 ($\pm$2.6) | 0.5 ($\pm$0.7) | 20.8 ($\pm$2.4) | 282.8 ($\pm$4.2) | 14.7 ($\pm$3.1) | 60.1 ($\pm$8.9) |
| ENCO (ours) | 2.2 ($\pm$1.4) | 1.7 ($\pm$1.3) | **1.6** ($\pm$1.6) | 9.2 ($\pm$3.4) | 1.7 ($\pm$1.3) | 4.6 ($\pm$1.9) |
| ENCO-acyclic (ours) | **0.0** ($\pm$0.0) | **0.0** ($\pm$0.0) | **1.6** ($\pm$1.6) | **5.3** ($\pm$2.3) | **0.6** ($\pm$1.1) | **0.2** ($\pm$0.5) |

Table 19: Experiments with a different data simulator, introducing independence among parents for each variable. Similar to the neural-based synthetic data, ENCO recovers most graphs with a minor error rate, outperforming other baselines.

| Graph type | bidiag | chain | collider | full | jungle | random |
|---|---|---|---|---|---|---|
| GIES | 30.7 ($\pm$3.1) | 16.9 ($\pm$2.4) | 18.6 ($\pm$2.5) | 238.6 ($\pm$4.0) | 30.1 ($\pm$3.5) | 110.0 ($\pm$11.8) |
| IGSP | 27.0 ($\pm$4.2) | 14.0 ($\pm$2.8) | 25.0 ($\pm$1.4) | 259.5 ($\pm$3.5) | 24.0 ($\pm$7.1) | 112.5 ($\pm$4.9) |
| SDI | 12.6 ($\pm$2.7) | 7.6 ($\pm$2.6) | 2.8 ($\pm$1.6) | 99.0 ($\pm$7.5) | 14.8 ($\pm$3.2) | 36.7 ($\pm$4.7) |
| DCDI | 15.7 ($\pm$1.9) | 8.4 ($\pm$1.3) | 4.7 ($\pm$2.5) | 25.3 ($\pm$6.2) | 18.9 ($\pm$3.7) | 9.8 ($\pm$3.6) |
| ENCO (ours) | 0.5 ($\pm$0.6) | 0.4 ($\pm$0.6) | 0.9 ($\pm$0.8) | 1.0 ($\pm$1.2) | 1.2 ($\pm$1.2) | 1.9 ($\pm$1.8) |
| ENCO-acyclic (ours) | **0.0** ($\pm$0.0) | **0.0** ($\pm$0.0) | **0.8** ($\pm$0.8) | **0.2** ($\pm$0.4) | **0.3** ($\pm$0.6) | **0.1** ($\pm$0.3) |

found by GES on observational data already contained couple of mistakes. This shows that for small datasets, observational data alone might not be sufficient to find the correct skeleton while by jointly learning from observational and interventional data, we can yet find the graph up to minor errors.

Further, we also apply GES on the categorical data with an additional hyperparameter search over the penalty discount. The results in Table 18 give a similar conclusion as on the continuous data. While the baseline attains good scores for chains, it makes considerably more errors on all other graph structures than ENCO. This shows that ENCO is much more robust by jointly learning from observational and interventional data.

## D.7 Non-neural based data simulators

Using neural networks to generate the simulated data might give SDI, DCDI and ENCO an advantage in our comparisons since they rely on similar neural networks to model the distribution. To verify that ENCO works for other simulated data similarly well, we run experiments on categorical data with other function forms for the causal mechanisms instead of neural networks. Since there is no straightforward way of defining 'linear' mechanisms for categorical data, we instead express a conditional distribution as a product of independent, single conditionals:

$$p(X_i|\text{pa}(X_i)) = \frac{\prod_{X_j \in \text{pa}(X_i)} p(X_i|X_j)}{\sum_{\tilde{x}_i} \prod_{X_j \in \text{pa}(X_i)} p(X_i = \tilde{x}_i|X_j)} \tag{48}$$

with $p(X_i|X_j) = \frac{\exp(\alpha_{X_i,X_j})}{\sum_{X_i} \exp(\alpha_{X_i,X_j})}, \alpha_{\cdot,\cdot} \sim \mathcal{N}(0,2)$. Hence, the effect of each variable in the parent set is independent of all others, similar to linear functions in the continuous case. The individual probability densities represent a softmax distribution over variables sampled from a Gaussian distribution.

We apply GIES, IGSP, DCDI, SDI and ENCO to the same set of synthetic graph structures with these new causal mechanisms. Similar to the previous experiments, we provide 200 samples per intervention and 5k observational samples to the algorithms, and repeat the experiments with 25 independently sampled graphs. The results in Table 19 give the same conclusion as the experiments on neural-based causal mechanisms, namely that ENCO outperforms all baselines. Most methods experience a decrease in performance since the average causal effect of each parent is lower than in the neural case where more complex interactions between parents can be modeled. Still, ENCO only shows minor decreases, having less than one mistake on average for every graph structure when applying the orientation heuristic.

