# OpenReview forum: "Efficient Neural Causal Discovery without Acyclicity Constraints"
_ICLR.cc/2022/Conference — ICLR 2022 Poster_

### Official Review · Reviewer_qKQN · 2021-10-31

**Correctness:** 3
**Technical Novelty And Significance:** 3
**Empirical Novelty And Significance:** 3
**Recommendation:** 6
**Confidence:** 3

**Main Review:**

This paper is well-written. The claims seem sound and the experimental results are convincing. The assumptions are clearly described, and the limitations are also discussed. Also, the authors extend their method to detect hidden confounders, which is interesting and important.

My only question is about the distribution fitting stage. The authors build a neural network for each variable to model the conditional distributions of that variable, conditioning on any variable set. That is, given a variable $X_i$ and a conditioning set ${\\bf X}_I$, where the indexing set $I\\subset \\{1, 2, \\cdots, N\\}\\setminus\\{i\\}$, the authors define a mask ${\\bf M}\\in \\{0, 1\\}^{N}$ such that ${\\bf M}(j)=1$ if and only if $j=i$ or $j\\in I$, and model $P(X_i \\mid {\\bf X}_I)$ by the NN $f({\\bf M}\\circ {\\bf X}; {\\phi_i})$. I am wondering how this objective relates to the loss function given in Equation (1).


**Summary Of The Paper:**

This paper proposes a new gradient-based method to learn causal DAGs from observational and interventional data. It models a probability for every possible directed edge between pairs of variables and searches for the graph which generalizes best from observational to interventional data.

**Summary Of The Review:**

Overall, this paper is well-written. The claims are supported by rigorous theoretical analysis and extensive empirical studies. It seems that there is only a minor issue in Section 3.2. I think this is a good submission if the authors could clarify this point.

---

> ### Author Response · Authors · 2021-11-16
> **Response to Reviewer qKQN**
>
> We thank the reviewer for the positive feedback and interest in the work. Please find our answer to the clarification question below.
>
> __The authors build a neural network for each variable to model the conditional distributions of that variable, conditioning on any variable set. [...] I am wondering how this objective relates to the loss function given in Equation (1).__
>
> The loss function in Equation (1) is a maximum likelihood objective, or more specifically here, a minimization of the negative log-probability of the samples obtained from the observational data.
> By maximizing the probability, the network $f\_{\phi\_i}$ is optimized to follow the observational distribution for variable $X_i$ conditioned on a masked-out set of parents $\mathbf{X}\_{I}$.
> Note that in this step, the mask is not optimized, but only the neural network is trained to fit the observational distribution.
> With a powerful enough neural network, $f_{\phi_i}$ can model the observational distribution for any possible parent set using the loss function in Equation (1).
> Thus, to summarize, by minimizing the loss function in Equation (1) with respect to the parameters $\phi_i$, the network learns to predict the observational distribution of the variable $X_i$ given a masked parent set $\mathbf{X}_I$, i.e. the objective of interest.

---

> > ### Comment · Reviewer_qKQN · 2021-11-28
> > **Thank you for your reply**
> >
> > I would like to thank the authors for the clarification, and I have no more questions for the authors.

---

### Official Review · Reviewer_rJAC · 2021-11-01

**Correctness:** 3
**Technical Novelty And Significance:** 3
**Empirical Novelty And Significance:** 2
**Recommendation:** 6
**Confidence:** 4

**Main Review:**

I find the proposal very interesting, especially in light of strong empirical results. The paper is well written but in my opinion, many of the claims were not sufficiently justified.

1. A starting question I have is why acyclicity is necessary to enforce if interventions are available on all variables. An approach such as ICP [Peters et al, 2016] applied to all variables separately should recover the underlying graph and does not require acyclicity.

2. It is not clear that the authors ensure that the recovered graph is acyclic. It is mentioned before section 3.3 that "... acyclicity can be enforced as a post-processing step". What is this step? Where is it discussed?

3. Interventions on all variables will obviously never be the case in practice, how do the authors think the proposed approach can be used in practice. What does the method converge to if interventions are not available on all variables? Can we characterize equivalence classes?

4. Theorem 3.1 assumes access to conditional distributions if my reading is correct, is it plausible to expect that there will be no local optima with finite samples? With overparameterized models, this seems unlikely.

5. After Theorem 3.1 the authors say "we can guarantee to converge to the true causal graph if, for every edge Xi → Xj in the ground truth, the variables are not conditionally independent in the interventional data of Xi". Isn't this a faithfulness condition, i.e. that a conditional independence implies a corresponding separation in the graph?

Minor.
Score-based methods are defined by the optimization of a score rather than by a search strategy in the space of DAGs which makes me uneasy about the new category of continuous-optimization methods.

**Summary Of The Paper:**

The problem of score-based structure learning using observational data as well as a set of interventional datasets is increasingly relevant in domains where experiments are readily available. The authors propose a new parameterization of the DAG underlying the data and score showing, provided that experiments are available on all observed variables and the system be Markovian, that the true causal graph can be recovered.

**Summary Of The Review:**

Interesting proposal if some of the claims can be better justified and if we can bring the algorithm closer to practice. In its current formulation, both interventions on all variables and unlimited sample sizes to guarantee convergence are not plausible in my opinion.

---

> ### Author Response · Authors · 2021-11-16
> **Response to Reviewer rJAC [1/2]**
>
> We thank the reviewer for the constructive and helpful feedback. Please find our answers below.
>
> __A starting question I have is why acyclicity is necessary to enforce if interventions are available on all variables. An approach such as ICP [Peters et al, 2016] applied to all variables separately should recover the underlying graph and does not require acyclicity.__
>
> While constraint-based methods such as ICP do not require an explicit acyclicity constraint in such a setting, acyclicity needs to be enforced in score-based methods, and more specifically continuous-optimization score-based methods, when the scoring function cannot always guarantee to have the correct, acyclic graph as the minimum. This also holds for the interventional case, since we look for a graph that, on average over interventions, models the highest likelihood of the data (e.g. see left part of the objective in Equation 2). As an example for why we need to enforce acyclicity, consider a graph with $N$ variables where $X_1$ is a root node and $X_2$ has $X_1$ as its only parent (other variables might be connected with having $X_2$ as parent). Under any intervention excluding $X_1$ or $X_2$, having the edge $X_2\to X_1$ in the graph will lead to a better estimate of $X_1$ than excluding the edge, because $X_1$ and $X_2$ are not independent. When the number of variables $N$ goes to infinity, each individual intervention obtains less and less weight in the score function, hence potentially leading to the graphs with the cycle $X_1\leftrightarrow X_2$ obtain a better score than acyclic graphs. The same argument can be applied to larger cycles.
>
>
> __It is mentioned before section 3.3 that "... acyclicity can be enforced as a post-processing step". What is this step? Where is it discussed?__
>
> The post-processing step is discussed in Section 4.2, second paragraph, where it is applied to the predictions of the proposed algorithm ENCO (referred to as ENCO-acyclic in Table 1). In short, we can strictly enforce acyclicity by taking the causal ordering which maximizes the product of orientation likelihoods learned by the orientation parameter $\mathbf{\theta}$. A more detailed description is given in Appendix C.1.2, paragraph 'Enforcing acyclicity', and its implementation can be found in the file causal_discovery/utils.py in the linked GitHub repository of the reproducibility statement.
>
>
> __Interventions on all variables will obviously never be the case in practice, how do the authors think the proposed approach can be used in practice. What does the method converge to if interventions are not available on all variables? Can we characterize equivalence classes?__
>
> In practice, ENCO still discovers the causal graphs accurately even when not intervening on all variables, see experiments in Section 4.4.
> Even though the convergence guarantees of Theorem 3.1 do not fully hold anymore when not intervening on all variables, many conditions still transfer, as we discuss in Appendix B.4.
> Specifically, ENCO is guaranteed to find the correct edges $X_i\to X_j$ for those variables on which we can intervene, either $X_i$ or $X_j$.
> For the rest of the variables, we identify two cases.
>
> In the first case, all remaining variables have no causal relations between them, namely they are conditionally independent of each other given the intervened variables.
> In this case, ENCO will still converge to the true graph since the provided interventions suffice for all edge orientations and irrelevant edges are removed due to independence.
>
> In the second case, there exist causal variables between the remaining variables.
> In this case, ENCO cannot guarantee to find the correct orientations of those edges anymore, especially since more local optima will occur.
> In the worst case, the model could add incorrect edges, e.g., in long chains with strong dependencies among variables without interventions.
> Still, if we can assume faithfulness in this case, we can use a skeleton as a prior over the undirected graph structure to prevent false positive edges.
>
> All that said, ENCO exhibits in Section 4.4 strong empirical gains also in settings where we can intervene only on few variables, as compared to DCDI. This brings the interesting insight that it does not only matter that a method (like DCDI) provides theoretical guarantees for the equivalence classes, but it is just as important having an efficient, practical algorithm that can get you to that optimum.

---

> > ### Author Response · Authors · 2021-11-16
> > **Response to Reviewer rJAC [2/2]**
> >
> > __Theorem 3.1 assumes access to conditional distributions if my reading is correct, is it plausible to expect that there will be no local optima with finite samples? With overparameterized models, this seems unlikely.__
> >
> > We want to clarify that the local optima in Theorem 3.1 are considered for the optimization of the graph parameters $\mathbf{\theta}$ and $\mathbf{\gamma}$, not directly to the neural networks which learn the conditional distributions. For sufficient data and large enough neural networks, local optima of those do not constitute a common issue in practice.
> >
> > In case we have a smaller set of samples available, learning the ground-truth conditional distributions with neural networks comes with the usual challenges as any few-shot training task for neural networks. We ablate and provide a detailed discussion on the adaption of Theorem 3.1 to the limited sample case in Appendix B.2.2. Next, we give a summary of the main points.
> >
> > If we train the (likely overparameterized) neural networks on the finite samples, we can ensure that the neural networks learn the distribution which fits the given samples the best. For a categorical distribution, this might correspond to the normalized count for each conditional set. For a Gaussian distribution, we would learn the mean and standard deviation functions for an input set. For those distributions, condition 1 to 3 needs to hold, with some additional conditions which were previously implied by having the true conditionals (e.g. if we intervene on a variable, it is independent of its parents). Lastly, in case we have very limited data, we can benefit a lot from having a prior/assumptions on the ground truth conditionals such as that the data is Gaussian-like, since otherwise, it becomes very difficult to come close to the true conditionals. We note, however, that in practice ENCO showed to work fine on small datasets, especially for interventional data, and outperform other methods.
> >
> >
> > __After Theorem 3.1 the authors say "we can guarantee to converge to the true causal graph if, for every edge $X_i\to X_j$ in the ground truth, the variables are not conditionally independent in the interventional data of $X_i$". Isn't this a faithfulness condition, i.e. that a conditional independence implies a corresponding separation in the graph?__
> >
> > The difference between the statement above and faithfulness is the regime of interventional vs observational data. In particular, faithfulness is, according to Pearl, 1988, defined as follows: A joint distribution $\mathbb{P}$ is said to be faithful to a graph $G$ if all and only the conditional independence relations in $\mathbb{P}$ are present in $G$. Thereby, $\mathbb{P}$ is the joint distribution of the observational data, while with interventions, we perform changes on $\mathbb{P}$. A common example of graphs that break faithfulness are those with deterministic relations. For instance, if in the graph $X_1\to X_2\to X_3$, $X_2$ can be deterministically inferred from $X_1$, we obtain that $X_3$ is conditional independent of $X_2$ given $X_1$. However, the graph implies that this statement is not true, thus making $p(X_1,X_2,X_3)$ not faithful with respect to the graph. Still, ENCO is able to recover such graphs by seeing interventions on $X_2$, which we empirically verify in Appendix D.4 with experiments on graphs with deterministic causal mechanisms.
> >
> >
> > __Score-based methods are defined by the optimization of a score rather than by a search strategy in the space of DAGs which makes me uneasy about the new category of continuous-optimization methods.__
> >
> > Thank you for pointing it out, the paragraph was indeed vague about the relation of continuous-optimization and score-based method. We have rephrased it to make clear that continuous-optimization methods are score-based methods, since they also optimize a score. We highlight them in a separate paragraph since they are closest to our work.
> >
> >
> > __References__
> >
> > [Pearl, 1988] Judea Pearl. Probabilistic Reasoning in Intelligent Systems: Networks of Plausible Inference. Morgan Kaufmann Publishers Inc., San Francisco, CA, USA, 1988. ISBN 0934613737.

---

> > > ### Comment · Reviewer_rJAC · 2021-11-18
> > > **Thank you for your clarifications**
> > >
> > > I appreciate the author's clarifications. There are two points on the answers above I'd like to get the author's feedback on.
> > >
> > > I am not fully satisfied with the answer on faithfulness because even though the authors show an example that allows to correctly infer edges from data distributions, it seems to me that what they require is simply a faithfulness condition on interventional distributions, which is still a faithfulness condition. For instance, what about the case {X -> Y, X -> Z -> Y}, such that the two paths of dependencies between X and Y cancel, which is also a common violation of faithfulness?
> > >
> > > If an important contribution of the paper is that there is no need to enforce acyclicity constraints, how to ensure acyclic graphs deserves more explanation than a couple of sentences in the experiments. This aspect remains unclear to me and I hope that the authors will describe this in more detail in a revision of their manuscript. In principle any method for learning adjacency matrices can be transformed into an acyclic graph using post-processing steps, is the approach that the authors propose novel or specific to ENCO, can it be used elsewhere?

---

> > > > ### Author Response · Authors · 2021-11-19
> > > > **Thank you for your additional questions**
> > > >
> > > > Thank you for your additional questions. Please find our answer below.
> > > >
> > > > __Clarification on faithfulness__
> > > >
> > > > We agree that the conditions 1 and 2 in Theorem 3.1 resemble a faithfulness statement on interventional distributions.
> > > > Still, there are subtle differences between the conditions and faithfulness here.
> > > > Specifically, condition 1 and 2 require that the _orientation_ of the edge can be learned from intervening on the adjacent nodes.
> > > > However, the _existence_ of the edge is learned over the aggregation of all interventions (i.e. the expectation over interventions on all variables), which can yet reveal edges between nodes that violate faithfulness.
> > > > For instance, in the suggested example of $X\to Z\to Y, X\to Y$ with the two paths of $X$ to $Y$ canceling each other out, we can find the edge $X\to Y$ by observing interventions on $Z$.
> > > > Further, this graph can be learned by ENCO because it fulfills the conditions of the Theorem 3.1 in the following way.
> > > > Condition 1 is fulfilled for $X\to Y$ since when we intervene on $X$, conditioning $Y$ on $X$ improves its estimate when $Z$ is not in the parent set ($p(Y)$ vs $p(Y|X)$), and remains unchanged when $Z$ is in the parent set ($p(Y|Z)=p(Y|Z,X)$). Thus, in both cases, the equation is greater or equals to zero.
> > > > Condition 2 holds as long as the edge probability $p(Z\to Y)$ is not exactly one. Since we are in a learning process where $p(Z\to Y)$ slowly converges to one, there will be always graph samples during the training where $Z$ is not the parent of $Y$, even when it becomes very rare in the later iterations.
> > > > Additionally, the orientation can also be learned from interventions on $Y$ since $p(X|Y)$ can only lead to a worse estimate of $X$ than $p(X)$ under this setting, ensuring that $Y\to X$ is not possible.
> > > > Finally, for condition 3, interventions on $Z$ break the independence of $X$ and $Y$ given $Z$, since $X$ does not influence $Z$ anymore and thus stays dependent on $Y$.
> > > > Thus, in conclusion, we can yet learn the correct graph structure.
> > > >
> > > > We also empirically verify this by running ENCO on the mentioned graph structure with $X,Z,Y$ being binary variables.
> > > > We set $p(Z|X)=\delta[Z=X]$ to be deterministic, hence canceling the two paths.
> > > > ENCO reconstructed the graph without errors, showing that it also works in practice.
> > > >
> > > > __Enforcing acylicity__
> > > >
> > > > Thank you for the suggestion! To bring more focus to this, we uploaded a new revised version where we moved the description of enforcing acyclicity into Section 3.4 (convergence guarantees, last paragraph).
> > > > The key aspect of ENCO in this acyclicity enforcement is that we rely on the learned orientation parameters instead of the adjacency matrix.
> > > > The differences between these two become clear when looking at the example of a chain, e.g. $X_1\to X_2\to X_3\to X_4$.
> > > > Suppose we have learned an adjacency matrix which predicts a cycle, i.e. an additional edge $X_4\to X_1$, which can happen when limited interventional data is provided.
> > > > Without further information, enforcing acyclicity in this graph would fall back to a random guess over edges, since we cannot say for sure which is the incorrect edge.
> > > > ENCO, however, also learns the orientations of ancestors to descendants like $X_1$ to $X_3$ from the interventions. Thus, for this example graph, we experienced that the probabilities over orientations are often close to the following:
> > > > $$\sigma(\mathbf{\theta})\approx\begin{bmatrix}- & 0.99 & 0.99 & 0.01\\\\0.01 & - & 0.99 & 0.99\\\\0.01 & 0.01 & - & 0.99\\\\ 0.99 & 0.01 & 0.01 & -\\\\\end{bmatrix}$$
> > > > where $\sigma(\theta_{ij})$ is the probability of the edge orientation $X_i\to X_j$.
> > > > We orient the graph by finding the node order that maximizes the product of these orientation probabilities.
> > > > For the example matrix above, the optimal order would be $X_1,X_2,X_3,X_4$ since this leads to the product: $$p([X_1,X_2,X_3,X_4])=\sigma(\theta_{12})\cdot\sigma(\theta_{13})\cdot\sigma(\theta_{14})\cdot\sigma(\theta_{23})\cdot\sigma(\theta_{24})\cdot\sigma(\theta_{34})$$
> > > > where only $\sigma(\theta_{14})$ is $0.01$, while any other permutation would include more than one of the small factors.
> > > > From that, it becomes apparent that $X_4\to X_1$ is the outlier and thus should be flipped in orientation.
> > > > Further, since our edge existence parameters $\mathbf{\gamma}$ are learned asymmetrically, i.e. $\gamma_{ij}\neq\gamma_{ji}$, we can use those to determine whether the flipped orientation $X_1\to X_4$ should have an edge or not.
> > > > These differences explain the performance improvement gained by enforcing acyclicity using this algorithm in ENCO.
> > > >
> > > > Please, let us know if you have any further questions or doubts remaining.

---

> > > > > ### Comment · Reviewer_rJAC · 2021-11-19
> > > > > **This clarifies my concerns**
> > > > >
> > > > > Thank you for taking the time to give detailed answers. I hope these discussions will serve to give more context to these issues which I find important and perhaps also allow to better contrast with related work. My score was weakly positive and my confidence was already high -- I maintain both of these opinions after factoring in our discussion.

---

### Official Review · Reviewer_Paee · 2021-11-02

**Correctness:** 3
**Technical Novelty And Significance:** 2
**Empirical Novelty And Significance:** 2
**Recommendation:** 5
**Confidence:** 3

**Main Review:**

The paper is in general well-written. The discussion on the latent confounder case and few interventions are appreciated. The strength of the work is its originality. It introduces a certain parameterization method for enforcing the acyclicity of the model without imposing acyclic constraints on the optimization problem.

Nevertheless, I do have some concerns about the work:
1. The sparsity penalty term in (2).
The graph with the maximum likelihood may not be the one with the minimum loss value of (2) because of the penalty term. The graph fitting procedure is actually not picking up the graph with the maximum likelihood but selecting a graph by jointly considering likelihood and the penalty term. So why would this be reasonable?

2. The convergence guarantee.
* It requires intervention on all variables. It would be necessary to justify the significancy of the convergence result. According to the citation (Eberhardt et al., 2005) in the paper, roughly speaking, in the worst case, it requires log2(N) + 1 experiments to determine all causal relations, where N is the number of variables.

* Moreover, in the case with intervention on fewer variables, it remains unclear that what would be the results in such case. Are they in the same Markov equivalent class as the ground-truth graph, or there can be arbitrary differences from the ground-truth graph?

* It would also be necessary to justify condition 3 in Thm. 3.1. Especially, how realistic is the condition? When can it be satisfied? And what will happen if it is not satisfied?
It would be better to make it clear whether the improper sparsity parameter can only lead to slow convergence or it can even lead to wrong results.

3. A minor suggestion. It would be better if the paper distinguishes causal discovery on observational from causal discovery on both observational and interventional data, and then introduce more about the later setting which is more related to the work.
Because when mentioning causal discovery, in common, it refers to only on the observational data, it can be misleading sometimes.
Moreover, instead of only generally mentioning the works based on observational and interventional data, it would be helpful for having an overview and a better judgment and understanding of the paper if including the important theoretical results and the related methods/conclusions in such setting.

**Summary Of The Paper:**

The paper works on causal discovery on both observational and interventional data.
It proposes a method for enforcing the acyclicity of DAGs which avoids imposing constraints on the optimization problem.
It proves that under some assumptions, when all the variables have interventional data, the resulting graph converges to the ground-truth one. The experimental results show the properties and performance of the proposed method, including the cases with latent confounders and interventions on fewer variables.

**Summary Of The Review:**

The strength of the paper is the proposed way for enforcing acyclicity and the good properties of the proposed model.
However, the convergence guarantee may require justification for its significancy and Condition 3. Moreover, the loss function in the graph fitting procedure may need a justification and discussion for the penalty term.

---

> ### Author Response · Authors · 2021-11-16
> **Response to Reviewer Paee [1/2]**
>
> We thank the reviewer for the constructive and helpful feedback. Please find our answers below.
>
> __The graph with the maximum likelihood may not be the one with the minimum loss value of (2) because of the penalty term. The graph fitting procedure is actually not picking up the graph with the maximum likelihood but selecting a graph by jointly considering likelihood and the penalty term. So why would this be reasonable?__
>
> The penalty term is required since multiple graphs can maximize the likelihood objective. The penalty term thereby guides the method to choose the sparsest graph in this set of graphs, maximizing the likelihood.
> As an intuitive example, consider the true graph being a chain of three variables, i.e. $X_1\to X_2\to X_3$. Alternatively to the chain, the fully connected graph $X_1\to X_2\to X_3, X_1\to X_3$ can model the same probability distributions since $p(X_3|X_2,X_1)=p(X_3|X_2)$.
> By introducing the penalty term to promote sparsity, the method will choose the chain $X_1\to X_2\to X_3$ over the fully connected graph.
>
> In case, however, the ground truth graph would indeed be fully connected, the method would not pick the chain in the first place.
> This is for as long as the difference in terms of log-likelihood estimates between using the chain and the fully connected graph is higher than the penalty term.
> This is especially important in the case of limited sample sizes to remove spurious correlations.
> Similar approaches have been used in previous score-based methods (Zheng et al., 2018; Ke et al., 2019; Brouillard et al., 2020) and is highly related to the significance threshold used in constraint-based methods.
>
>
> __It would be necessary to justify the significancy of the convergence result. According to the citation (Eberhardt et al., 2005) in the paper, roughly speaking, in the worst case, it requires log2(N) + 1 experiments to determine all causal relations, where N is the number of variables.__
>
> We agree and clarify that the convergence results do not show novel results in terms of theoretical identifiability, i.e. that the graph could be identified with an extensive search/ideal algorithm under these constraints. In fact, our interventional setting falls into Theorem 3.7 of Eberhardt et al. (2005), which says that $N-1$ interventions are, in the worst case, necessary. In this paper, instead, the convergence results show that the proposed algorithm, ENCO, will return the correct graph upon convergence, thus justifying the algorithm strategy. Thereby, we want to emphasize that this is a gradient-based discovery algorithm that scales to large datasets and graph sizes. While previous works have mostly focused on giving guarantees for the global optima, the algorithm might converge to a local optima, which the given convergence guarantees eliminate for ENCO.
>
>
> __In the case with intervention on fewer variables, it remains unclear that what would be the results in such case. Are they in the same Markov equivalent class as the ground-truth graph, or there can be arbitrary differences from the ground-truth graph?__
>
> When interventions on all variables are not available, the convergence guarantees of Theorem 3.1 do not fully hold anymore. However, many conditions can still be transferred. We have provided a more detailed discussion in Appendix B.4 about the convergence guarantees for this setting. In general, we can provide guarantees for finding the correct edges for those variables that interventions have been provided, since the existence and orientation of an edge $X_i\to X_j$ can be learned from interventions on either $X_i$ or $X_j$. If the remaining variables without interventions are conditionally independent of each other given the intervened parents, ENCO would converge to the true graph since all edge orientations can be found with the provided interventions, and edges between the other variables are removed due to independence between the variables. Nonetheless, when the variables without interventions have causal relations between each other, we cannot guarantee to find the correct orientations of those edges anymore, especially since more local optima will occur. In the worst case, additional edges can be predicted among those variables if incorrect orientations have been chosen, such as in long chains with strong dependencies among variables without interventions. Those could be prevented by, for instance, providing a skeleton if faithfulness is assumed. Still, the experiments in Section 4.4 show that empirically, ENCO still outperforms DCDI on most settings which has a guarantee of the global optimum of its objective being the interventional Markov equivalence.

---

> > ### Author Response · Authors · 2021-11-16
> > **Response to Reviewer Paee [2/2]**
> >
> > __It would also be necessary to justify condition 3 in Thm. 3.1. Especially, how realistic is the condition? When can it be satisfied? And what will happen if it is not satisfied?__
> >
> > Please see our general response on the convergence guarantees, specifically the paragraphs _Intuition behind conditions_ and _Violations of conditions_, for the full details. In general, condition 3 in Theorem 3.1 provides us an upper bound for the sparsity regularizer $\lambda_{\text{sparse}}$ which for almost all graphs is greater than zero. Thus, to guarantee its recovery, we only need to set the sparsity regularizer small enough. We can follow similar practice of choosing this value as other methods such as NOTEARS (Zheng et al., 2018), SDI (Ke et al., 2019), and DCDI (Brouillard et al., 2020) since all have a similar sparsity regularizer. In case the condition is violated, we might converge to a graph where some edges are missing compared to the ground truth.
> >
> >
> > __It would be better if the paper distinguishes causal discovery on observational from causal discovery on both observational and interventional data, and then introduce more about the later setting which is more related to the work.__
> >
> > Thank you for the suggestion. We have adapted the related work, in particular the introduction to Section 2.2, to make this difference clearer and discuss more about the latter aspect.
> >
> >
> > __References__
> >
> > [Eberhardt et al., 2005] Eberhardt,  F.,  Glymour, C., Scheines.  R., (2005).  On the Number of Experiments Sufficient and in the Worst Case Necessary to Identify All Causal Relations Among N Variables, in Proceedings of the 21  st Conference on Uncertainty and Artificial Intelligence,  Fahiem Bacchus and Tommi Jaakkola (editors), AUAI Press, Corvallis, Oregon, pp. 178-184.
> >
> > [Zheng et al., 2018] Zheng, X., Aragam, B., Ravikumar, P., Xing, E. P. (2018). Dags with no tears: Continuous optimization for structure learning. In Proceeding of the 32th Conference on Neural Information Processing Systems, 2018.
> >
> > [Ke et al., 2019] N.R. Ke, O. Bilaniuk, A. Goyal, S. Bauer, H. Larochelle, C. Pal, Y. Bengio. Learning neural causal models from unknown interventions. arXiv preprint arXiv:1910.01075
> >
> > [Brouillard et al., 2020] P. Brouillard, S. Lachapelle, A. Lacoste, S. Lacoste-Julien, A. Drouin. Differentiable Causal Discovery from Interventional Data. In Proceeding of the 34th Conference on Neural Information Processing Systems, 2020.

---

> > ### Comment · Reviewer_Paee · 2021-11-25
> > **Grateful for the authors' response; One point to be clarified.**
> >
> > I would thank the reviewer for the effort on the rebuttal. As the authors said, the discussion and the review process make both the authors and I have a better understanding of the paper. And the paper will indeed be improved significantly.
> >
> > All my reviews are well answered, which is very much appreciated. And I do know that the contribution of the work is for causal discovery on both observational and interventional data, which is an important problem for causal discovery. But just for making sure, *when there is only observational data, can the method guarantee the results are having the correct causal skeleton and/or in the correct Markov equivalent class.
> > In other words, can the method capture the correct (conditional) independent relationship? Furthermore, can the sparsity penalty term help in such a case?*

---

> > > ### Author Response · Authors · 2021-11-26
> > > **Thank you for your response**
> > >
> > > Thank you for your positive words and additional question! The algorithm as presented in the paper requires interventional data for learning the graph by testing the generalization of observational distributions to interventions. However, one could adjust the algorithm by using observational data only in the graph fitting stage. In this case, the sparsity regularizer will indeed help by removing edges between conditionally independent variables. Although we cannot guarantee a strict convergence to the correct Markov equivalence class then, an interesting future direction is to combine current continuous-optimization score-based methods for observational data with ENCO's parameterization and gradient estimators to achieve such while gaining the efficiency and accuracy benefits from ENCO.
> > >
> > > Please let us know if you have any further doubts that stop you from raising your initial score.

---

> > > > ### Comment · Reviewer_Paee · 2021-11-26
> > > > **Thank for the reply. No more further questions.**
> > > >
> > > > I would thank the authors again for the response to my questions. No further questions from my side.

---

### Official Review · Reviewer_H7r5 · 2021-11-02

**Correctness:** 3
**Technical Novelty And Significance:** 3
**Empirical Novelty And Significance:** 3
**Recommendation:** 8
**Confidence:** 3

**Main Review:**

Overall, the paper is well written and clearly structured. I appreciate Figure 1 which is a concise graphical overview of the method.

The method itself and how to train it is clearly explained in the sections 3.1-3.3. Regarding the convergence guarantees made in section 3.4, I am not an expert in this domain so I cannot confirm with high certainty that the proof is correct but the logical reasoning seems to be fine. I acknowledge that the authors have an example of how to check the conditions of Theorem 3.1 in appendix B.2.1. However, as far as I understand the paper, these steps can only be applied if the probabilistic model is known but this is not the case in an real world scenario. Especially condition 3 seems problematic as it sets the scale of the regularisation parameter $\lambda_{sparse}$. For large graphs, it seems to be very complicated to estimate $\lambda_{sparse}$ and, hence, there will be no guarantee that we recover the full causal graph.

The experiments are extensive and the authors proved that their method is clearly better than several baselines on different datasets. What seems strange to me is that they did only compare their method to SDI in section 4.6. They argue that it is in general the strongest baseline but in fact it is outperformed by DCDI on the sachs and child dataset according to Table 9. These results should be reported in the main paper as well, especially since they do do not need a lot of space.
For the paper to be truly outstanding, it would have been nice to see the method being applied to a real-word dataset. Although the true causal graph is not known in this setting, the plausibility of the result could be discussed in this case to evaluate the performance of the method.

**Summary Of The Paper:**

The paper introduces a method, called Efficient Neural Causal Discovery (ENCO), to determine the structure of a causal graphical model of a set of random variables, which uses both observational and interventional data.

The training procedure of the method consists of two steps: a distribution of one random variable given the others is fitted through a neural network and by performing interventions the graph is fitted using the approximate distributions. By alternating the two steps the causal graph can be recovered. The authors state the conditions when this procedure converges and prove the respective convergence theorem. Furthermore, they show how latent confounders can be detected within their framework.

The authors apply their method to several synthetic datasets, among them real-world inspired data form the Bayesian Network Repository (BnLearn), to show that they can reconstruct the causal graph for various numbers of random variables as well as find latent confounders if they are present. Based on various performance metrics they outperform competing methods, such as Structural Discovery from Interventions (SDI).

**Summary Of The Review:**

Overall, the paper is a significant contribution to the area of learning causal graphs. Besides several minor points, the article is well written and should be accepted.

---

> ### Author Response · Authors · 2021-11-16
> **Response to Reviewer H7r5**
>
> We thank the reviewer for the positive and valuable feedback. Please find our answers below.
>
> __I acknowledge that the authors have an example of how to check the conditions of Theorem 3.1 in appendix B.2.1. However, as far as I understand the paper, these steps can only be applied if the probabilistic model is known but this is not the case in an real world scenario. Especially condition 3 seems problematic as it sets the scale of the regularisation parameter $\lambda\_{\text{sparse}}$. For large graphs, it seems to be very complicated to estimate $\lambda\_{\text{sparse}}$ and, hence, there will be no guarantee that we recover the full causal graph.__
>
> Determining the exact upper bound for $\lambda_{\text{sparse}}$ would indeed require the true probabilistic model and can be expensive for larger graphs. However, we emphasize that _this condition only gives us an upper bound_ for the sparsity regularizer weight $\lambda_{\text{sparse}}$. Hence, for almost all graphs, we can fulfill the condition by choosing a small enough value for $\lambda_{\text{sparse}}$ as long as it is larger than zero.
> The value can be estimated by knowing the minimum impact that a parent might have on a child.
> Also, even for slightly larger $\lambda_{\text{sparse}}$, the algorithm would simply sparsify edges towards parent variables of little practical relevance.
>
>
> __What seems strange to me is that they did only compare their method to SDI in section 4.6. They argue that it is in general the strongest baseline but in fact it is outperformed by DCDI on the sachs and child dataset according to Table 9. These results should be reported in the main paper as well, especially since they do not need a lot of space.__
>
> Thank you for the suggestion, we have now moved the results of DCDI into the main text. The baseline was initially chosen based on the better performance of SDI especially for scalability, since we test on graphs with more than 400 variables. Still, the conclusion of the section, namely that ENCO is outperforming both baselines in various settings, remains the same.
>
>
> __For the paper to be truly outstanding, it would have been nice to see the method being applied to a real-world dataset. Although the true causal graph is not known in this setting, the plausibility of the result could be discussed in this case to evaluate the performance of the method.__
>
> Thank you for the suggestion, we surely agree that an application on real-world data where the ground truth graph is unknown, would be very interesting. Unfortunately, we are not aware of such datasets that are publicly available, fit our experimental settings, and are sufficiently intuitive to discuss the plausibility of the results. The graphs used in the experiments of Section 4.6 are derived from real-world use cases, such that these experiments give a direction of what we could expect in applications. Nonetheless, having actual real-world datasets would be an interesting future direction, which would need to be done in collaboration with researchers of the application domain (e.g. biology, economy, or other sciences).

---

> > ### Comment · Reviewer_H7r5 · 2021-11-17
> > **Response to Paper657 Authors**
> >
> > I acknowledge the extensive effort of the authors to improve the paper during this rebuttal phase. Given the helpful comments and all the improvements of the paper, I will stick to my rating although it is above the those of all the other reviewers, and I am in favour of accepting the paper.

---

> > > ### Author Response · Authors · 2021-11-17
> > > **Thank You for Your Response**
> > >
> > > Thank you for acknowledging the rebuttal and your support of the paper and in the process!

---

### Official Review · Reviewer_b5p5 · 2021-11-02

**Correctness:** 3
**Technical Novelty And Significance:** 3
**Empirical Novelty And Significance:** 4
**Recommendation:** 6
**Confidence:** 4

**Main Review:**

"Unfortunately, the solution space of DAGs grows super-exponentially with the number of variables. Current methods are thus typically limited to a few dozens of variables"
I am not sure if this reasoning is accurate since exhaustive search is not used.

"Yet, we show that under certain assumptions like being able to intervene on all variables, the proposed optimization is guaranteed to converge to the correct, acyclic graph"
This is not an insignificant assumption so it should be front and center in the introduction instead of being mentioned in passing.

"Thirdly, under mild conditions including interventions on all variables, we show that ENCO is guaranteed to converge to the correct causal graph"
Similarly, it is hard to justify this claim that this is a mild condition. I think it would be much more transparent to call the paper causal discovery from observational and interventional data - which describes the setting - rather than trying to present this as just another assumption.

"A DAG G and a joint distribution P are faithful to ea􏰍ch other"
Distribution is said to be faithful to a graph, not the other way around.

Important recent works that learn from obs+int data are missing from literature review under constraint-based methods:
- Mooij et al. "Joint causal inference from multiple contexts"
- Kocaoglu et al. "Characterization and Learning of Causal Graphs with Latent Variables from Soft Interventions"
- Jaber et al. "Causal Discovery from Soft Interventions with Unknown Targets: Characterization and Learning"

"Since the search space of DAGs is super-exponential in the number of nodes, many score-based approaches rely on a greedy search"
As far as I am aware of, there are consistency guarantees that these methods will output graphs in the true equivalence class. So it would be more accurate to mention this.

"We do not assume faithfulness"
This is a great addition.

"perfect" intervention is also called a "hard" intervention.

It would be nice to list all the assumptions in 3.1 under a numbered list. Specifically, authors assume that we have n interventional datasets for n variables where in each intervention a different node is intervened on (intervention size 1).

The discussion under "Graph fitting" on page 4 mentions that the method requires intervention on a node that is adjacent to the edge (although others might eventually be sufficient as well). However, if such an abundance of interventional data is available, one can simply learn each edge separately by contrasting two distributions and hypothesis testing without the need for a continuous optimization framework. Say X-Y and p_1 = p(y|x), p_2=p(y|do(x)) is available. Then we can check if p_1 and p_2 are the same or different to infer this edge. So essentially I am not sure if there is a need to leverage the power of continuous optimization or neural networks in this setting. More on this later.

The conditions of Theorem 3.1. seems to have been developed by essentially reverse-engineering the proof. In that sense, they are far from being intuitive about the data-generating conditions or system settings. Furthermore, due to this, it is not possible to assess whether these assumptions are realistic. This renders the theorem not really useful in my opinion and the authors can simply remove this unless the assumptions can be converted to intuitive ones descriptive of the setting irrespective of the proof.

The authors propose an interesting method to handle latent confounders. They have to assume that latent confounders cannot exist between adjacent variables. Even then though I am not sure if there is any guarantee because authors say "which is maximized by latent confounders" but I don't see a proof of this claim. It would help to clarify the exact claim here.

In the experiments, I am not sure why authors only compare with DCDI but not with the other mentioned methods.

Could you comment on whether the improvements seen in Section 4.3 are due to the new edge-wise detection framework (not having to enforce acyclicity constraint) or due to the better-variance estimator described in Section 3.3?

I really would like to see the performance of a very simple baseline:
- Learn the skeleton somehow (e.g. w/ GES)
- Orient each edge X-Y by learning the skeleton on the interventional distribution, say under an intervention on X. If X-Y still then X->Y else X<-Y. This is the basic idea that has been used in several constrained-based methods in the past when hard-interventional data is available. If only soft-interventional data is available one can check p_obs(y|x)==p_x(y|x) and orient X->Y if so and X<-Y otherwise. Of course, these require some sort of faithfulness but in settings with faithfulness, I think it is important to have these comparisons to clarify the contribution of the method. Perhaps your method is more sample-efficient and it would be great to demonstrate this empirically.

Finally, I am not absolutely sure if your method can avoid faithfulness. Suppose X->Y is the true graph but we have X\indep Y. Your (or any method really) will remove the edge between the two but that won't give the true graph. Any comments on this as well will be appreciated.

After the rebuttal:
--------------------
I would like to thank the authors for the additional experiments they conducted and the further explanations about their assumptions. Accordingly, I will increase my score. However, due to the shortcomings mentioned in my original review and the theoretical assumptions still being non-intuitive, I will not be able to recommend a strong acceptance. In case the paper goes through, please implement all the discussed changes, and thank you again for keeping an open mind and enabling a productive rebuttal period!

**Summary Of The Paper:**

The authors propose a differentiable causal discovery method from interventional data. Each edge is learned separately using a score-based approach that relies on Monte-Carlo sampling. The main difference is that - since edges are learned separately - acyclicity need not be enforced.

**Summary Of The Review:**

A new approach for causal discovery and some interesting ideas but the utility should be demonstrated by comparing with simple baselines since interventional data is assumed to be available.

---

> ### Author Response · Authors · 2021-11-16
> **Response to Reviewer b5p5 [1/4]**
>
> We thank the reviewer for the very detailed feedback and helpful suggestions to make the paper stronger. Please find our answers below.
>
> __However, if such an abundance of interventional data is available, one can simply learn each edge separately by contrasting two distributions and hypothesis testing without the need for a continuous optimization framework. Say $X$-$Y$ and $p_1 = p(y|x)$, $p_2=p(y|\text{do}(x))$ is available. Then we can check if $p_1$ and $p_2$ are the same or different to infer this edge. So essentially I am not sure if there is a need to leverage the power of continuous optimization or neural networks in this setting.__
>
> With a known skeleton of the graph and assuming faithfulness, simply checking for a change of $p(y|\text{do}(x))$ in comparison to $p(y|x)$ in the infinite data regime is indeed sufficient for determining the orientations of all edges. In practice, however, with limited data, the setup of ENCO brings three major benefits.
>
> Firstly, learning the skeleton from the combination of observational and interventional data is considerably more effective and efficient, especially when data is not truly infinite but limited.
> For instance, while correlations/noise in the observational data might suggest a causal relation among two variables, the interventional data can be used for verifying this relation beyond finding its orientation.
> This is integrated efficiently using continuous optimization in ENCO. The effects of it can also be seen in experiments we show in the next paragraph, comparing a baseline of finding the skeleton from observational data and orientating the edges from interventional data. While ENCO mostly identifies the graph without errors, the baseline approach already makes mistakes in the skeleton from observational data which cannot be recovered upon afterwards. Additionally, we can get around assumptions such as faithfulness by incorporating interventional data in the skeleton learning process (more on it further below).
>
> Secondly, neural networks are universal function approximators, or here distribution approximators, learned from data. They are indispensable when only limited data is available, in which case taking the normalized counts over samples, for example, is not sufficient for categorical distributions to get a decent estimate. This is especially important for an increasing number of possible conditionals, and no prior knowledge about the distributions is known.
>
> Finally, continuous optimization over the graph parameters makes it more efficient to take into account a larger parent sets than just the intervened variable $x$ for testing the orientation (i.e. comparing $p(y|x,\text{pa}(y)\setminus x)$ vs $p(y|\text{do}(x),\text{pa}(y)\setminus x)$). This makes differences in distributions much more apparent and, thus, more robust for smaller dataset sizes.
>
> __I really would like to see the performance of a very simple baseline. (1) Learn the skeleton somehow (e.g. w/ GES). (2) Orient each edge $X$-$Y$ by learning the skeleton on the interventional distribution, say under an intervention on $X$.__
>
> Thank you for the interesting suggestion. Current commonly available implementations of GES require continuous data and/or Gaussian-like distributions, which makes it not straight-forward to fairly compare such methods on the categorical datasets. Thus, instead, we performed the experiments on the continuous datasets of Appendix D.5 / Table 16. The datasets consist of graphs with 10 nodes, and 909 samples for the observational and per interventional dataset. We implement the suggested baseline by applying GES on the observational dataset to identify the original skeleton, and orient the edges by checking the skeleton found on the hard interventional datasets. More details can be found in Appendix D.5.
>
> We show results in the table below (next comment). The suggested baseline has the lowest performance, while ENCO attains the best accuracies. This highlights the benefits of incorporating interventional data in the learning of the skeleton and graph structure. Further, to show the impact with more and more limited data, we rerun the experiments of GES and ENCO-G (i.e. ENCO with networks predicting a Gaussian) on the same datasets, but only with 500/100 samples instead of the original 909. The GES baseline becomes noticeably worse, while ENCO is only mildly affected. We conclude that jointly learning from observational and interventional data as in ENCO provides a considerable performance gain over the baseline.

---

> > ### Author Response · Authors · 2021-11-16
> > **Response to Reviewer b5p5 [2/4]**
> >
> > | __Graph type__          | __Linear Gaussian__        | __Nonlinear with additive noise__      | __Nonlinear with non-additive noise__      |
> > |-------------------------|----------------------------|----------------------------------------|--------------------------------------------|
> > | GIES                    | $0.6$ ($\pm 1.3$)          | $9.1$ ($\pm 5.3$)                      | $4.4$ ($\pm 6.1$)                          |
> > | IGSP (best)             | $1.9$ ($\pm 1.8$)          | $5.3$ ($\pm 3.0$)                      | $4.1$ ($\pm 2.8$)                          |
> > | DCDI-G                  | $1.3$ ($\pm 1.9$)          | $5.2$ ($\pm 7.5$)                      | $2.3$ ($\pm 3.6$)                          |
> > | DCDI-DSF                | $0.9$ ($\pm 1.3$)          | $4.2$ ($\pm 5.6$)                      | $7.0$ ($\pm 10.7$)                         |
> > | GES + Orientating       | $2.5$ ($\pm 4.3$)          | $12.5$ ($\pm 9.2$)                     | $7.9$ ($\pm 7.4$)                          |
> > | ENCO-G (ours)           | $\mathbf{0.0}$ ($\pm 0.0$) | $\mathbf{0.2}$ ($\pm 0.4$)             | $\mathbf{0.1}$ ($\pm 0.3$)                 |
> > | ENCO-DSF (ours)         | $0.3$ ($\pm 0.7$)          | $1.4$ ($\pm 1.4$)                      | $1.2$ ($\pm 1.5$)                          |
> > | GES + Orientating (500) | $3.2$ ($\pm 4.3$)          | $12.0$ ($\pm 9.2$)                     | $9.2$ ($\pm 7.5$)                          |
> > | ENCO-G (ours) (500)     | $\mathbf{0.1}$ ($\pm 0.3$) | $\mathbf{0.5}$ ($\pm 0.7$)             | $\mathbf{0.0}$ ($\pm 0.0$)                 |
> > | GES + Orientating (100) | $6.1$ ($\pm 5.3$)          | $10.2$ ($\pm 6.7$)                     | $9.0$ ($\pm 6.5$)                          |
> > | ENCO-G (ours) (100)     | $\mathbf{0.2}$ ($\pm 0.4$) | $\mathbf{0.9}$ ($\pm 1.1$)             | $\mathbf{1.3}$ ($\pm 0.8$)                 |
> >
> >
> >
> >
> > __The conditions of Theorem 3.1. seems to have been developed by essentially reverse-engineering the proof. In that sense, they are far from being intuitive about the data-generating conditions or system settings.__
> >
> > Please refer to the general response for a more detailed discussion on the conditions in Theorem 3.1. In general, the conditions in Theorem 3.1 guarantee the absence of local optima which makes them less intuitive and seem closer related to the proof strategy, but go beyond ensuring the global optimum being the correct graph. For the global optimum, the conditions are much more intuitive. For instance, the conditions on the orientation parameters translate to each ancestor-denscendant pair being non-independent when intervening on the parent. Further, for the condition with respect to sparsity, the regularizer weight $\lambda_\text{sparse}$ must be selected small enough to only remove edges between conditionally independent variables.
> >
> >
> > __The authors propose an interesting method to handle latent confounders. They have to assume that latent confounders cannot exist between adjacent variables. Even then though I am not sure if there is any guarantee because authors say "which is maximized by latent confounders" but I don't see a proof of this claim. It would help to clarify the exact claim here.__
> >
> > The proof for the claim can be found in Appendix B.3. Specifically, we can prove that, under the same assumptions as for the convergence guarantees and ones we specify below, the proposed score $\text{lc}(X_i,X_j)$ converges to one for all variable pairs where $X_i$ and $X_j$ have a latent confounder, and zero for all others. The additional assumptions we need to take are, on an intuitive level, that the dependency between the two variables $X_i$ and $X_j$ must be greater than the chosen sparsity regularizer. Otherwise, the edges between $X_i$ and $X_j$ are removed, and we don't detect the confounder, but at the same time, don't introduce any errors in the graph for other nodes.
> >
> > For cases where $X_i$ indirectly causes $X_j$ (i.e. $X_i$ being an ancestor but not direct parent of $X_j$), we cannot strictly guarantee a perfect convergence of $\text{lc}(X_i,X_j)$ to one because $\sigma(\theta_{ij})$ (the orientation) might converge faster to one than $\sigma(\gamma_{ji}^{(O)})$ (the edge existence of $X_j\to X_i$). Nonetheless, in practice, this is less relevant since it optimizes to a value close to one, and in the shown experiments, almost 60\% of the latent confounders were between ancestors and descendants, which were yet well detected.

---

> > > ### Author Response · Authors · 2021-11-16
> > > **Response to reviewer b5p5 [3/4]**
> > >
> > > __Could you comment on whether the improvements seen in Section 4.3 are due to the new edge-wise detection framework (not having to enforce acyclicity constraint) or due to the better-variance estimator described in Section 3.3?__
> > >
> > > We have included ablation studies on the importance of the gradient estimators in Appendix D.3, Table 14. On graphs of 25 variables, such as those from the experiments in Section 4.2, ENCO still achieves good performance when using the gradient estimators of previous works with higher variance. However, when looking at the large graphs of Section 4.3, the effect of the lower variance estimator becomes more important. Within the 30 epochs, the model without the proposed gradient estimator obtained an SHD of 15.4 on average over 5 experiments, which is considerably higher than with the proposed gradient estimators (0.0). Still, this is only half of the errors that SDI (Ke et al., 2019) achieved with the same gradient estimator. Hence, we conclude that the proposed gradient estimators are beneficial for ENCO but not strictly necessary for small graphs. For large graphs, the low variance of the estimator becomes much more important.
> > >
> > >
> > > __Finally, I am not absolutely sure if your method can avoid faithfulness. Suppose $X\to Y$ is the true graph but we have $X$ indep $Y$. Your (or any method really) will remove the edge between the two but that won't give the true graph. Any comments on this as well will be appreciated.__
> > >
> > > We clarify that we do not claim that ENCO can accurately model all graphs where faithfulness is violated, but that faithfulness is not strictly necessary for our algorithm.
> > > In the provided example, there is no indication of $X$ and $Y$ being causally related, which would also violate condition 2 in Theorem 3.1.
> > >
> > > However, while graphs with deterministic variables violate faithfulness, ENCO can still model them. We refer to Appendix B.5 for a detailed discussion of a simple example graph.
> > > We also empirically verify that ENCO can be applied to cases where faithfulness does not hold, by including experiments on graphs with deterministic variables in Appendix D.4. The results show that ENCO still performs well with an average SHD of 1.4 on graphs with 25 nodes, where all causal mechanisms except the ones of root nodes are deterministic.
> > >
> > >
> > > __In the experiments, I am not sure why authors only compare with DCDI but not with the other mentioned methods.__
> > >
> > > DCDI is the closest to our method as a continuous optimization method with an explicit acyclicity regularizer. We also compare to GIES, IGSP and SDI in multiple experiments. However, for instance for the partial intervention case, we only compare to DCDI since SDI does not support a smaller intervention set, and GIES and IGSP showed already considerably lower performance in the previous experiments. One reason for that is the categorical nature of the data we experiment on, as, for instance, IGSP uses hypothesis tests like in constraint-based method in its search algorithm. Categorical data without any known structure of its distribution is difficult to work on for those tests since the conditional distributions can only be estimated very sparsely, as we have experienced from our experimentation with IGSP. Further, most common implementations of constraint-based method are only designed for continuous data. Thus, we restricted the experimental comparison to the shown methods.

---

> > > > ### Author Response · Authors · 2021-11-16
> > > > **Response to reviewer b5p5 [4/4]**
> > > >
> > > > __Clarifications/Corrections in the text__
> > > >
> > > > Thank you for your suggestions to clarify several aspects of the paper in terms of writing. We list below the changes made we made accordingly.
> > > >
> > > >
> > > > * _Exponential DAG search space - I am not sure if this reasoning is accurate since exhaustive search is not used._ We rephrased this sentence to make it clearer that exhaustive search is not needed, but efficient methods are key for searching such a large space.
> > > > * _Stating assumptions more clearly in the introduction_ Our intention was not to downplay or hide those aspects. To make them more apparent, we clearly state it in the abstract and make it more prominent in the introduction, as well as reiterating that all theoretical guarantees assume interventions on all variables. We want to note, however, that even interventions on all variables are needed only for the theoretical guarantee. In experiments, the algorithm works fine also when intervening on subsets.
> > > > * _Distribution is said to be faithful to a graph, not the other way around._ This is now corrected in Section 2.1.
> > > > * _Important recent works that learn from obs+int data are missing from literature review under constraint-based methods_ We have added the works in the respective paragraph of the related work.
> > > > * _Greedy score-based methods - As far as I am aware of, there are consistency guarantees that these methods will output graphs in the true equivalence class._ We have added it in the corresponding sentence.
> > > > * _List of assumptions in 3.1_: We have rephrased this section to have a more explicit list of the assumptions.
> > > >
> > > >
> > > >
> > > > __References__
> > > >
> > > > [Brouillard et al., 2020] P. Brouillard, S. Lachapelle, A. Lacoste, S. Lacoste-Julien, A. Drouin. Differentiable Causal Discovery from Interventional Data. In Proceeding of the 34th Conference on Neural Information Processing Systems, 2020.
> > > >
> > > > [Kalainathan et al., 2020] Kalainathan, D., Goudet, O., and Dutta, R. (2020). Causal Discovery Toolbox: Uncovering causal relationships in Python. J. Mach. Learn. Res., 21, 37-1.

---

### Author Response · Authors · 2021-11-16
**General Response [1/2]**

We would like to thank all reviewers for their detailed and valuable feedback. We take the detailed reviews and suggestions as a sign of genuine interest, and welcome the opportunity to clarify all points raised. Based on the reviews, we have done the following main edits to the paper:

* Section 2.2 (related work): Clarified the difference between causal discovery from observational and observational+interventional data, and added additional works for constraint-based methods.
* Section 3.2 (method overview): Clarified the role of the sparsity regularizer.
* Section 3.4 (convergence guarantees): Rewrote the text around Theorem 3.1 to focus more on the intuition and practical relevance (more on it below).
* Section 4.6 (real-world inspired experiments): Moved the results of DCDI on the real-world inspired datasets from the appendix to the main paper.
* Appendix B.2.4 (convergence guarantees): Added a discussion of the global optimum being the ground-truth graph (more on it below).
* Appendix B.5 (faithfulness): Added a walkthrough of a causal model that violates faithfulness, but can yet be recovered by ENCO.
* Appendix D.5 (experiments on continuous data): Added a constraint-based baseline to the experiments on continuous data.


As a second part of this response, we take the opportunity to clarify below common questions on the convergence guarantees given in Theorem 3.1, especially with the focus on the intuition behind them. The conditions are very close to the proof strategy and the algorithm/gradient estimators, since we prove for a gradient-based algorithm that no local optima exist. A much more intuitive perspective can be given when we consider the conditions for when the global optimum is the correct graph, since it requires fewer constraints. Hence, we start by discussing the difference between the local and global optima conditions, then go to the motivation and intuition behind the convergence guarantees in Theorem 3.1, and finally discuss what a violation of these conditions imply in practice.

__Local vs Global optimum__

Most existing work on continuous optimization methods like DCDI (Brouillard et al., 2020) prove that the global optimum corresponds to the ground truth graph. However, this does not necessarily imply that a gradient-based optimization will return the optimal solution, as it could get stuck to a bad local optimum, especially with biased gradient estimators or similar.

By contrast, the assumptions and conditions in Theorem 3.1 not only show when the ground truth graph is the global optimum of the objective in Equation (2), but further guarantee that no other _local_ optimum exists with respect to the graph parameters $\mathbf{\theta}$ and $\mathbf{\gamma}$.
In other words, ENCO will always converge to the single, correct global optimum under these conditions, as ENCO is a gradient-based algorithm, and the proposed low-variance gradient estimators are unbiased.
This explains also the superior empirical performance of the algorithm in such cases.

In this discussion, of course, we focus on the parameters $\mathbf{\theta}$ and $\mathbf{\gamma}$ when talking about the global optimum. That is, we do not take into account the training of the neural networks on the observational data. This is not a problem both in theory and practice, however: with a sufficiently large network, we observed no issues.

__Intuition behind conditions__

We start with the more straightforward case by only considering the global optimum. Namely, if we drop the requirement of having no local minima, and focus on the global optimum being the true graph -as is anyway the case for the previous setting-, the conditions in Theorem 3.1 can be intuitively simplified as follows.

To learn the correct orientation of the edges, _conditions 1 and 2_ state that ancestors and descendants in the graph have to be dependent when intervening on the ancestors. If they are not, we cannot identify the impact of a parent on its descendants. This aligns with the technical interpretation in Theorem 3.1, specifically condition 1 and 2, that the likelihood estimate of the child variable must improve when intervening and conditioning on its ancestor variables, thus ensuring the convergence towards the true orientation parameters.

According to _condition 3_, the sparsity regularizer needs to be selected such that it chooses the sparsest graph among those graphs with equal joint distributions as the ground truth graph, without trading sparsity for worse distribution estimates.
We emphasize that this condition only gives an upper bound for $\lambda_\text{sparse}$ when sufficiently large datasets are available. Setting a low enough $\lambda_\text{sparse}$ typically works fine in practice. The same holds for condition 3 in Theorem 3.1, which states that we can learn the parent set by successively adding or removing them once we know the orientations, hence making it possible for gradient-based algorithms to find the global optimum.

---

> ### Author Response · Authors · 2021-11-16
> **General Response [2/2]**
>
>
> We include a further discussion and proof for the statements for the global optimum in Appendix B.2.4.
>
> __Violations of conditions__
>
> The conditions in Theorem 3.1 are sufficient but not necessary, in the sense that the predicted graph returned by ENCO is not necessarily an incorrect one when not all conditions apply.
> Still, when these conditions do not apply, we cannot guarantee that the algorithm will _always_ converge to the correct graph. The reason is that there might be local minima in the optimization, and our gradient-based algorithm might converge to one of those.
> Next, we provide an intuition what mistakes this could imply.
>
> _Violating condition 1_, causal models can lead to local optima solutions, in which some edges are incorrectly orientated. This most commonly only happens when very limited data is available. For example, in sparse graphs like chains $X_1\to X_2\to ...\to X_n$, we experienced that due to inevitable biases in small sample sizes, a variable was not always independent of its parents or ancestors when intervened on. This violates condition 1 (more specifically its version for limited data in Appendix B.2.2) and can cause ENCO to predict an edge from a descendant to one of its ancestors, e.g. $X_4\to X_1$ in the chain. Nonetheless, most of those edges can be removed by the post-processing step for enforcing acyclicity, since these are outliers for single variables.
>
> _Violating condition 2_ would mean that a child is independent of its parent under interventions of the parent. This leads to the respective edge being missed in the prediction. However, this would require intervention distribution with no support for a subset of values, or causal relations that cannot be detected from data in general.
>
> _Violating condition 3_ alone will yet guarantee that a graph with correct orientations of all edges is returned. Still, we might end up with a different, smaller set of parents for some variables that describe the distribution of the respective variable also well enough. A common mistake, such as in the graph alarm in Section 4.6/Table 3, is that single edges with low influence of a parent on its child are missed. However, in practice, missing out such edges might not be critical. The reason is that these missed ancestor variables would de facto not influence the respective child variable much.
>
> We appreciate the feedback that helped us refine the intuition behind our proposition and improve the clarity of the paper.
>
> __References__
>
> [Brouillard et al., 2020] P. Brouillard, S. Lachapelle, A. Lacoste, S. Lacoste-Julien, A. Drouin. Differentiable Causal Discovery from Interventional Data. In Proceeding of the 34th Conference on Neural Information Processing Systems, 2020.

---

### Decision · Program_Chairs · 2022-01-20

**Decision:**

Accept (Poster)

**Comment:**

This paper studies the problem of learning a graphical model given observational and experimental data. The main novelty is the use of interventions to avoid the acyclicity constraint that plagues existing methods. Although this idea is quite standard and well-known, the generality of the approach merits consideration. After the discussion, there was a consensus among the reviewers to accept this paper. Some valid concerns have been raised and we expect that the authors will take into account all of the suggestions raised by the reviewers.